# The EarthCARE lidar cloud and aerosol profile processor (A-PRO): the A-AER, A-EBD, A-TC and A-ICE products

David Patrick Donovan[1], Gerd-Jan van Zadelhoff[1], and Ping Wang[1]

[1]Royal Netherlands Meteorological Institute (KNMI), de Bilt, the Netherlands

**Correspondence:** D.P. Donovan (donovan@knmi.nl)

**Abstract.** ATLID ("ATmospheric LIDar") is the lidar flown on the multi-instrument Earth Clouds and Radiation Explorer (EarthCARE). EarthCARE is a joint ESA/JAXA mission which was launched in May 2024. ATID is a 3 channel, linearly polarized , High-Spectral Resolution Lidar (HSRL) system operating at 355 nm. Cloud and aerosol optical properties are key EarthCARE products. This paper provies an overview of the ATLID level-2a (L2a, i.e. single instrument) retrieval algorithms being developed and implemented in order to derive cloud and aerosol optical properties. The L2a lidar algorithms that retrieve the aerosol and cloud optical property profiles and classify the detected targets are grouped together in the so-called A-PRO (ATLID-profile) processor. The A-PRO processor produces the ATLID L2a Aerosol product (A-AER), the Extinction, Bakscatter and Depolarization product (A-EBD), the ATLID L2a Target Classification (A-TC), and the ATLID L2a Ice microphysical estimation product (A-ICE). This paper provides and overview of the processor and its component algorithms.

## 1 Introduction

The Earth Clouds and Radiation Explorer mission (EarthCARE) (Illingworth et al., 2015) is a multi-instrument cloud-aerosol-radiation process study oriented mission embarking a high spectral resolution ATmospheric lidDAR (ATLID), a Doppler Cloud Profiling Radar (CPR), a Multi-Spectral Imager (MSI), and a three-view Broad-Band Radiometer (BBR). EarthCARE will measure the global height resolved distribution of clouds, aerosols, and precipitation and estimate their macro, microphysical and radiative properties (Wehr et al., 2023).

This document describes the algorithms within the EarthCARE L2a ATLID profile processor (A-PRO). Within the Earth-CARE project, single-instrument geophysical property retrievals are referred to as level-2a (L2a) retrievals (Eisinger et al., 2023). A-PRO has been developed with support from the European Space Agency (ESA) for specific application to ATLID (do Carmo et al., 2021) and comprises a number of new developments. Within this processor, four main sub-algorithms exist: a procedure aimed at deriving the large-scale aerosol (and thin cloud) extinction and backscatter (A-AER), an optimal estimation based Extinction and Backscatter retrieval algorithm (A-EBD), a lidar target classification procedure (A-TC), and an ice microphysical property estimation procedure (A-ICE). Output products corresponding to each of these component procedures are generated. Collectively, these algorithms produce multi-horizontal-resolution profiles of lidar extinction, backscatter, optical depth, particle type, ice effective radius, ice water content as well as target type (e.g. cloud phase, aerosol-type etc.). This paper

presents the theoretical background of the algorithms that comprise the A-PRO processor as well as presenting and discussing various examples. The examples shown are based on the simulations described in Donovan et al. (2023).

## 1.1 ATLID

ATLID is a linearly polarized three-channel lidar operating at 355nm. The vertical resolution of the return signal is about 100 m throughout most of the atmosphere. The pulse-repetition-frequency (PRF) is 51 Hz and nominally it is planned that two

shots will be averaged on-board giving a horizontal resolution on the order of 305 m. The lidar delivers profiles of the parallel particulate attenuated backscatter (ATB), the Rayleigh (molecular) attenuated backscatter and the perpendicular particulate attenuated backscatter.

ATLID is a so-called High Spectral Resolution Lidar (HSRL) (Eloranta, 2005). The first successfully operating HSRL system emphasising aerosol and cloud sensing was flow in 2022 (Liu et al., 2024; Dai et al., 2024). The Aerosol and Carbon

Detection Lidar (ACDL), on board the Atmospheric Environment Monitoring Satellite known as Daqi-1 (DQ-1) uses an atomic Iodine filter technique at 532 nm (Dong et al., 2018) to separate molecular and particulate backscattering. A HSRL lidar that previously operated (2018-2023), at the same wavelength as ATLID, was Aeolus (Straume, A.G. et al., 2020). Aeolus, unlike ATLID, was primarily oriented to the retrieval of line-of-sight winds, however, aerosol and cloud algorithms were also developed and applied to Aeolus data (Flament et al., 2021; Ehlers et al., 2022). Aeolus cloud and aerosol products are limited

by the characteristic of Aeolus (e.g. limited vertical resolution and the lack of a depolarization channel). Never-the-less useful aerosol and cloud products can be retrieved. Indeed, some of the techniques originally developed for ATLID and described in this paper have been successfully 'back-ported' to Aeolus (Wang et al., 2024). Unlike Aeolus, ATLID has been optimized exclusively for aerosol and cloud sensing. Thus, even more useful aerosol and cloud products are expected from ATLID.

ATLID, uses a Fabry-Pérot etalon to (imperfectly) separate the spectrally narrow return from aerosols and clouds ('Mie')

from the thermally broadened return from atmospheric molecules ('Rayleigh'). Note that Mie scattering, properly, refers only to scattering by perfect spheres. In this paper (and indeed through much of EarthCARE related documentation) the term is used rather loosely to broadly cover what should be termed 'particulate' scattering. Further, the term 'Rayleigh' scattering is also used loosely. A more accurate term would be 'molecular' or 'Rayleigh-Brillouin' scattering. Since the Rayleigh and Mie signal separation is imperfect, a degree of cross-talk between the channels exists. In order to separately quantify the pure Mie

and Rayleigh scattering contributions a cross-talk correction procedure is applied as part of the L1 processing. ATLID emits linearly polarized light and separates the returned backscatter into components polarized parallel and perpendicular to the plane defined by the emitted beam. The polarization separation comes before the HSRL spectral filter. Accordingly, the three ATLID physical channels are:

- A parallel (or co-polar) Mie channel.

- A parallel (or co-polar) Rayleigh channel.

- A perpendicular (or cross-polar) channel.

For details of ATLID's design see do Carmo et al. (2016) and do Carmo et al. (2021). Some of the important ATLID technical specifications are repeated in Table 1.

| Parameter | Value |
|---|---|
| Telescope Diameter (m) | 0.62 |
| Wavelength (nm) | 355 |
| Receiver field-of-view (full-angle) (mads) | 66.5 |
| Laser Divergence (full-angle) (mrads) | 36 |
| Pulse Energy (mJ) | 35 |
| Range resolution (m) | 100m(0-20km)/500m(20–40km) |
| PRF (Hz) | 51 |
| End-of-Life Parallel-Mie transmission | 45 |
| End-of-Life Perpendicular-Ray transmission | 43 |
| End-of-Life Cross-polar transmission | 43 |
| Quantum Efficiency | 79/75/79 % |
| Molecular backscatter in Mie parallel channel fraction | 25 % |
| Mie backscatter in Rayleigh perpendicular channel fraction | 16 % |

**Table 1.** ATLID Technical Specifications

A simple depiction of the ATLID receiver operation and level-1 (L1) processing is presented in Figure 1. Here the left-section of the figure depicts the hardware components, while the right-section depicts the application of the cross-talk correction and calibration procedures carried out by the L1 processing software. The perpendicular detection channel measures the depolarized signals from a combination of molecular Rayleigh and particulate scattering, however, the level-1 (L1) ATLID processor spectral-polarization crosstalk correction procedure applied to the detected signals delivers the perpendicular signals due to particulate scattering only. The cross-talk monitoring and correction procedures use a combination of averaged high-altitude (e.g. 30-40 km) pure molecular Rayleigh scattering returns, (non-water) surface returns, as well as suitable cloud returns to determine the relative amount of cross-talk present in each channel. Once the cross-talk coefficients are specified, absolute calibration is linked mainly to the use of high altitude (e.g. 30-40 km) pure molecular Rayleigh scattering returns. Calibration and cross-talk correction issues are not described further here. It is anticipated that these specific topics, however, will be the subject of a detailed post-launch publication.

After spectral and polarization cross-talk correction and calibration, the ATLID attenuated backscatter coefficient profiles can be related to the atmospheric extinction and backscatter coefficients (neglecting multiple-scattering effects for the time being) as:

$$b_R(z) = \beta_R(z) \, \exp\left[-2 \int_{z_{lid}}^{z} (\alpha_M(z') + \alpha_R(z'))dr(z')\right] \tag{1}$$

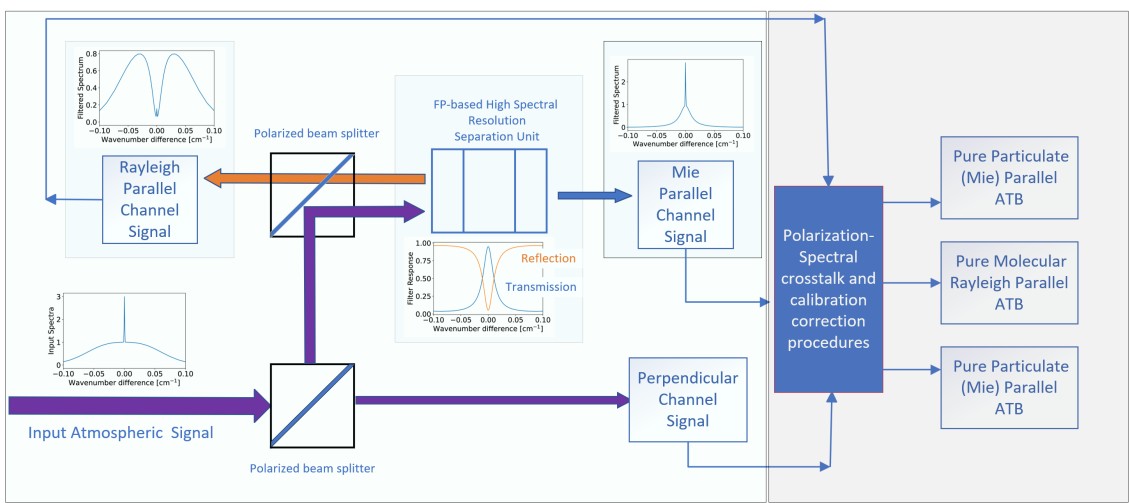

**Figure 1.** Simple schematic depiction of the ATLID receiver and L1 processing.

$$b_{M,\parallel}(z) = \beta_{M,\parallel}(z) \, \exp\left[-2 \int_{z_{lid}}^{z} (\alpha_M(z') + \alpha_R(z')) dr(z')\right] \tag{2}$$

$$b_{M,\perp}(z) = \beta_{M,\perp} \exp\left[-2 \int_{z_{lid}}^{z} (\alpha_M(z') + \alpha_R(z')) dr(z')\right], \tag{3}$$

where $b_R$ is the Rayleigh attenuated backscatter, $b_{M,\parallel}$ is the parallel Mie attenuated backscatter, $b_{M,\perp}$ is the perpendicular Mie attenuated backscatter, $z$ is the atmospheric altitude and $r(z)$ is the range from the lidar. The corresponding $p$ terms represent the detected power. $\alpha_M$ is the aerosol and cloud extinction and $\alpha_R$ is the atmospheric Rayleigh extinction. $\beta_{M,\parallel}$ is the parallel Mie backscatter, $\beta_R$ is the Rayleigh backscatter, and $\beta_{M,\perp}$ is the perpendicular Mie backscatter. Referring to Eqs. (2) and (3) the total Mie attenuated backscatter is given by

$$b_M = b_{M,\parallel} + b_{M,\perp}. \tag{4}$$

In general, for space-based lidars, multiple-scattering can be an importance contribution to the detected signals and must be accounted for (Winker, 2003). In this work, a novel approach has been used which lies in terms of speed and accuracy between the simple effective extinction approach due to Platt (1981) and the approach of Hogan (2008). The multiple scattering formalism used in this work is detailed in Appendix B.

The primary function of the ATLID Level-2 (L2) A-PRO processor is to invert the lidar signals to obtain estimates of backscatter and extinction, and using these values together with the particle linear depolarization ratio ($\delta_M = \left(\beta_{M,\perp}/\beta_{M,\parallel}\right) = \left(b_{M,\perp}/b_{M,\parallel}\right)$) in order to classify the detected targets.

## 2   A-PRO retrieval processor

In principle, HSRL retrievals can yield direct estimates of extinction and backscatter profiles (Eloranta, 2005), however, the direct method for estimating the backscatter involves calculating the ratio of the Mie (Eqs. 4) and Rayleigh (Eq. 1) signals while the extinction estimation involves taking the range derivative of the logarithm of the Rayleigh signal (Eq. 1). Both these mathematical operations are sensitive to noise, particularly when small (or even possibly negative) values may be present in the Rayleigh channel. Thus, direct inversions are only practical when the data is of a suitably high Signal-to-Noise ratio (SNR).

The SNR of the attenuated HSRL backscatter signals can be increased by along-track averaging of the signals. However, this can produce, at best, biased, and at worst, ambiguous or non-physical results, if, for example, 'strong' (e.g. cloud) and 'weak' signals are averaged indiscriminately together. Thus, any averaging of the signals must respect the structure of the atmospheric scene being probed. Aerosol fields may be homogeneous enough and the signals weak enough that averaging along track for several tens of kilometers may be justified. On the other hand, cloud returns may be strong and inhomogeneous to the point that it is desirable to apply inversions on the finest available resolution.

What is required is a means to guide the averaging the signals when appropriate and a multi-scale approach for retrieving optical properties and target classification for both aerosols and clouds. The A-PRO processor structure is designed with such goals in mind.

### 2.1   General structure

A-PRO is divided into three main algorithms as depicted in Fig. 2.

The main inputs to A-PRO are the:

**L1 ATLID product**  Calibrated cross-talk corrected attenuated backscatter (ATB) profiles.

**X-JSG Auxiliary data product**  The Joint Standard Grid product facilitates the co-location of the EarthCARE L2 products with one and other (Eisinger et al., 2022). The X-JSG essentially provides a common coordinate system for the Earth-CARE instruments. In addition, the JSG contains explicit information used to map the indices of the ATLID L1 data to the common grid. This strategy was adopted in order to help insure consistent geolocation and to e.g. avoid the need for downstream processors (which may combine ATLID and/or CPR and/or MSI L1 or L2 products) to perform their own regridding. The X-JSG horizontal resolution is about 1 km and the vertical resolution follows the ATLID vertical grid.

**X-MET Auxiliary data product**  X-MET contains the atmospheric pressure, temperature, etc.. built using ECMWF forecast data (Eisinger et al., 2022).

**A-FM L2 product**  The ATLID Feature-Mask provides a high-resolution mask of detected targets. The use of A-FM helps facilitate the appropriate averaging of the data. A-FM uses a combination of image processing techniques in order to identify regions of clouds/aerosols, surface returns, clear air, or attenuated regions. The detected aerosol/cloud regions are separated into cloud phase and aerosol type later in subsequent processing steps.

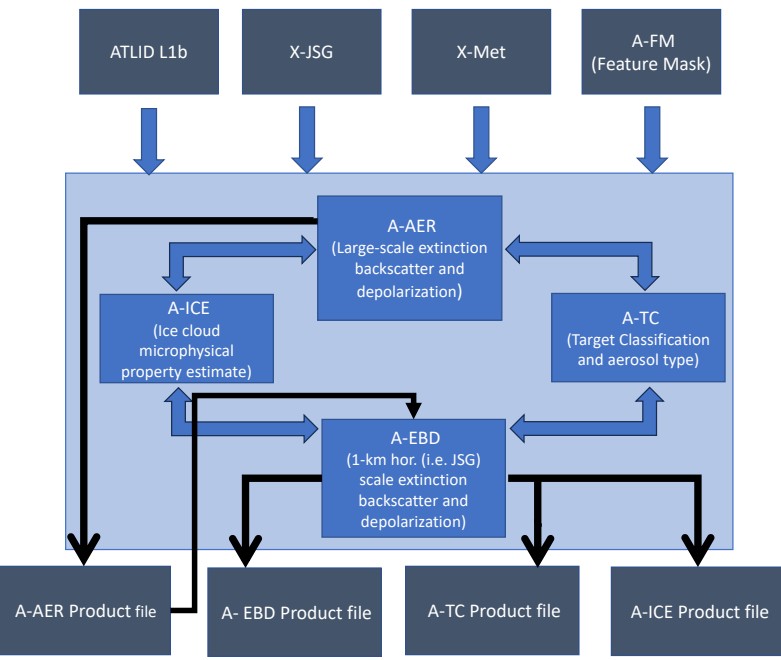

**Figure 2.** Schematic depiction of the high-level structure of the A-PRO processor. The top dark-Grey boxes represent the input products while the bottom dark-Grey boxes represent the output products. The double-headed arrows are used to indicate that the data-flow is bi-directional between the main procedures (A-AER and A-EBD) and the A-ICE and A-TC related procedures.

The A-FM product variable most relevant for A-PRO is the featuremask index with ranges between 0-to 10 (for non attenuated and non-surface pxiels) which is based on the probability of a cloud/aerosol target being present in the particulate backscatter. Thus, the index is a reflection of the SNR of the Mie attenuated backscatter. Thus, higher indices are usually being associated with e.g. thick clouds and lower indices are usually associated associated with e.g. optically thinner aerosols. Attenuated regions are identified using the Rayeligh channel.

The featuremask index is produced using a combination of the iterative application edge preserving median-hybrid filters (Russ, 2007) of different dimensions (e.g. 11 horizontal pixels $\times$ 11 vertical pixels and $11 \times 3$ pixels), the removal of detected features, then the iterative application (with up to 200 iterations) of a Gaussian smoothing kernel (with a Gaussian width of e.g. 13 by 1.5 pixels) together with a noise-level estimation procedure. Features are detected and recorded for a fixed sequence of iterations (e.g. after 15, 70, 140 and 200 iterations. Thus, A-FM output are inherently "multi-scale" but the approach is, in a sense, more continuous due to the iteration convolution approach, compared to e.g. the fixed averaging interval strategy used by the CALIPSO Vertical Feature Mask algorithm (Vaughan et al., 2009). Details of A-FM can be found in van Zadelhoff et al. (2023)

Within A-PRO the following main procedures are present.

**Aerosol oriented extinction and backscatter retrieval (A-AER)** This procedure uses direct HRSL retrieval methods for determining extinction and backscatter large horizontal e.g. 50km+ horizontal scales (e.g. deriving the extinction based on the log-derivative of the Rayleigh signal (Eloranta, 2005)). In order to do this the lidar signals must be appropriately masked and averaged to achieve a target SNR while avoiding mixing e.g. clouds and aerosol together. The averaging mask originates, in part, from the A-FM output which is used to avoid averaging "strong" and "weak" features together. The methods used by A-AER to estimate extinction and backscatter are described in Section 2.2 and Appendix A. A-AER outputs are directly used by the high horizontial resolution A-EBD component of A-PRO.

**Cloud and aerosol Extinction, Backscatter and Depolarization procedure (A-EBD)** This routine retrieves the aerosol and cloud extinction and backscatter profiles at the 1-km horizontal scale. At this scale, the SNR of the molecular scattering channel return is too low to enable the techniques employed by the A-AER approach. Instead, the method relies on a forward-modelling Optimal Estimation (OE) approach. As a priori information, the lidar-ratio (S) estimates produced by A-AER are used as inputs in A-EBD. In order to deal with the generally low SNR of the lidar data at high horizontal resolutions and to improve the run-time of the algorithm A-EBD operates on a layer-by-layer basis. That is, for each aerosol of cloud layer (which may span a number of range-gates), the lidar-ratio e.g. is assumed to be constant. A-EBD is described in detail in section 2.3.

**Target classification procedure (A-TC)** A-TC uses extinction, backscatter and depolarization ratio, as well as auxiliary inputs such as ECMWF forecast temperature, in order to classify targets into classes such as water or ice cloud or aerosol type. The aerosol typing scheme is based primarily on using the lidar-ratio and particle depolarization ratio to assign the aerosol to a type (Wandinger et al., 2023) . The cloud phase determination scheme uses layer integrated backscatter and depolarization in a manner similar to that employed for the CALIOP retrievals (Hu et al., 2009).

**Ice Microphysical property Estimation (A-ICE)** A-ICE employs a simple parameterization approach for estimating ice cloud effective radius and ice-water content (IWC) using retrieved extinction values. In particular, an empirical parameterization based on in-situ observations which uses temperature and extinction is employed (Heymsfield et al., 2014a).

The above components work in a cooperative fashion. A-AER provides a first-pass focusing on the optically thin targets and A-EBD performs another pass to retrieve both the optically thick and thin targets using A-AER output as input. A-TC and A-ICE component procedures are called by both A-AER and A-EBD.

## 2.2 A-AER

The A-AER procedure retrieves the large-scale optical properties of the optically thin regions (weak-features) and sets the stage for the high resolution A-EBD stage of the A-PRO procedure. Thus, in addition to providing estimates of extinction and lidar-ratio, A-AER also provides the column-by-column layer structure and a priori classification used as input by E-EBD. The inputs are the L1b ATLID data (attenuated cross-talk corrected backscatters for the three ATLID channels), the ATLID

featuremask (A-FM) product (van Zadelhoff et al., 2023), the auxiliary X-JSG (Joint standard Grid multi-instrument grid definition), and the X-MET (ECMWF supplied meteorological fields) product (Eisinger et al., 2022).

### 2.2.1 Pre-processing

Due to the generally low SNR of lidar signals at the few kilometer horizontal scale, before the A-AER procedure can derive quantitative extinction and lidar-ratios and determine the layering structure and layer classification, averaging of the input L1 attenuated backscatters is necessary. This averaging, however, must respect the structure of the observations to avoid e.g. averaging both cloud and aerosol regions together. Thus, a large part of what A-AER does involves preprocessing the attenuated backscatters based on the observed structure of the observations themselves before final quantitative values of extinction and
lidar-ratio are derived. This "pre-processing" is schematically described in Figure 3 which corresponds to steps 1 to 7 in the more detailed overall A-AER algorithm flowchart shown in Figure 4 described in section 2.2.2.

Referring to Figure 3, first the ATBs are binned according to the JSG definition (A). Then the A-FM data is also binned to the X-JSG (B). The X-JSG binned A-FM index is then used to separate "strong" and "weak" features by applying a threshold in order to create a mask (C). The mask is then used to separately smooth the strong and weak feature ATBs using a box-car
window (typically 40 horizontal $\times$ 1 or 3 vertical pixels) (see also step 3 of Figure 4). After smoothing, the strong and weak fields are merged to produce the "Hybrid smoothed ATBs" (E) which have improved SNRs compared to the input signals but the main features have not been blurred out.

The Ray and Mie hybrid smoothed ATBs generally have associated SNRs suitable for the useful calculation of backscatter and scattering-ratio but not for the direct determination of extinction or lidar-ratio. Thus, further guided smoothing is needed.
The A-FM indices could be used directly as a guide, however, this was found to be problematic since low index values can be produced by attenuation rather than the absence of a significant target. Accordingly, the Ray and Mie hybrid smoothed ATBs are then used to estimate the lidar scattering ratio (F), i.e.

$$R(z) = \frac{\beta_{Mie}(z) + \beta_{Ray}(z)}{\beta_{Ray}(z)} = \frac{b_M(z) + b_R(z)}{b_R(z)} \tag{5}$$

. By using $R$ for further masking we will have, in a sense, accounted for attenuation in the smoothing mask definition procedure.
To create the mask used in the next averaging steps, an atmospheric density dependent threshold is applied to the scattering ratio (G). The masked and unmasked data fields are then separately horizontally averaged (up to e.g. 50 km) and merged (H) and finally vertically averaged (e.g. by 1 to 4 km) (I). More detail on this process is given in the description associated with steps 5 and 6 of Figure 4.

### 2.2.2 Full algorithm description

A flow-chart depiction of the main elements of the A-AER procedure is presented in Figure 4. The inputs are represented by the upper right parallelograms.

In **step 1** the level-1 ATBs are rebinned according to the the JSG grid. At the same time, the associated errors in the ATB profiles are generated either by calculating the standard deviation of the ATBs or quadraticlly summing the error estimates

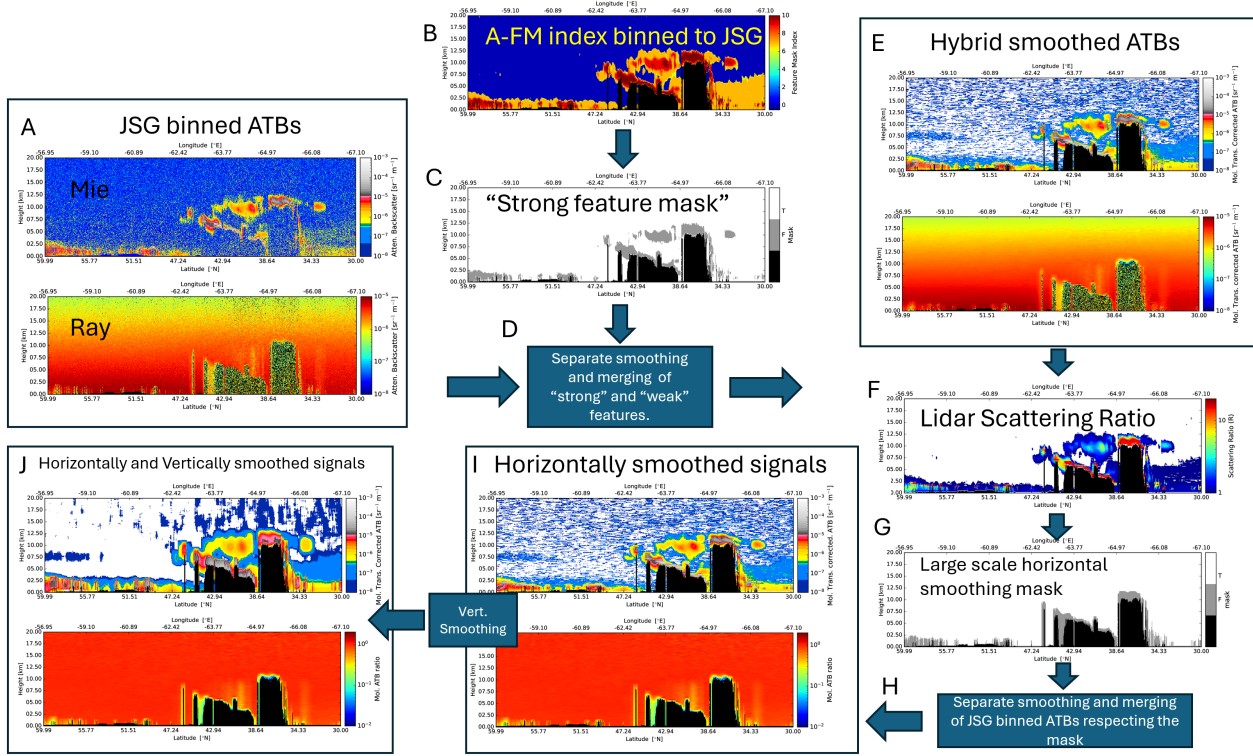

**Figure 3.** Conceptual depiction of the A-AER pre-processing processing illustrated using simulated data corresponding to the Halifax scene Donovan et al. (2023). The processing steps depicted here correspond to steps 1 to 7 in Figure 4.

coming from the ATLID L1 product. A-FM data which is at native ATLID resolution is also put onto the JSG. In the case of the A-FM data the A-FM feature probability indices, are not averaged. Rather, the highest index within the appropriate JSG pixel is used, except in the case where the input indices contain a surface detection indication, in this case, the JSG pixel is flagged as a surface pixel. As well, in this step, the temperature, pressure and atmospheric number density for each JSG pixel are determined using the X-MET auxiliary product.

**Step 2** is to create a mask to guide the horizontal averaging of the input ATBs. A strong-feature mask is created by thresholding the JSG-gridded A-FM product. This mask is then used in step 3 to smooth the data respecting the mask.

In **step 3**, the JSG-grided ATBs are averaged using a moving average box-car, but pixels identified as either strong-features or weak-features are not segregated and are averaged separately. The smoothed strong-feature and weak-feature pixels are then merged together. The strong-feature A-FM threshold and the averaging window are both configurable. The prelaunch defaults are set to $\geq 8$ (corresponding to likely-cloud) for the A-FM index threshold and 1 vertical by 40 horizontal JSG pixels for the averaging window. The setting of these configuration parameters, along with any others, will be re-evaluated during commissioning phase (and indeed throughout the mission life-time) using in-orbit observations.

In **step 4** the lidar scattering ratio is calculated along with corresponding error estimates using the ATBs resulting from the previous step. These scattering ratio estimates are then used in the next step to create a mask to help guide the horizontal averaging that will be ultimately applied to the ATBs.

In **step 5**, the altitude dependent limits that will be applied on a column-by-column basis in order to average the ATBs in preparation for the quantitative determination of extinction and backscatter are found. The first sub-step is to define a mask representing pixels where it is considered valid to average over horizontally. This mask is first built by masking out:

- Fully attenuated pixels.

- Surface pixels.

- Pixels with an associated scattering ratio at 355 nm (as calculated in Step 4) above a specified threshold, e.g. 2 corresponding to the scattering ratio at the surface. This threshold is reduced with height according to the atmospheric density profile, i.e. $R_{th}(z) = 1.0 + (R_{th}(z_s) - 1)\rho(z)/\rho(z_s)$, where $z_s$ is the surface altitude and $R_{th}$ is the threshold value. Setting $R_{th} = 2$ corresponds to a particulate backscatter threshold of about $1.6 \times 10^{-5}$ [ $\text{m}^{-1}\text{sr}^{-1}$ ].

Averaging attenuated regions (e.g. below optical thick clouds) together with unattenuated regions would ultimately lead to inaccurate extinction and backscatter profiles. In order to avoid this, the averaging mask is set to false for all altitudes in each respective column that are below the highest altitude of the mask in the given column. In addition, single pixel isolated true mask values (i.e. a true value completely surrounded by false values) are set to false.

The height-and-time dependent horizontal averaging limits are then found. This is accomplished by first considering a box centered around the JSG column in question. Then using all the data identified by the averaging mask the height averaged SNR of the resulting horizontally averaged Rayleigh ATB is calculated. If this average SNR is below a set threshold (e.g. 50) then the horizontal extent of the box is expanded until the threshold is met or a specific maximum box extent is reached. The average weak-feature ATBs for all three channels and their respective error estimates are then calculated (**step 6**) for each height bin.

In **step 7**, the weak-feature Mie ATBs are corrected for the effects of Rayleigh transmission and are smoothed vertically using a sliding linear least-square fitting procedure. The same is done for the ratio of of the Rayleigh ATB to molecular backscatter ratio. The fitting window is configurable (e.g. 5 vertical range-bins). The use of a linear fit provides a natural way to handle the edge-effects at the top altitudes and the near ground-pixels i.e. linear extrapolation is used to find appropriate values of the pixels closer than the fitting window half-width from the top of the profile or the ground-pixel. The use of a sliding linear fit enables the range derivative of the signals to be calculated in the same step in a consistent manner. This procedure produces the following *horizontally and vertically smoothed* quantities:

$$B_{M,\|,hv} = F_{hv}\left(b_{M,\|}\exp[2\tau_{Ray}]\right), \tag{6}$$

$$B_{M,\perp,hv} = F_{hv}\left(b_{M,\perp}\exp[2\tau_{Ray}]\right), \tag{7}$$

$$B_{M,hv} = B_{M,\perp,hv} + B_{M,\parallel,hv} = F_{hv} \left( b_M \exp[2\tau_{Ray}] \right), \tag{8}$$

where $b_M = b_{M,\parallel} + b_{M,\perp}$ and

$$B_{R,hv}^{Rat} = F_{hv} \left( \frac{b_R}{\beta_R} \exp[2\tau_{Ray}] \right). \tag{9}$$

where the $Rat$ superscript is used to indicate that the ratio between the observed molecular extinction corrected Rayleigh attenuated backscatter and the unattenuated Rayleigh backscatter is being used. In Eqs. 6–9, $F_{hv}$ is used to denote the masked vertical and horizontal smoothing operation, and $\tau_{Ray}$ is the Rayleigh optical depth from the Top-of-Atmosphere to the height in question. Here, for simplicity the range dependence is not written explicitly.

**Step 8** involves the estimation of the particulate lidar-ratio $S$, the extinction and the backscatter. The output of the previous step (i.e. Eqs. 6 − 8) are used. Depending on the configuration, this can be accomplished in two ways. Which method is applied can be selected as a configuration option. One of the methods is build using a conventional log-derivation approach for estimating the extinction profile, the other uses a new "local-forward-modelling approach". The two different approaches are described and discussed in Appendix A. The new approach generally produces better behaved extinction-to-backscatter ratio profiles. This is important since, in subsequent steps, the A-EBD algorithm will separate and classify layers using, in part, the extinction-to-backscatter ratio. $S$ is also a primary input to the A-EBD algorithm. No multiple-scattering correction is performed at this stage, since that can only be done once an target classification has been performed.

In **step 9** the coarse layer structure calculated. The JSG gridded A-FM indices are used along with the scattering ratio calculations performed in the previous step. This process is described in detail in Irbah et al. (2023). In brief, layer boundaries are assigned whenever significant changes in the JSG A-FM indices or, optionally, significant differences in temperature and/or scattering ratio are encountered. Also, layers whose vertical extent is above a specified threshold (e.g. 2 km) are split into two.

In **step 10** each coarse layer is examined to see if the layer should be further subdivided. The basic idea is to test if it is valid to represent the layer as a homogeneous entity or if it is better to split the layer into a number of homogeneous sub-layers. The procedure relies on examining the behaviour of a reduced chi-squared goodness-of-fit variable applied to the scattering–ratio, lidar–ratio and the depolarisation–ratio which is calculated for all possible sub-layering for up to four distinct sub-layers. This process can be somewhat computationally demanding for extended regions so that it is notably more efficient to employ a coarse layer and splitting process to find the fine-layer structure rather than trying to directly determine the fine layering structure. The layer splitting algorithm is described in more detail in Section 2.1.2 of Irbah et al. (2023).

An example, using a subsection of the Halifax scene, of the determined layer boundaries is shown in Figure 5. The coarse layer boundaries (shown in black) were determined using the FM index and scattering ratios. Within the cloud-free aerosol columns there is little contrast in the scattering ratio and the FM indices respectively. The aerosol layer is divided into two coarse layers in any case, since its extent would otherwise exceed the maximum allowed extent (here set to 4 km). It can also be seen that the fine-layer determination procedure results in the coarse layers being further divided here usually into 3 or 4 sub-layers.

Based on the fine layer structure calculated in step 10, the mean backscatter, extinction, lidar-ratio and depolarization ratio are calculated. This information is then passed to the classification procedures (**steps** 13 and 14) which are described in Sections 2.2 and 2.3 of (Irbah et al., 2023). The simple classification procedure (**step 13**) can be viewed as a back-up that can be used when the detailed classification procedure (**step 14**) can not be applied reliably and is used in the A-EBD component of A-PRO. It should be noted that extinction and backscatter information used at this stage have not been corrected for multiple-scattering effects, thus, the classification procedures are called using a version of the classification priors adjusted to approximately account for MS effects (e.g. reduced effective lidar-ratio in ice clouds).

In **step 15** the extinction and lidar-ratio values calculated in step 9 are corrected for multiple-scattering effects. The treatment of multiple-scattering is treated in general within Section B and the specific adjustment of the values for extinction and backscatter within A-AER are treated within Appendix B4. After the MS correction, to maintain consistency, the layers are then re-classified using a call to the A-TC procedures but this time using the classification priors appropriate for single-scattering are used.

After steps 1 – 15 are completed for the whole frame, the data product file is written out including the layering and classification information.

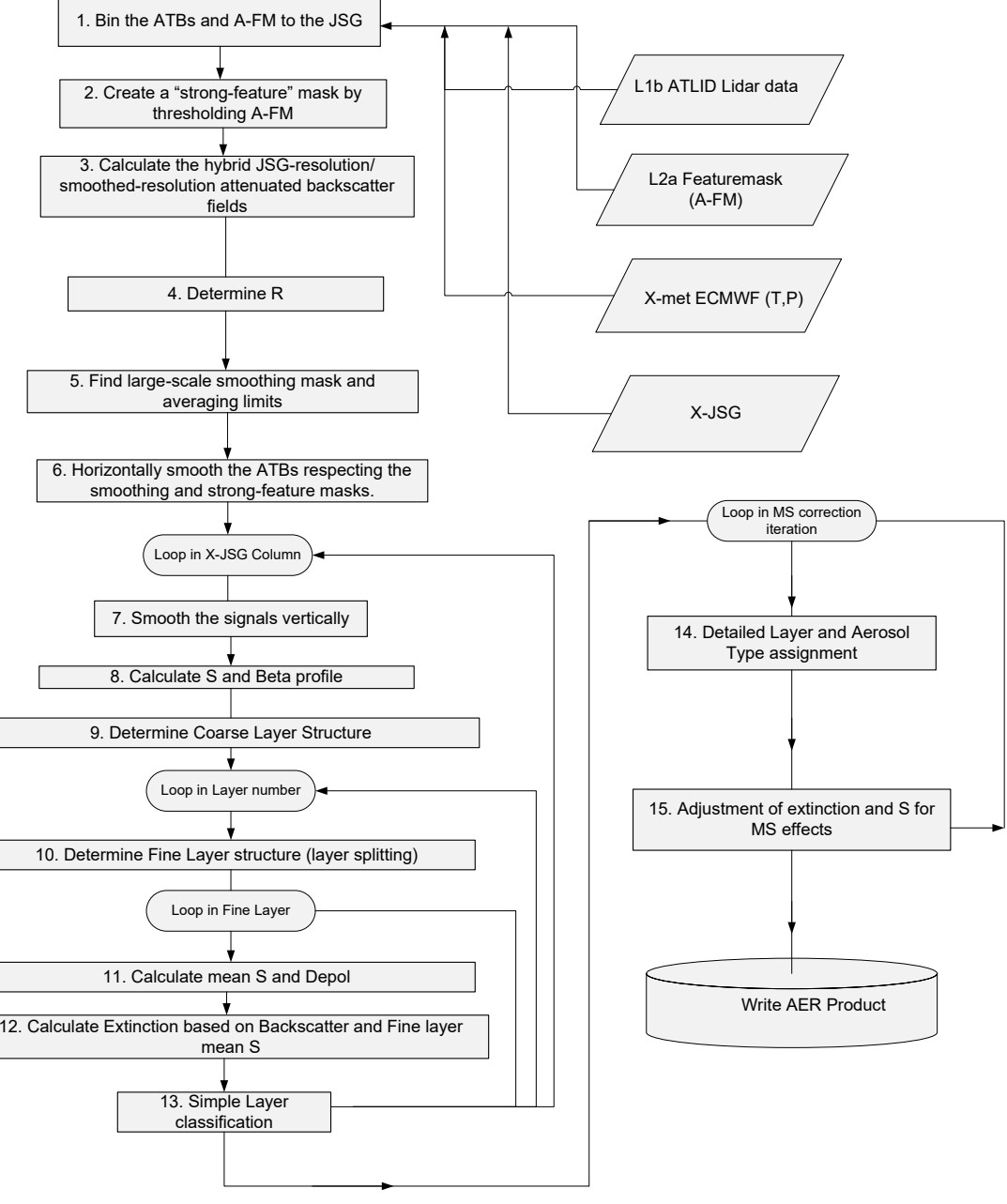

**Figure 4.** Schematic depiction of the structure of the A-AER components of the A-PRO processor.

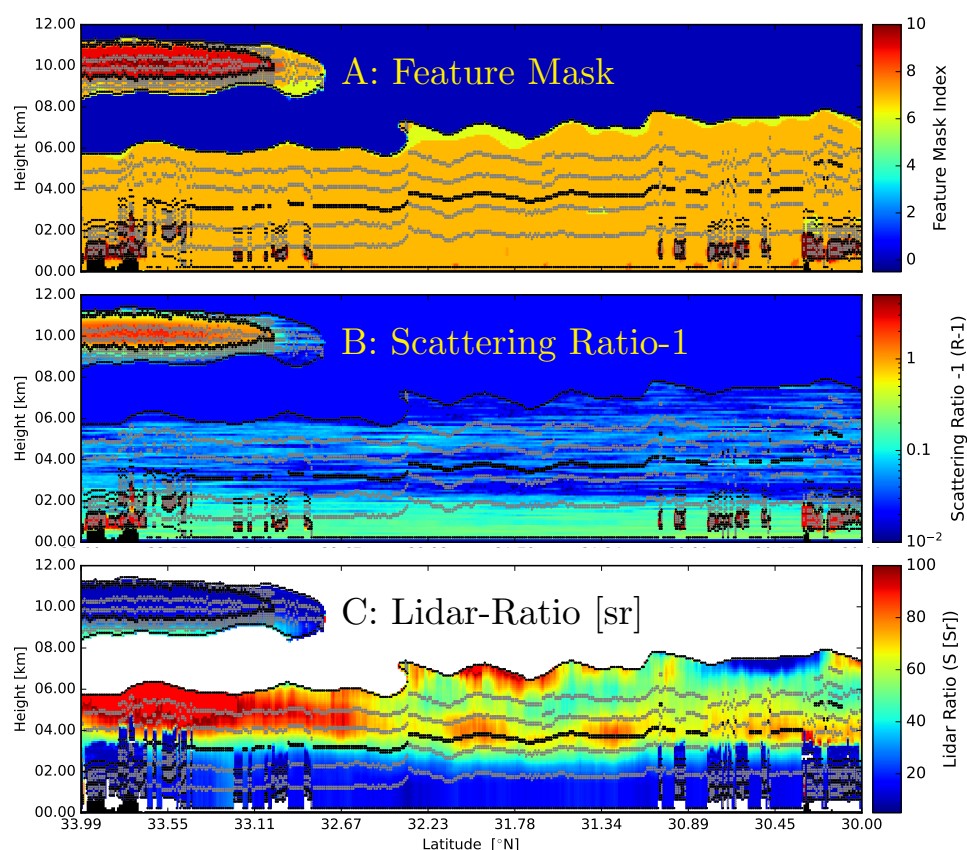

**Figure 5.** Layer tops and bottoms over-plotted on the A-FM index, the scattering ratio -1, (as calculated in step 4) and the lidar-ratio (as calculated in step 8). The black lines represent the coarse layer boundaries, while the Grey lines represent the fine layer boundaries.

## 2.3 A-EBD

After A-AER has been executed, the ATLID Extinction-backscatter and depolarization procedure is applied. The core of this procedure is built upon a column-by-column optimal estimation (OE) forward model inversion performed at a higher resolution than A-AER with the aim of supplying extinction and backscatter values for both 'weak' and 'strong' targets. A-EBD uses the classification and (where possible) the lidar-ratio estimates generated by A-AER as a starting point. Like all optimal-estimation approaches a *cost-function* ($\mathbf{J}$) is formulated which expresses the sum of the weighted difference between the observations and the observations predicted by a forward model ($\mathbf{F}$) given a certain state ($\mathbf{x}$) and the weighted difference between the state and and an a priori state ($\mathbf{x_a}$).

The A-EBD approach can be compared to the Cloud, Aerosol, and Precipitation from mulTiple Instruments using a VAriational TEchnique (CAPTIVATE) approach (Mason et al., 2023b) in the particular case when only ATLID observations are used. They are both forward modelling optimal estimation approaches than take multiple-scattering into account, however, they differ in several ways e.g.:

- The multiple scattering treatment is different.

- A-EBD uses a layer-based approach to represent the clouds and aerosol properties, while CAPTIVATE uses splines to represent a continuous vertical structure of the clouds and aerosol properties.

- The cost-functions used are different. In particular, CAPTIVATE uses a fully logarithmic approach to express difference between the observations and the forward model, while A-EBD uses a linear approach.

The particular cost function used by A-EBD can be written as

$$J = [\mathbf{y} - \mathbf{F}(\mathbf{x})]^T \mathbf{S_e^{-1}} [\mathbf{y} - \mathbf{F}(\mathbf{x})] + [\mathbf{x_r} - \mathbf{x_a}]^T \mathbf{S_a^{-1}} [\mathbf{x_r} - \mathbf{x_a}], \tag{10}$$

where

- $\mathbf{y}$ is the observation vector including the observed Rayleigh and Mie attenuated backscatters

$$\mathbf{y} = \left( B_{R,1}^c, B_{R,2}^c, .... B_{R,n_z}^c, B_{M,1}^c, B_{M,2}^c, ... B_{M,n_z}^c \right)^T \tag{11}$$

where $n_z$ is the number of range-gates, $B_{R,i}^c$ is the observed Rayleigh attenuated backscatter corrected for the effects of molecular Rayleigh attenuation

$$B_{R,i}^c = b_R(z_i) \exp \left[ 2 \int_{z_{lid}}^{z_i} \alpha_R(z') dr(z') \right] \tag{12}$$

and $B_{M,i}^c$ is the observed total Mie attenuated backscatter corrected for the effects of molecular Rayleigh attenuation i.e.

$$B_{M,i}^c = \left( b_{M,\parallel}(z_i) + b_{M,\perp}(z_i) \right) \exp \left[ 2 \int_{z_{lid}}^{z_i} \alpha_R(z') dr(z') \right] \tag{13}$$

– $\mathbf{x}$ is the state-vector defined as:

$$\mathbf{x} = \log_{10}\left(\alpha_{M,1}, \alpha_{M,2}, ...\alpha_{M,N}, S_1, S_2, ..S_{nl}, Ra_1, Ra_2, ...Ra_{n_l}, C_{lid}\right)^T \tag{14}$$

where $S$ are the lidar-ratios, $R_a$ are the effective area radii, and $C_{lid}$ is a factor used to account for calibration errors. Here $N$ is the number of range-gates identified within A-AER as being **non-clear-sky** and $n_l$ is the number of layers for the along-track column being treated. This formulation (where the particulate extinction is set to zero for clear-sky range gates and the lidar-ratio and particle sizes are constant within layers) is used to reduce the dimensionality of the problem which can have a significant positive impact on the computational requirements.

The log form constrains the retrieved state-vector to be positive and is consistent with the the state variable statistics being more accurately thought of as being log-normal rather than normal (Kliewer et al., 2016). For example, the lidar-ratio is well described by a log-normal form (see e.g. Fig. 6 of Wang et al. (2016)). In our experience, the use of the log-form for the state-vector does not present any particular challenges for the minimization. In fact, it has been shown to have benefits in the quality of the retrievals (see e.g. Maahn et al. (2020))

– $\mathbf{x_a}$ is the logarithmic a-priori state vector. Here defined as a vector consisting of the log base 10 values of the a priori lidar-ratios, effective area particle sizes and the value of $C_{lid}$ appropriate for calibrated attenuated backscatter signals (i.e. 1). Using a log form here is consistent with the a priori errors being proportional in nature rather than absolute.

$$\mathbf{x_a} = \log_{10}\left(S_{a,1}, S_{a,2}, ..S_{a,n_l}, Ra_{a,1}, Ra_{a,2}, ...Ra_{a,n_l}, 1\right)^T \tag{15}$$

Note that here no a-priori constraints are placed upon the log extinction values so that they are not present in the a-priori state-vector. The a priori values of the lidar-ratio and and their associated error estimates are taken from the A-AER results, when quantitative retrievals are flagged as valid. Otherwise, The a priori values of the lidar-ratio and errors depend on per-type tabulated values associated with the A-AER target classification. For the ice particle effective radii, the a priori values (and errors) are provided by the A-ICE procedure which is described in Section 2.3.2 or fixed values can be used (as specified in a configuration file). For water cloud and aerosols the effective radii are specified a priori by type.

– $\mathbf{x_r}$ is the reduced state-vector, which is a subset of $\mathbf{x}$ consisting of the the non-extinction associated elements, consistent with the definition of $\mathbf{x_a}$.

– $\mathbf{S_a}$ is the a priori error covariance matrix. It is assumed that the a priori errors are uncorrelated so that the matrix takes a diagonal form. Here the form of the entries is the one appropriate for a logarithmic state-vector Kliewer et al. (2015) i.e

$$S_{a_{i,i}} = \log_{10}\left(1 + \left(\frac{\sigma_{x_{a,i}}}{x_{a,i}}\right)^2\right) \tag{16}$$

where $\sigma_{\mathbf{x_{a_i}}}$ is the a priori (linear)uncertainty assigned to the $i$th component of $\mathbf{x_a}$.

The assumption that the a priori errors are uncorrelated is only an expedient. In reality, correlations between the different elements of $\mathbf{x_a}$ exist (e.g. the lidar-ratio of different vertical levels in cirrus clouds). However, populating the off-diagonal elements of $\mathbf{S_a}$ in a general sense would a non-trivial exercise. However, when enough real observations have been acquired, it may be possible to iteratively assess the main correlations and fill in the correlations via a "bootstrap" process.

– $\mathbf{S_y}$ is the observational error covariance matrix. For practical reasons, in the present version of the algorithm, it is assumed that the observational errors are uncorrelated so that the matrix takes a diagonal form so that.

$$S_{y_{i,i}} = \sigma_{y_i}^2 \tag{17}$$

where $\sigma_{\mathbf{x_{a_i}}}$ is the (linear)uncertainty assigned to the $i$th component of $\mathbf{Y}$. In fact, the errors for the Mie and Rayleigh signals at the same altitudes will be correlated due to spectral crosstalk, this issue is planned to be addressed in future investigations.

– $\mathbf{F}$ is the forward model which predicts the Rayleigh and Mie attenuated backscatter profiles given the state-vector as an input. The forward-model accounts for multiple-scattering. The multiple-scattering lidar equation used in this work is described in detail in Appendix B and the exact discrete form used in this work along with its Jacobian is described in Appendix C.

### 2.3.1 A-EBD procedure

The optimal-estimation retrieval is embedded within a broader framework. A high-level flow diagram of the A-EBD component of the A-PRO processor is shown in Fig. 6. **Step 1** is similar to the first step of the A-AER procedure. That is, the L1 ATLID signal are re-binned to the JSG resolution and the auxiliary met data is read and processed etc. In addition, however, the A-AER results (layering structure, classification, retrieved extinction and lidar-ratios and error estimates etc.) are also read in.

**Step 2** involves the set-up of the OE inversion. In particular:

– For the lidar-ratio elements of $x_a$ and their associated error estimates, A-AER supplied values are used for layers when available.

– For layers where A-AER could not derive a valid quantitative lidar-ratio, a priori values based on the A-AER classification (e.g. ice-cloud, water cloud, aerosol(type)) are used.

– The per-layer effective area sizes and associated a priori uncertainties are based on the A-AER classification. For ice-clouds a temperature dependent parameterization can be used (see Section 2.3.2).

– The starting values for the extinction elements of the state-vector are taken from A-AER when valid. Otherwise they are based on either the layer-averaged scattering ratio and the a priori lidar-ratio when valid, or fixed values depending on the classification.

Once the OE problem has been set-up, within **step 3** the cost-function is minimized using a version of the well-known Broyden–Fletcher–Goldfarb–Shanno (BFGS) Quasi-Newton numerical minimization procedure (Press et al., 2007). The errors in the retrieved state-vector (**step 4**) are computed following the procedure outlined in Section 15.5 of Press et al. (2007).

Once all the columns have been processed, the A-TC procedure is called to assign a classification to each layer (**step 5**). The A-TC classification procedure is described in Sections 2.2 and 2.3 of (Irbah et al., 2023). In **step 6**, the medium and low-resolution fields are formed. Here, the extinction, backscatter and depolarization values are horizontally smoothed to medium and low resolutions. The smoothing is guided by the weak-feature mask (the complement of the strong-feature mask, see steps 2 and 7 of the A-AER procedure described in Section 2.2) modified to exclude E-EBD extinction values above an adjustable

threshold, and excluding any water-cloud pixels. Medium-resolution is set to default to 40 km and low-resolution is set to a default of 150 km. These setting are adjustable and will be re-visited when ATLID in flight data is available. For pixels not covered by the weak-feature mask, the high resolution single JSG column values are used (i.e. the high resolution results are merged with the low and medium fields respectively). Using the merged low and medium the A-TC classification routines are then called to generate the low and medium resolution classification fields (step 7). Lastly, the A-ICE product variables are

calculated (see Section 2.3.2.).

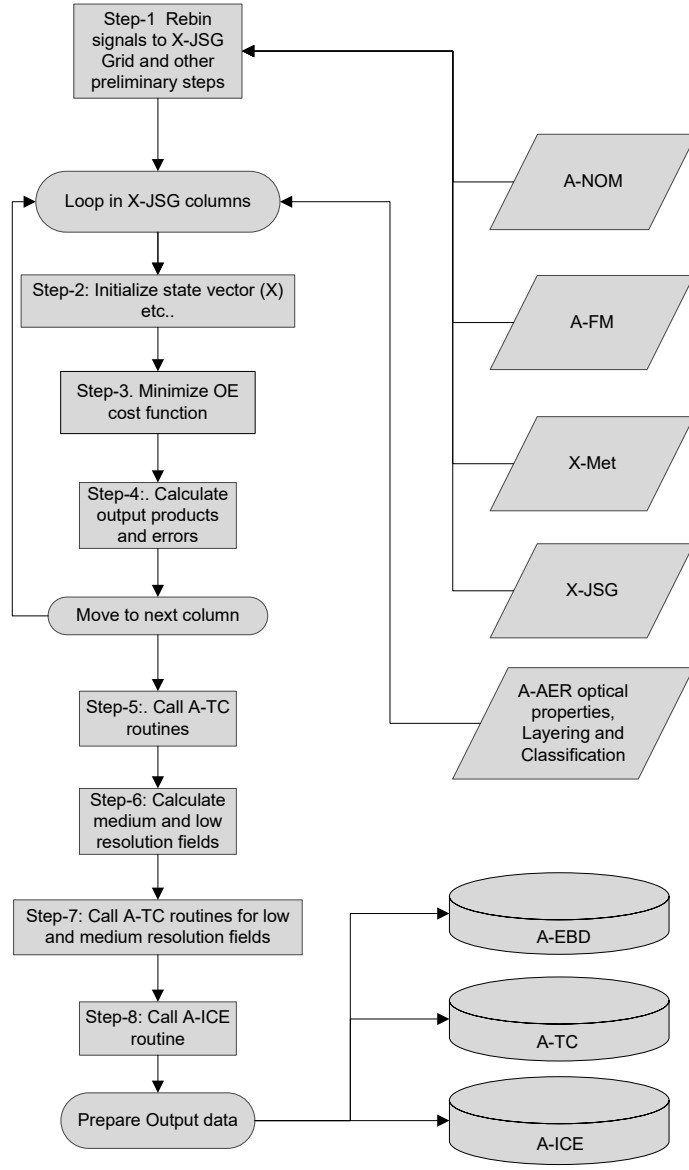

**Figure 6.** Schematic depiction of the structure of the A-EBD component of the A-PRO processor.

### 2.3.2 A-ICE

A-ICE is used to supply estimates of the ice-water-content (IWC) and effective particle radius using parametizations supplied with retrieved extinction values and the atmospheric temperature. One of two options can be specified. The two approaches used are based on (Heymsfield et al., 2005) and (Heymsfield et al., 2014b) respectively. The exact coefficients used in each approach are configurable and may be adjusted based on the experience with actual EarthCARE data. IWC and effective radii estimates are produced for all pixels identified as being ice in the A-TC product (Irbah et al., 2023).

## 3 Case Studies using Simulated Data

In this section, the application of A-PRO to the three main EarthCARE test scenes (Qu et al., 2023; Donovan et al., 2023) will be presented and discussed. Here the focus is on the retrieved optical properties (extinction and lidar-ratio), however, target classification results (A-TC) are also presented. For each of the three scenes (Halifax, Baja, and Hawaii) overall results are presented, and a few extracts are presented in detail. More information is presented in Mason et al. (2023a), where the results of various EarthCARE L2 retrieval algorithms are compared with each other and the model truth. It should be noted, however, that the data shown in this paper (Version 11) is not the same as the version used in (Mason et al., 2023a) (Version 10). In general, the results shown here (being based on a more developed version of A-PRO) are superior but not dramatically so.

One of the aims of this section is to give the reader a feeling for the nature of the product that will be supplied to the community including the limitations and possible caveats. To this end, the sample retrievals have not been overtuned to produce optimal results. For example, it would be possible to tune the priors to the model scenes (e.g. lidar-ratios, $F_{MSp}$, Ra, $\eta$) to closely match the three scenes. This is not done here, since it is instructive to gain insight into the relative robustness of the various retrieved variables in different circumstances given reasonable limited knowledge of the priors with respect to actual observations. For example, most of the ice clouds present in the test scenes have an associated lidar-ratio of 30 sr, however, in these retrieval, an a priori value of 25 sr with a relative uncertainty of 20% is used. The values of $F_{MSp}$ are also fixed and were not tuned, these were set based on idealized off-line simulations (see e.g. section B) which were conducted independently of the main tests scene scene construction process. As a result, bias differences from the model-truth are to be expected. These differences will help guide investigations that will be conducted post-launch.

### 3.1 Halifax

The attenuated co-polar Mie, Rayleigh, and cross-polar backscatter signals as well as the corresponding A-FM Feature-mask (van Zadelhoff et al., 2023) for the Halifax scene are shown in Fig. 7. These fields, along with the X-JSG and X-MET products (Eisinger et al., 2023) form the inputs for A-PRO. The particle extinction products produced by A-PRO corresponding to the inputs shown in Fig. 7 are shown in Fig. 8. The black regions are regions flagged as being not valid (e.g. attenuated or otherwise invalid), the black vertical strip is the result of a gap in the simulated lidar signals after re-binning to the JSG. In Fig. 8, the differences between the different products can be seen. Compared to the A-EBD fields, the A-AER estimate is the smoothest,

however, no extinction estimates flagged as being valid are generated for the strong e.g. clouds. For the A-EBD estimates it can be seen that the aerosol and thin ice cloud areas become smoother as the horizontal resolution decreases, however, the cloud extinction regions are not smoothed (see the last step described in 2.3.1).

Two example regions of the Halifax scene extinctions results are presented in more detail within Fig. 9. Here the retrieved extinction is compared to the model truth for a representative high altitude ice cloud region and an aerosol region. In the profile plots of the retrieved extinction ($\alpha_r$) shown in the left part of panel B and C, the light-blue regions correspond to the average estimated uncertainty in a relative sense i.e.

$$\frac{\overline{\alpha_r}}{1+\overline{\sigma}_{\alpha_r}} < \alpha_r < \overline{\alpha_r} + \overline{\sigma_{\alpha_r}}, \tag{18}$$

which is consistent with the logarithmic nature of the state-vector used in the retrievals. The black lines represent the estimated error of the average profiles i.e.

$$\sigma_{\overline{\alpha_r}} = \frac{\sum_{i=1}^{N} \sigma_{i,\alpha_r}^2}{N^{1/2}}, \tag{19}$$

where $N$ is the number of samples contibuting to the average.

    The sample ice-cloud region (panel-B) shows that in in this case, on the 10-km scale that the extinction values above 435   $10^{-2}$ km$^{-1}$ are accurately retrieved ( e.g. within a factor of 1.5), albeit with a low estimated precision. In the more attenuated areas of the cloud at lower altitudes (e.g. below 8.75 km), the accuracy and precision degrade due to worse SNR and the imperfect correction of MS effects. The 2nd region (panel-C) corresponds to a cloud-free aerosol case. Here it can bee seen that on the 50 km$^{-1}$ scale, for extinction values above $10^{-2}$ km$^{-1}$ are retrieved with an accuracy of about 50 %.

    The lidar-ratio retrievals corresponding to Figs. 8 and 9 are shown in Figs. 10 and 11 respectively. Here the fact that $S$ is only 440   retrieved per-layer is evident (see esp. The lower-right panel of Fig. 11. In Fig. 10 it can be seen that the A-AER estimate, for the cloud aerosol are generally too low (15-20 sr vs 25 for the model-truth) this is roughly consistent with may be expected due to the limitations of correcting for multiple-scattering effects (see the discussion in section B4). In the EBD optimal estimation retrieval, the particle size is an element of the state-vector, and a fuller treatment of multiple-scattering is possible, thus EBD generally retrieves lidar-ratios that are closer to the model truth. Referring to Fig. 11, it can be seen that the retrieved lidar-445   ratios are within about 10-15% for the cirrus case. For the aerosol section presented in Fig. 11, (where multiple-scattering is less-important) the retrieved lidar-ratios are generally within 10% and there is not such a greater difference between the AER and EBD retrievals.

    Within Figs. 10 and 11 it may be observed that the highest estimated values of the lidar-ratio occur in the aerosol layer at around 32.5 $^o N$. This is even clearer in Fig. 12 where it can be seen that the anomalously high estimated lidar-ratio leads to 450   a missclassification of the aerosol type. This anomalous region coincides with the presence of a semi-transparent ice cloud present between about 8 and 10 km and is examined in more detail in Fig. 13. Referring to Appendix B, this is an example of the decaying multiple-scattering tail beneath clouds influencing the signals below. The (g)–(k) sequence of panels show that the Mie and Rayleigh attenuated backscatters are indeed well-fitted and the extinction is generally accurately retrieved (albeit with large relative estimated uncertainty), however, the retrieved lidar-ratios exhibit large differences from the model-truth.

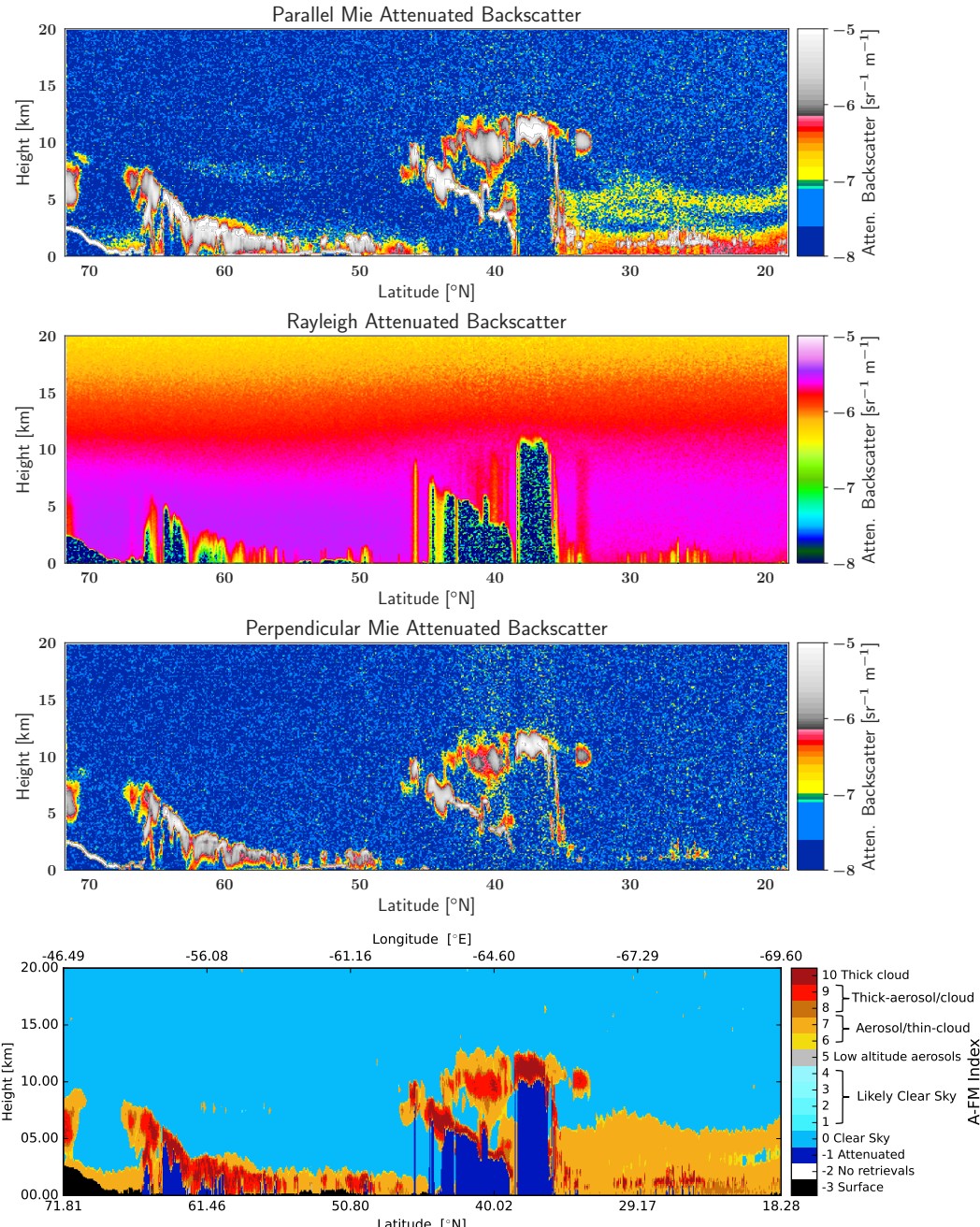

**Figure 7.** Simulated Mie,Rayleigh, and Cross-polar attenuated backscatter for the Halifax scene along with the A-FM L2 Featuremask.

These errors are largely the result of the difficulty in fitting the decaying tail and the use of fixed-values for the $f_{MSp}$ factors. This observation, points to an avenue of inquiry once ATLID observations are available. In particular, if similar features in actual ATLID retrievals are found below semi-transparent cloud layers, then it may be necessary to refine the setting to $f_{MSp}$ by e.g. including it the state-vector or parameterizing it as a function of multiple-scattering ratio and particle-size/type. The (a)-(f) sequence of panels (the "Platt's approach" panels) are used to illustrate a related point. Namely, that not allowing for

the occurrence of tails in the forward model leads not only to somewhat higher retrieved lidar-ratios but much higher retrieved extinction values below the ice cloud.

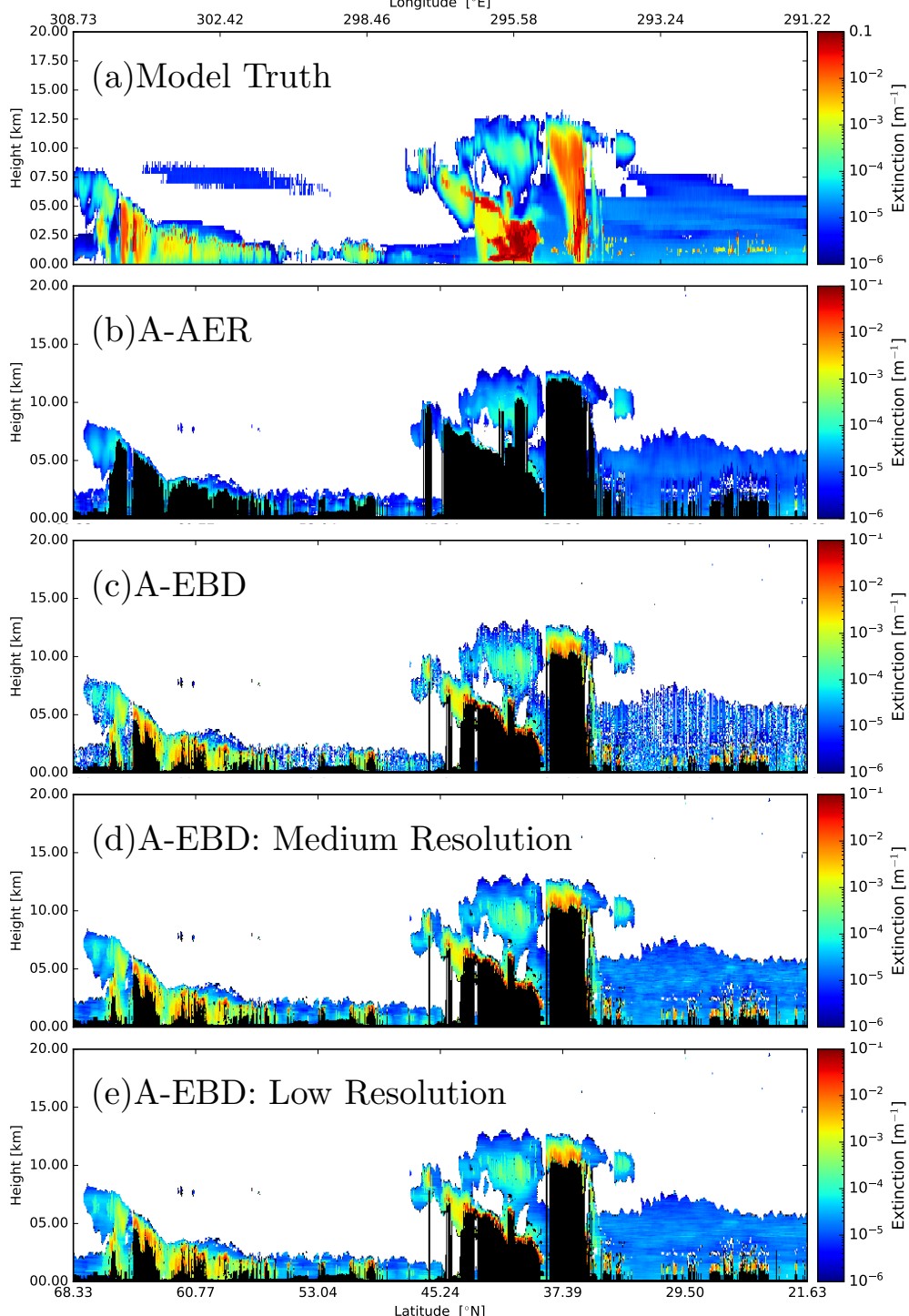

**Figure 8.** Model-truth extinction for the Halifax scene and the corresponding A-AER and A-EBD products. Here "Medium" resolution is 50 km while "Low" resolution corresponds to 100 km.

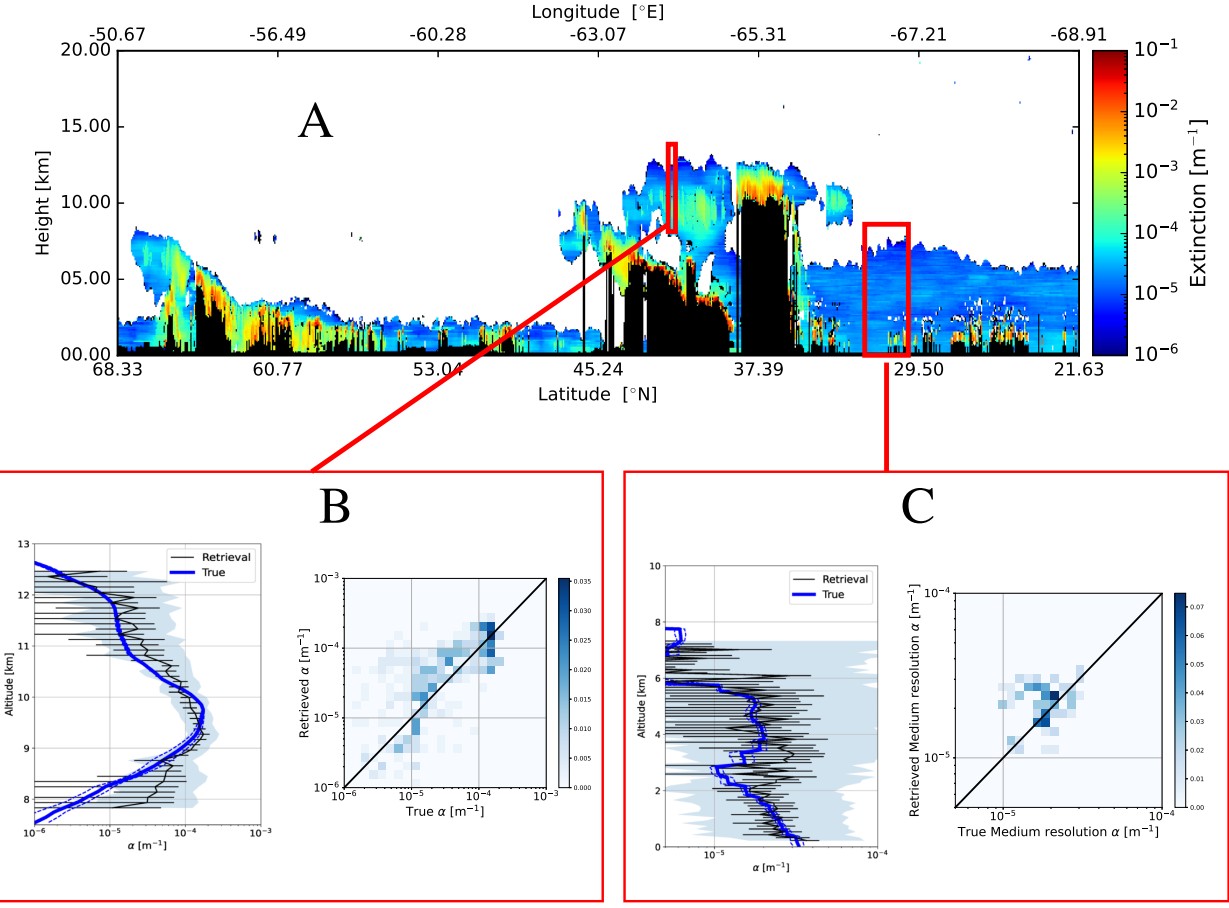

**Figure 9.** Low-resolution retrieved extinction for the Halifax scene (Top) and two representative average profiles. Panel-B corresponds to a 10 km horizontal average of the A-EBD high-resolution extinction profiles and the corresponding model-truth extinction profiles. The panel-C corresponds to a 50 km horizontal average. The Solid-Blue lines represent the average model-truth and the dashed-Blue lines delineate the +/- the model-truth standard deviation region. The solid black line represents the average retrieved extinction, the light-Blue shaded region the average relative uncertainty, and the error-bars represent the uncertainty of the mean retrieved profile The color-scale assigned to the True-vs-Retrieved histogram plots correspond to the normalized number of counts in each 2D histogram. The histograms were constructed using the medium resolution outputs in the respective windows and not the horizontally averaged data (which is displayed in the associated profile plots.)

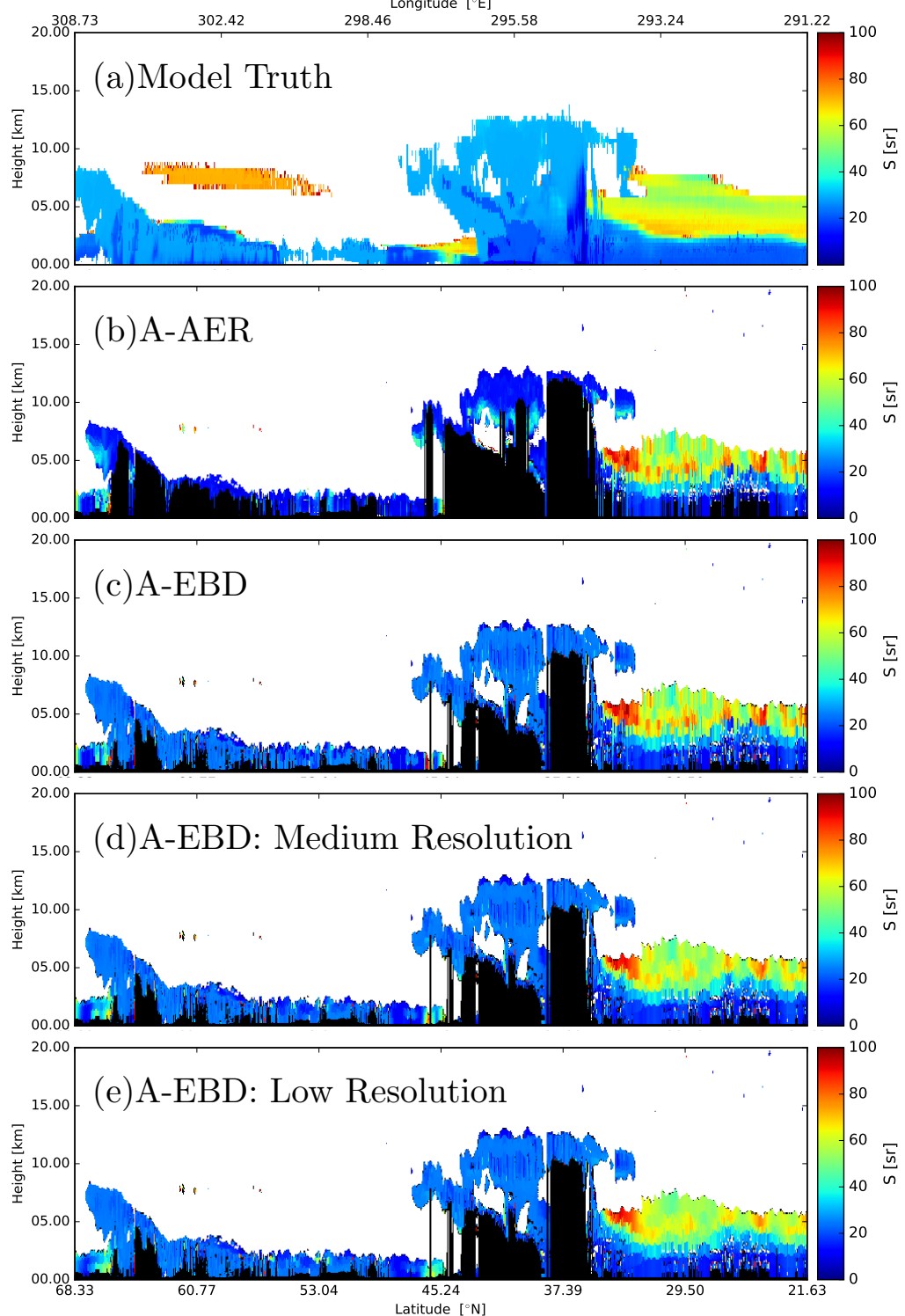

**Figure 10.** Model-truth lidar-ratio for the Halifax scene and the corresponding A-AER and A-EBD products. Here "Medium" resolution is 50 km while "Low" resolution corresponds to 100 km.

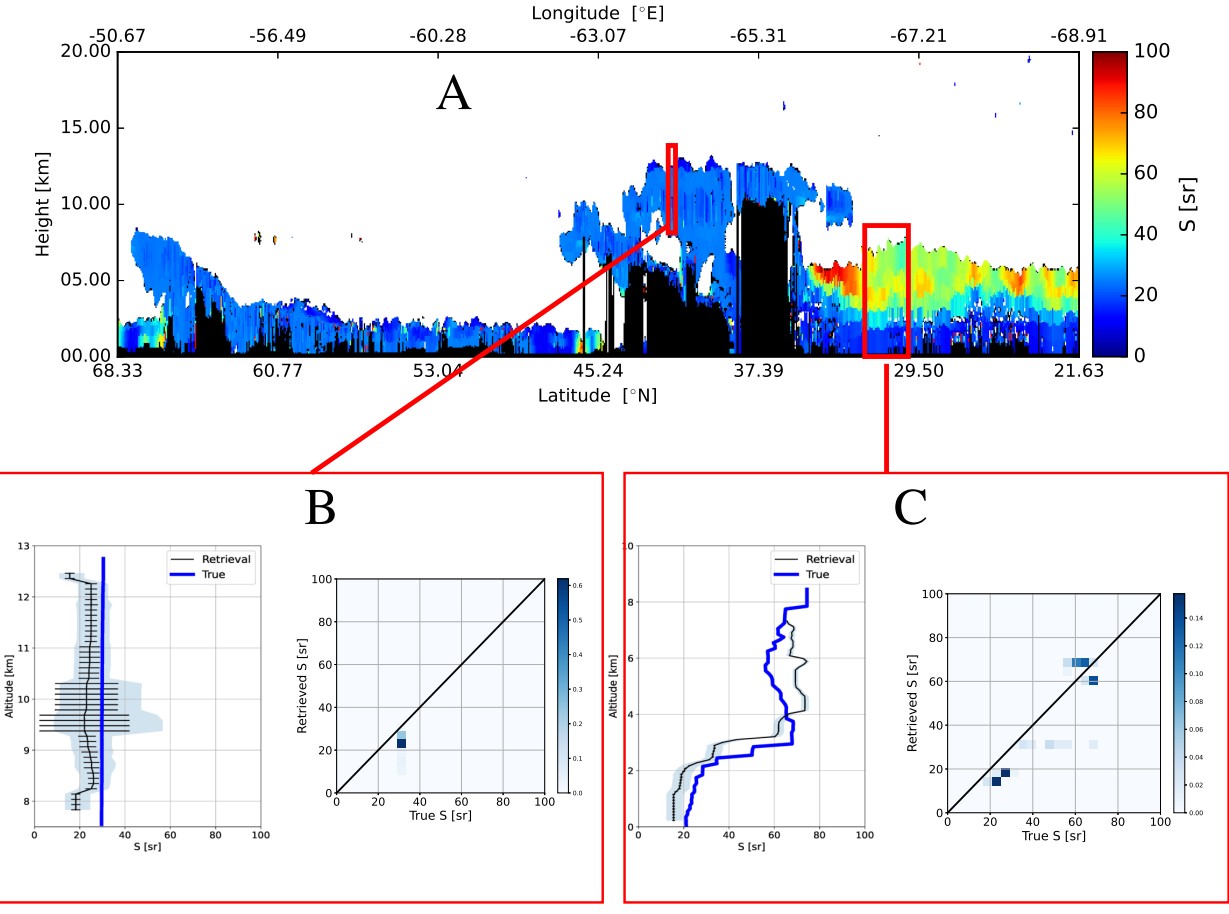

**Figure 11.** As Fig. 9 except for the lidar-ratio.

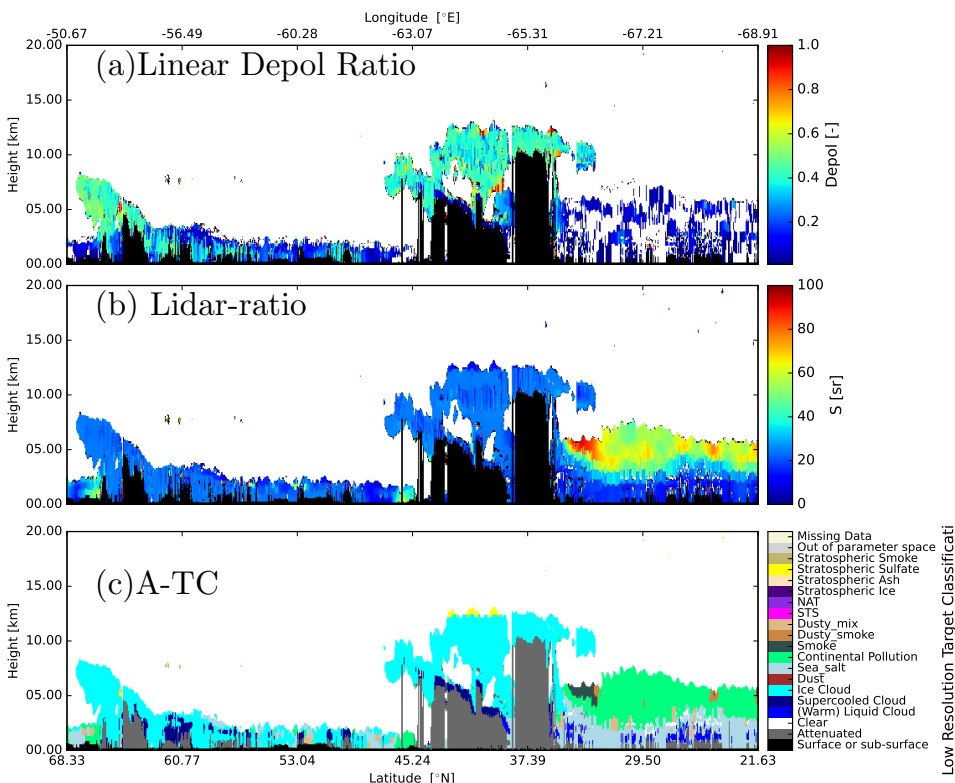

**Figure 12.** (a) Low-resolution particulate depolarization ratio. (b): Retrieved low-resolution lidar-ratio and (c) the corresponding AC-TC classification field.

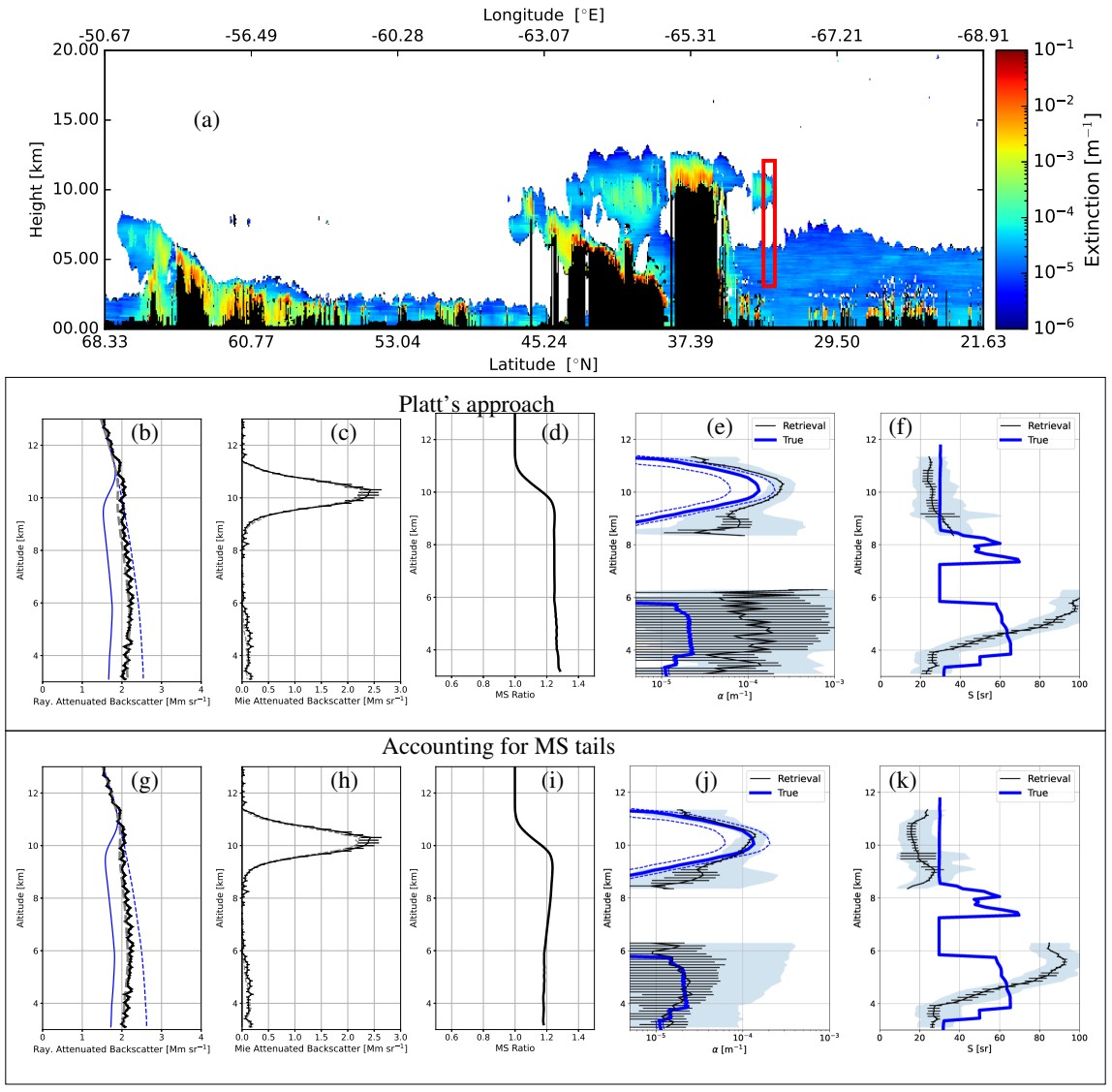

**Figure 13.** (a): Retrieved low-resolution extinction. (b): Average Rayleigh attenuated backscatter for the indicated (10km horizontal) interval (black line), the forward-model fit (grey-dashed-line), the corresponding single-scatter Rayleigh attenuated backscatter (solid-Blue-line) and the Rayleigh-clear attenuated backscatter profile. (c): Average Mie attenuated backscatter for the indicated interval (black line) and corresponding forward-model fit (grey-dashed-line). (d): Ratio of (total) multiple-scattering plus single scattering to single scatting return for the Rayleigh backscatter (e) Average retrieved extinction profile and associated standard deviation profile (black lines) and average model-truth extinction profiles (solid blue-line) (f) Retrieved lidar-ratio corresponding to (e). The dashed blue-lines correspond to plus and minus the model-truth standard deviation profile. The (b)–(f) plots correspond to retrievals conducting using Platt's approach while the (g)–(k) plots correspond to retrievals conducted using the default multiple-scatting (Platt+tails) model.

## 3.2 Baja

The attenuated co-polar Mie, Rayleigh, and cross-polar backscatter signals as well as the corresponding A-FM Feature-mask (van Zadelhoff et al., 2023) for the Baja scene are shown in Fig. 14. The corresponding model-truth extinction, the retrieved low-resolution extinction field as well as details of two selected areas are shown in Fig. 15. The two sections selected here are both cloud-free with no overlying semi-transparent cloud layers. The extinction are generally retrieved to within 10-15%. The corresponding lidar-ratio results are shown in Fig. 16. Here it can be seen that the lidar-ratios are retrieved usually within about 10-20 %.

A detailed view of an ice cloud region in shown in Fig. 17. Here it can be seen that, consistent to what was observed in general for the Halifax scene, that the extinction profile is well-retrieved (within 5-10%), however, the lidar-ratio is underestimated (especially below 4 km). This is likely largely in part due to $f_{MSp}$ not being set optimally. Even though below 4 km the ice cloud is giving way to optically thinner and smaller particle aerosol, the multiple-scattering ratio is still high (see Fig. 18), thus, it can be important to treat $f_{MSp}$ accurately even if the target in question possesses a small effective radius and is not optically thick if underlies a layer which generates significant amount of multiple-forward-scattered light.

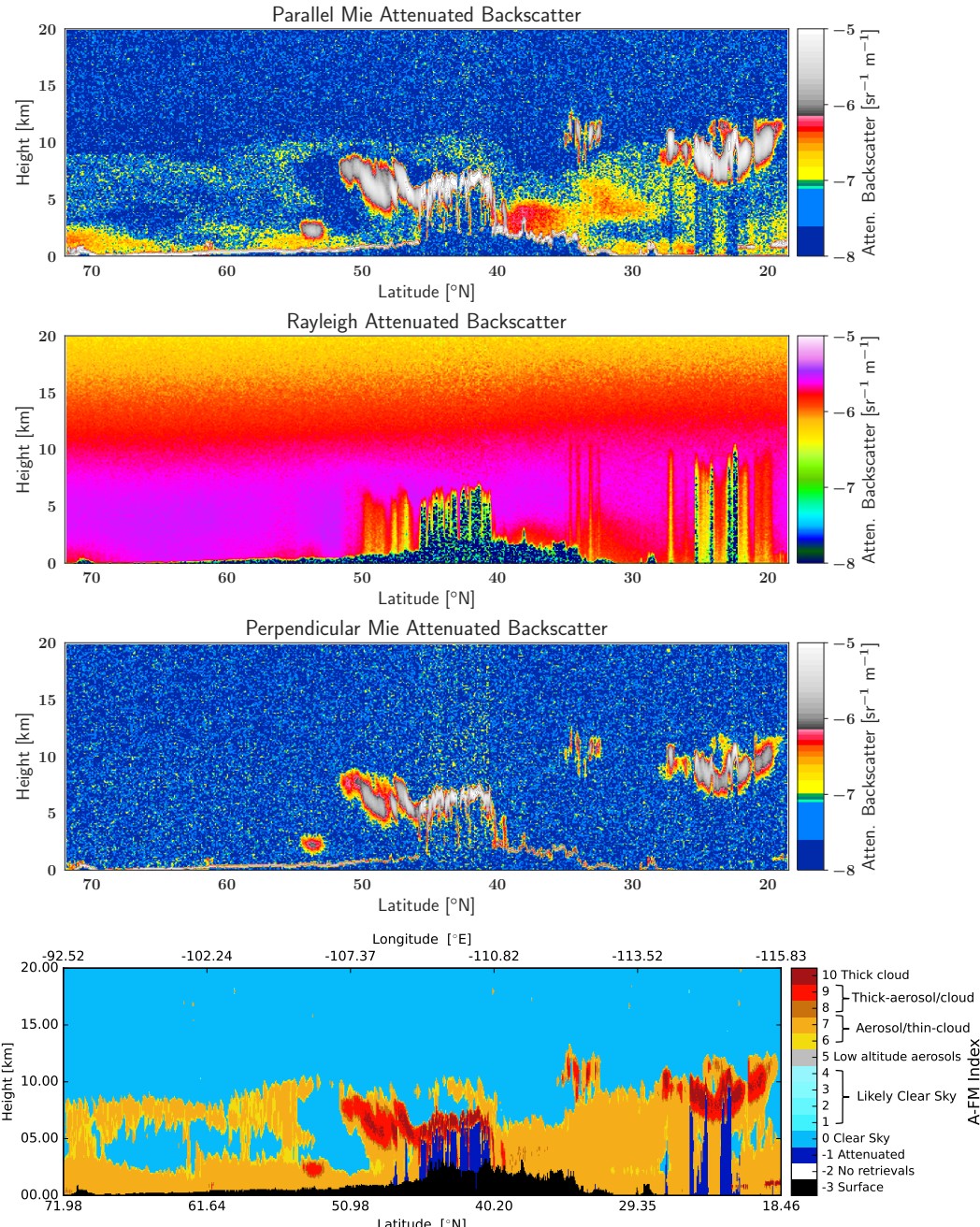

**Figure 14.** Simulated Mie,Rayleigh, and Cross-polar Attenuated back scatters for the Baja scene along with the corresponding A-FM L2 Featuremask.

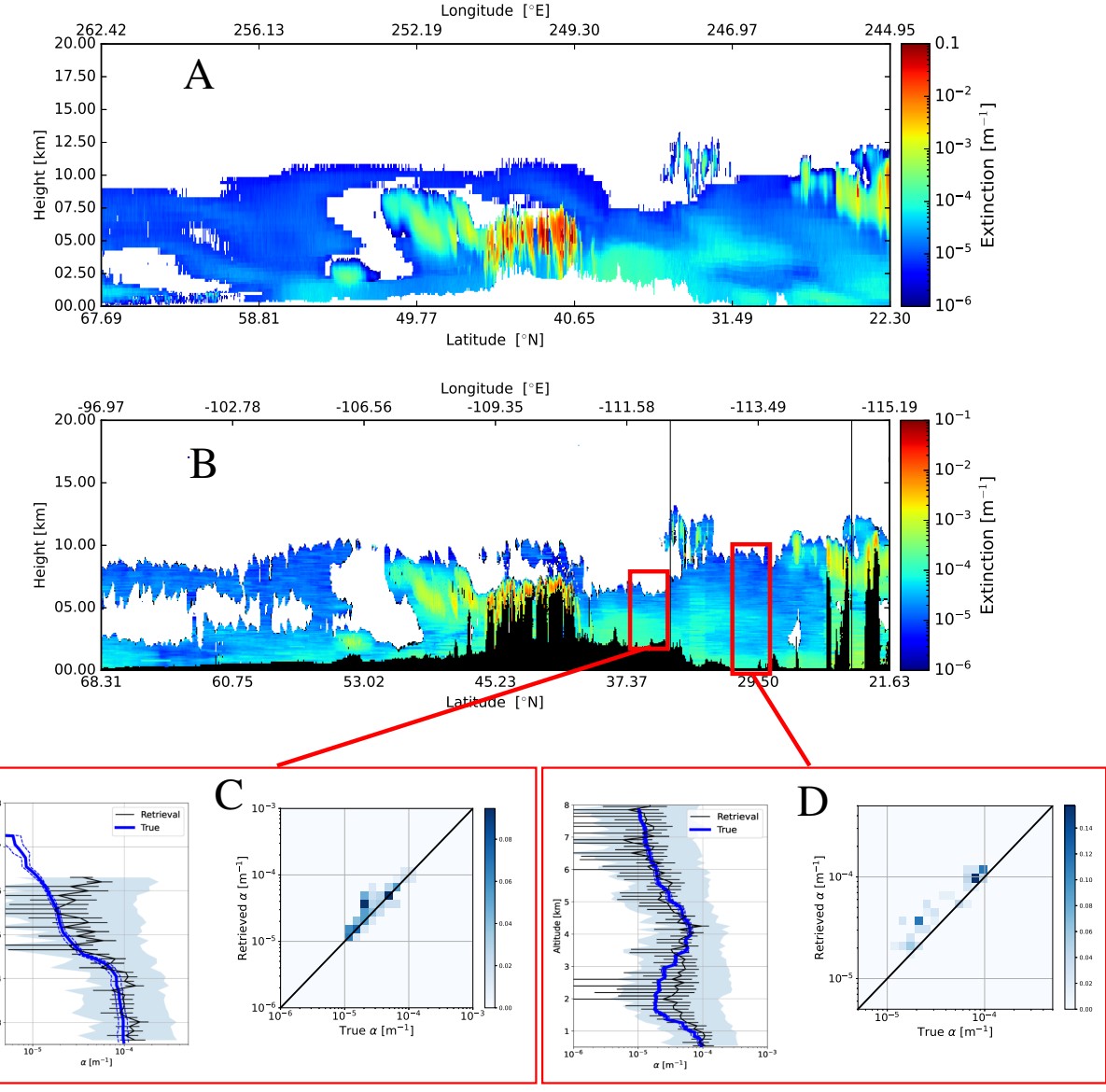

**Figure 15.** Model-truth extinction (panel-A) and low-resolution retrieved extinction for the Baja scene (panel-B) and two representative average retrieved profiles (panels C and D) for the indicated intervals. The bottom panels correspond to the low-resolution (100km resolution)and the corresponding model-truth extinction profiles.

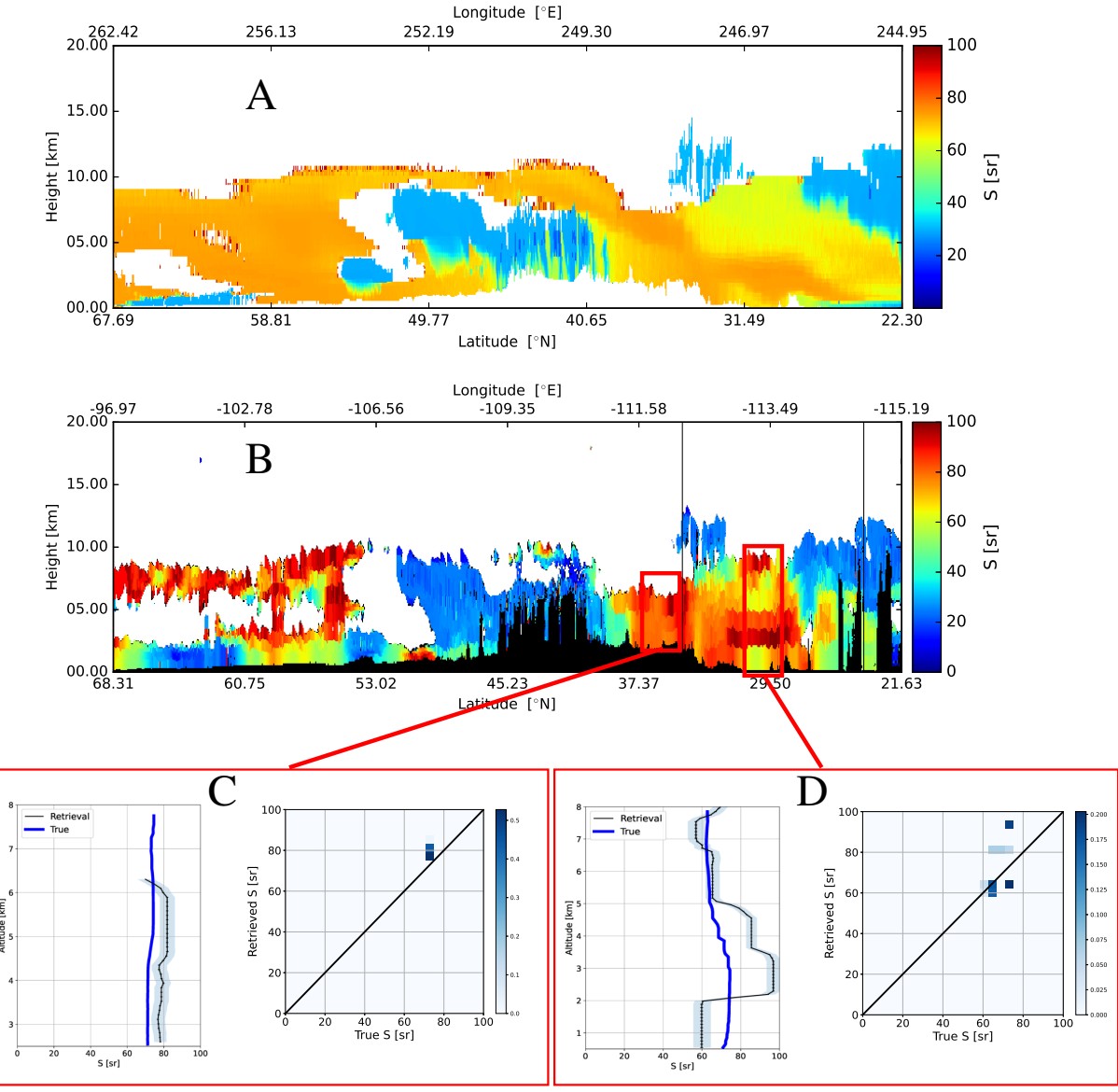

**Figure 16.** As Fig. 15 except that lidar-ratio is considered.

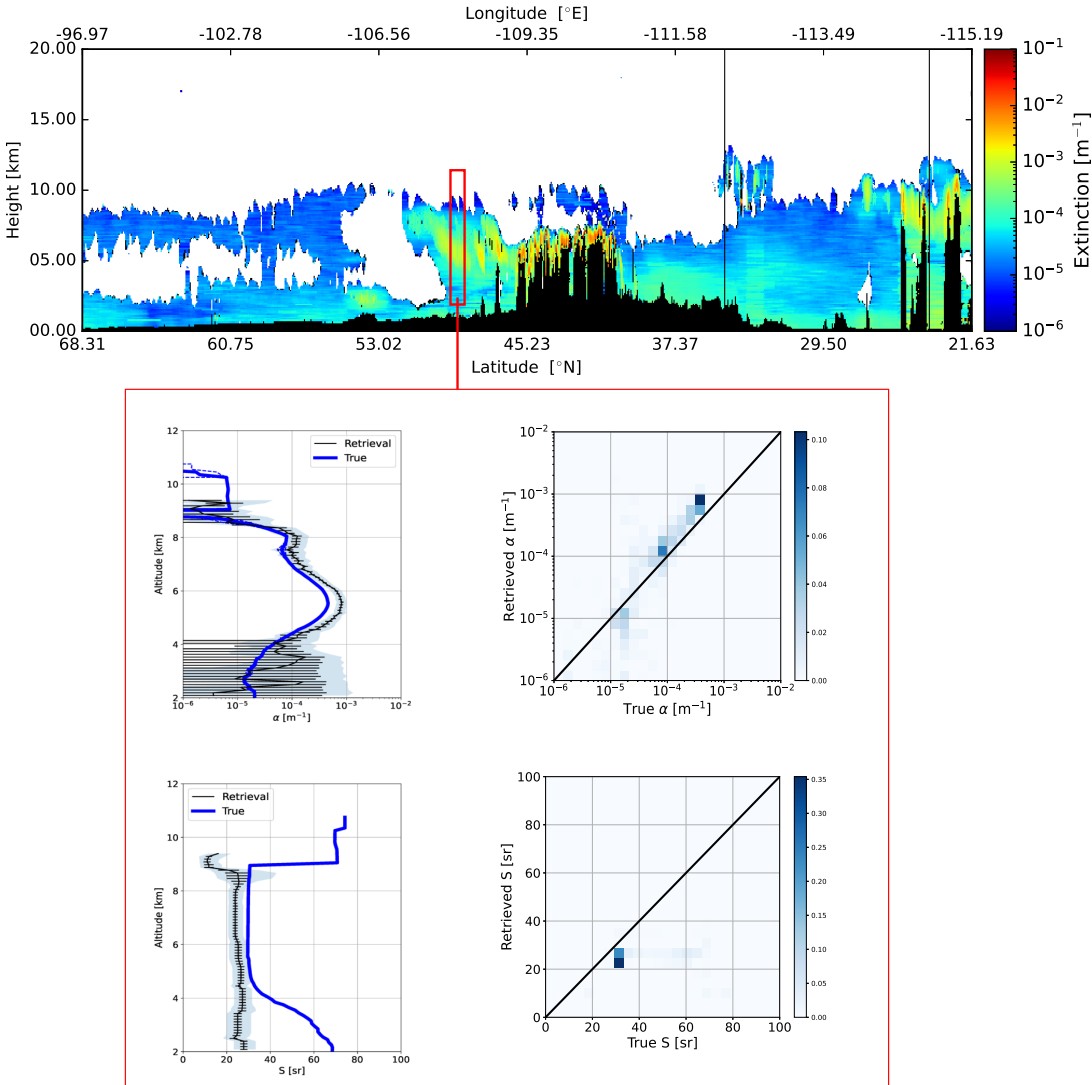

**Figure 17.** Low-resolution retrieved extinction for the Baja scene (top-panel). The bottom panels correspond to the indicated 10km horizontal indicated region.

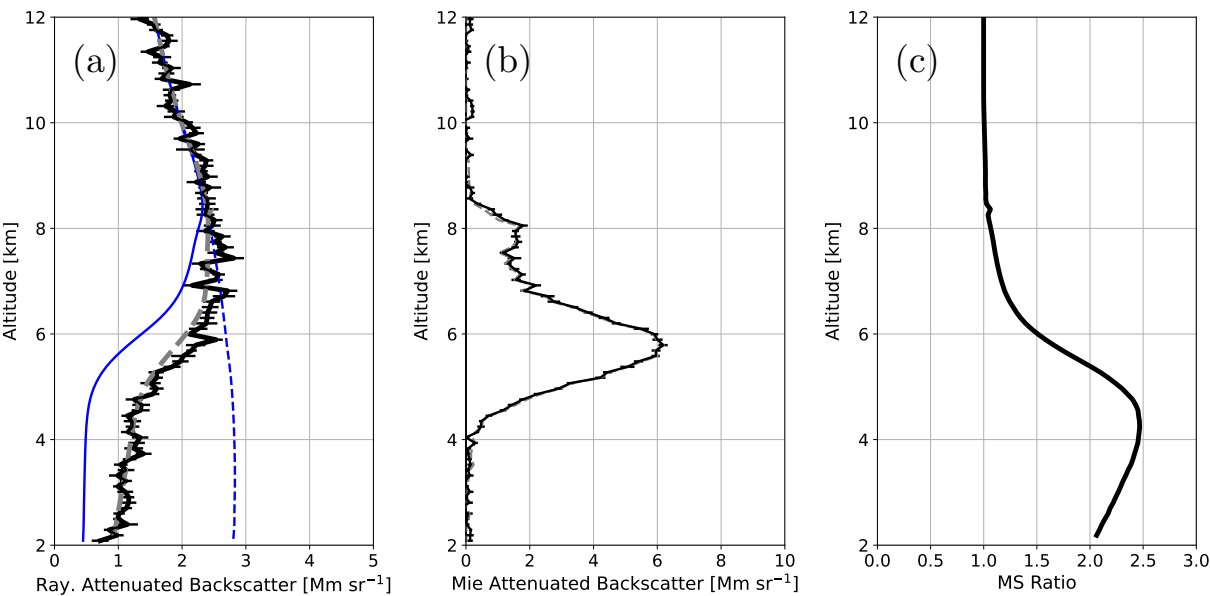

**Figure 18.** (a): Average Rayleigh attenuated backscatter for the interval indicated in Fig 17 (black line), the forward-model fit (grey dashed line), the corresponding single-scatter Rayleigh attenuated backscatter (solid blue line) and the Rayleigh-clear attenuated backscatter profile. (b): Average Mie attenuated backscatter for the indicated interval (black line) and corresponding forward-model fit (grey dashed line). (c): Ratio of (total) multiple-scattering plus single scattering to single scatting return for the Rayleigh backscatter.

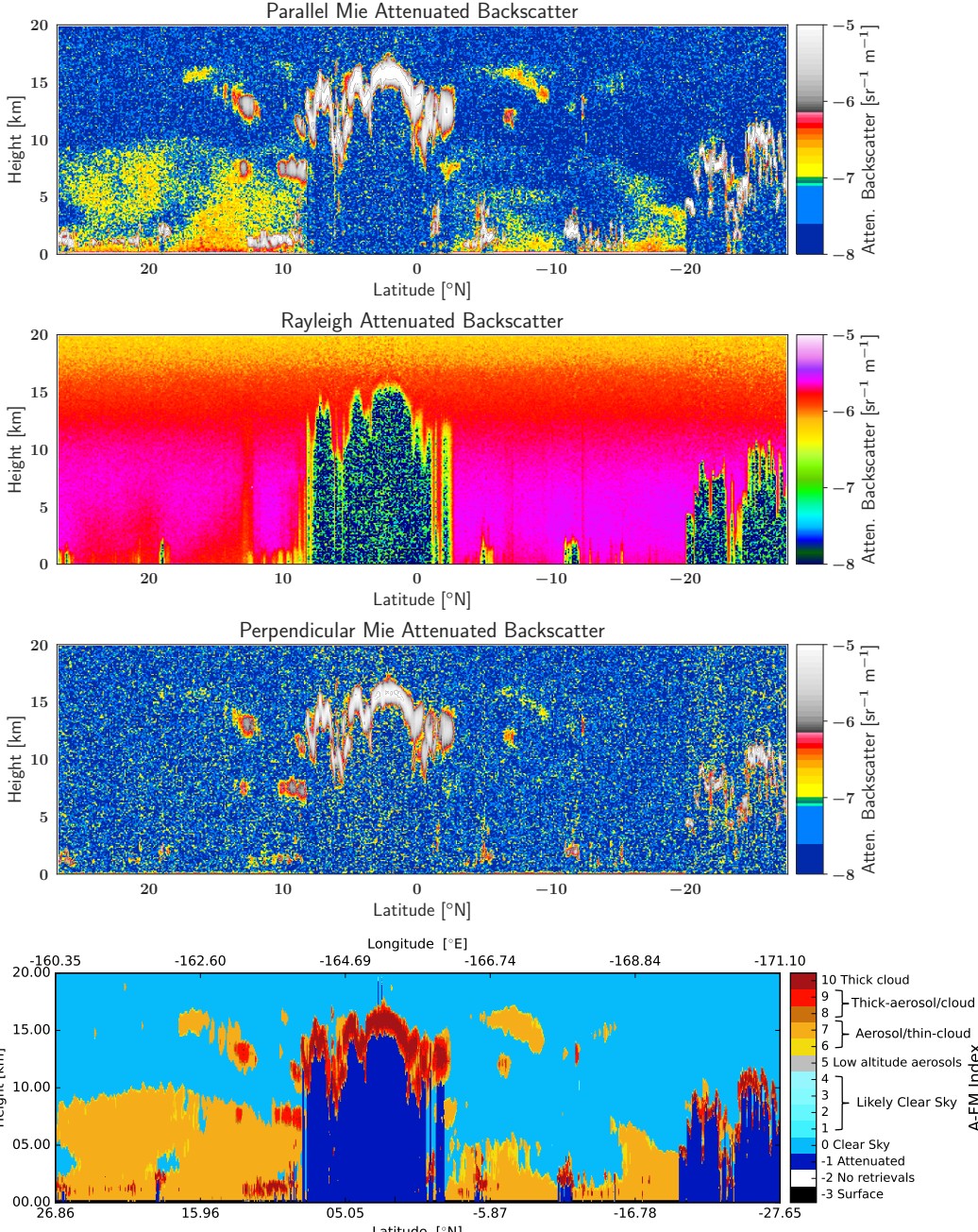

**Figure 19.** Simulated Mie,Rayleigh, and Cross-polar attenuated back scatters for the Hawaii scene along with the A-FM L2 Featuremask.

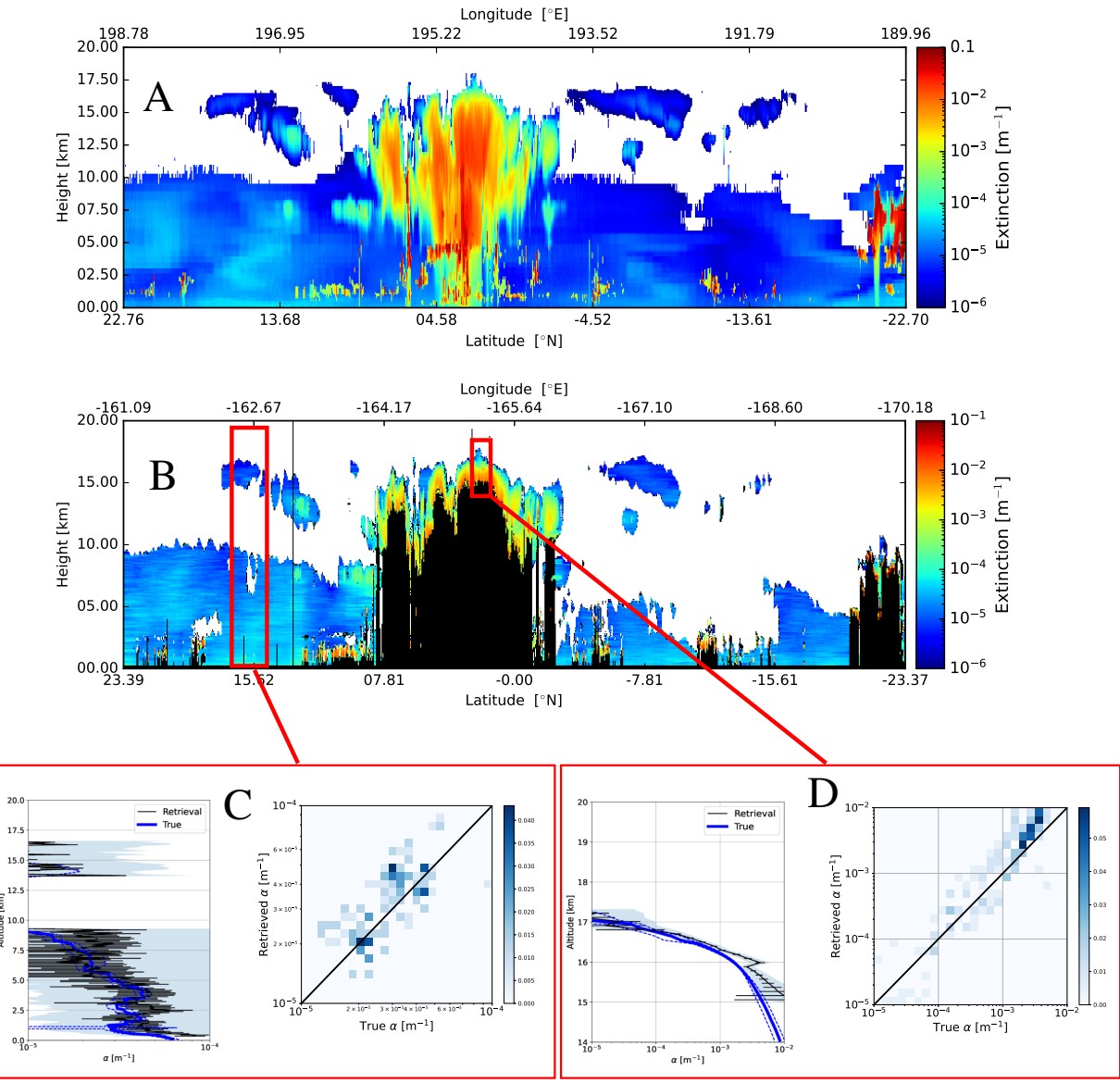

**Figure 20.** Model-truth extinction (panel-A) and low-resolution retrieved extinction for the Hawaii scene (panel-B) and two representative average retrieved profiles (bottom-panels) for the indicated intervals. The panel-C correspond to the low-resolution (100km resolution) and the corresponding model-truth extinction profiles while panel-D corresponds to a 10 km horizontal interval.

The attenuated co-polar Mie, Rayleigh, and cross-polar backscatter signals as well as the corresponding A-FM Feature-mask for the Hawaii scene are shown in Fig. 19. The corresponding model-truth extinction, the retrieved low-resolution extinction field as well as details of two selected areas are shown in Fig. 20. The extinction accuracy is seen to be consistent with the other two previously discussed scenes. Referring to the corresponding lidar-ratio details shown in Fig. 21 it can be seen that for the

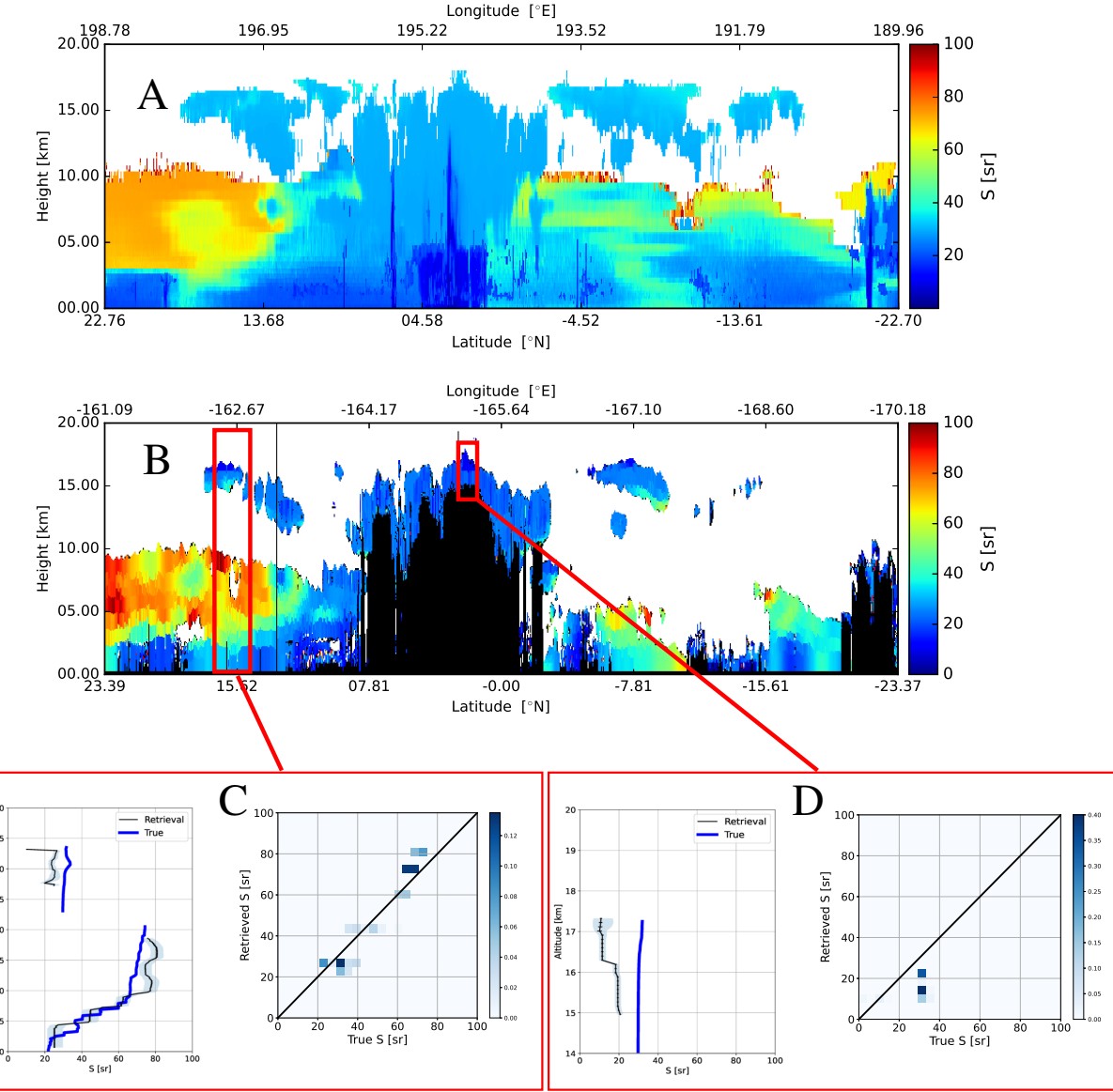

**Figure 21.** As Fig. 20 except that lidar-ratio is considered.

left selected area, that in this case, where the upper layer ice cloud is optically quite thin that it does not seen to much effect the retrieval of underlying lidar-ratio. For the optically think ice-cloud selected area, the extinction profile is well-retrieved near up to the point of complete attenuation, however, as was seen before the retrieved lidar-ratio is biased low.

## 4 Conclusion and Outlook

The accurate retrieval of aerosol and cloud properties from space-based lidar is a challenging endeavor, even when the extra information provided by an HSRL system is exploited. Accounting for the generally low SNR ratios involved coupled with the need to respect the structure of the aerosol and cloud fields being sensed present particular challenges. A-PRO addresses these challenges by implementing a multi-scale approach resulting in a viable practical approach for both clouds and aerosols.

In this paper, a detailed overview of the A-PRO processor has been presented. The focus has been purposely limited to the extinction and lidar-ratio retrievals. For a more complete picture, the interested reader should also consult Irbah et al. (2023) where the layer determination and classification procedures are detailed.

The development of the A-PRO processor has mainly been based on synthetic data (though a "cousin" algorithm, called AEL-PRO has been applied to Aeolus (Straume, A.G. et al., 2020) data which is the subject of another paper (Wang et al., 2024)) and, no doubt, further refinements and extensions will be made when actual ATLID observations are available. One of the issues that has been noted, and indeed highlighted within this paper, is the potential difficulty in properly accounting for multiple-scattering effects. Retrievals below higher scattering layers can be affected by the presence of "tails" which can be difficult to accurately model. Further, it is likely that a more comprehensive investigation and treatment of the role of anisotropic backscattering of multiple-scattered light (i.e. issues surrounding the $f_{Msp}$ factor. It seems that often, in situations where multiple-scattering is important, that the extinction is more robustly retrieved than the lidar-ratio. This is not surprising given that the extinction information is closely related to the Rayleigh signal profile which can be modeled, independently of $f_{Msp}$. The modeling of the Mie backscatter signal involves both $S$ and $f_{Msp}$, this extra degree of freedom can create additional uncertainty when retrieving the backscatter (or equivalently the lidar-ratio). Based upon further simulation-based studies as well as actual ATLID observations, this issue will be revisited and may results in extensions to the A-PRO procedures (i.e. parameterization of $f_{Msp}$ or including $f_{Msp}$ in the state-vector of the A-EBD optimal-estimation algorithm).

*Data availability.*  The EarthCARE Level-2 nadir model-truth data and the simulated L2 products discussed in this paper, are available from https://doi.org/10.5281/zenodo.7117115.

*Author contributions.*  The lead author of this paper (D.P. Donovan) was also the lead developer of the A-PRO processor. G-J van Zadelhoff contributed to the writing of the code and was the lead developer of the A-TC components. P. Wang assisted in the evaluation and testing of the code. All authors contributed to the writing and editing of the manuscript.

*Acknowledgements.*  This work has been funded by ESA grants 22638/09/NL/CT (ATLAS), ESA ITT 1-7879/14/NL/CT (APRIL) and 4000134661/21/NL/AD (CARDINAL). We thank Tobias Wehr, Michael Eisinger and Anthony Illingworth for valuable discussions and their support for this work over many years. A special acknowledgement is in order for Tobias Wehr, ESA's EarthCARE Mission Scientist

who unexpectedly passed away recently. Tobias was eagerly looking forward to EarthCARE's launch, a mission to which he had dedicated a considerable span of his career. His support for the science community, collaborative approach and enthusiasm for the mission science will not be forgotten.

*Competing interests.* The authors declare that they have no conflict of interest.

## Appendix A:  A-AER extinction, backscatter and lidar-ratios retrieval methods

### A1    Standard Estimation of lidar-ratio, extinction, and backscatter

Neglecting multiple-scattering effects for the time being, using Eq. 8 and using the fact that $b_M = \beta_M \exp[-2(\tau_M + \tau_{Ray})]$ we have

$$B_{M,hv} = F_{hv}\left(\beta_M \exp[-2\tau_M]\right). \tag{A1}$$

Further, using Eq. 9 and using the fact that $b_M = \beta_M \exp[-2(\tau_M + \tau_{Ray})]$

$$B_{R,hv}^{Rat} = F_{hv}\left(\exp[-2\tau_M]\right) \tag{A2}$$

where the $Rat$ superscript is used to denote the fact that the ratio between the observed Rayleigh extinction corrected attenuated Rayleigh backscatter and the unattenuated Rayleigh backscatter is being calculated. $F_{hv}$ is used to denote the masked vertical
and horizontal smoothing operation, and $\tau_{Ray}$ is the Rayleigh optical depth from the Top-of-Atmosphere to the height in question. Here, for simplicity the range dependence is not written explicitly.

If we assume that the action of the smoothing operation can be ignored then, taking the range derivative of Eq. A2 and re-arranging yields an expression for the particulate extinction i.e.

$$\alpha_M(r) = -\frac{1}{2}\frac{dB_{R,hv}^{Rat}}{dr}\frac{1}{B_{R,hv}^{Rat}}, \tag{A3}$$

and dividing Eq. A2 by Eq. A1 yields an expression for the particulate backscatter coefficient i.e.

$$\beta_M(r) = \frac{B_{M,hv}}{B_{R,hv}^{Rat}}. \tag{A4}$$

Dividing the two previous equations then yields an expression for $S$, i.e.

$$S(r) = -\frac{1}{2}\frac{dB_{R,hv}^{Rat}}{dr}\frac{1}{B_{M,hv}}. \tag{A5}$$

Deriving the extinction, backscatter and the associated lidar extinction-to-backscatter ratio using any approach related to that
just outlined in which the signals are smoothed either explicitly or implicitly can lead to inaccurate results for the determination of $S$ e.g. near the edges of clouds or aerosol regions. This is due to fact that the extinction information (which is related to the signal derivative) and the backscatter information are linked to different vertical scales (i.e. the action of $F_{h,v}$ can not be ignored in the above derivation). In particular, the vertical derivative of the ATBs can not be unambiguously be linked to a single range-bin but can only be assess using two or more bins, however, the backscatter can be assessed, in principle at the
scale of a single range-bin. This fundamental difference in the scale at which the extinction and backscatter information can be retrieved gives rise to undesirable edge effects. This problem can be made worse when vertical smoothing of the ATBs over a number of range-gates must be applied in order to increase the effective SNR as is done here. In effect, applying the same

smoothing strategy to both the Rayleigh and Mie ATBs, due to their dissimilar structure, does not result in the $S$-ratio being preserved even in cases where no noise were to present.

One of the uses of the $S$ profiles within the A-AER is to help determine the layer-structure (See steps 10–11 in Section 2.2) and spurious features in the $S$ profile can give rise to spurious layers. In part for this reason, an alternation procedure was developed and implemented which tends to produce fewer edge effects in the $S$ determination process. This procedure is described subsequently.

## A2    Local Forward-modeling based estimation of S, extinction and backscatter

An approach which attempts limit the issues involved with spurious edge effects with $S$ profile determinations is to perform a local forward model fit which, in a sense, puts the retrieved extinction and backscatter on the same scale. The basic idea is to find the best value of $S$ over a vertical fitting window which, together with the conventionally derived backscatter profile, best predicts the observed Mie and Rayleigh ATB profiles.

As a starting point, the backscatter profile and extinction profiles and the subsequent values of $S$ are determined using
Eq. A4. Then the algorithm proceeds as follows.

1. For the fitting window the average backscatter is determined

$$\overline{\beta_M} = \frac{\sum_{i=i_{bot}}^{i_{top}} \beta_{M,i}}{N} \tag{A6}$$

   $i_{top}$ and $i_{bot}$ are the range indices of the fitting window boundaries and $N$ is the number of range-gates in the fitting window.

2. Using a specified value of $S$, the average particulate extinction within the fitting window $\overline{\alpha_M}$ is estimated i.e.

$$\overline{\alpha_M} = S\overline{\beta_M} \tag{A7}$$

3. The un-normalized predicted local profiles corresponding to $B_{R,hv}^{Rat}$ and $B_{M,hv}$ are calculated as:

$$B_{R,hv,i}'^{Rat} = \exp[-2\tau_{M,i}] \tag{A8}$$

and

$$B_{M,hv,i}' = \overline{\beta_M} \exp[-2\tau_{M,i}] \tag{A9}$$

   where $\tau_{M,i} = \alpha_M(r_i - r_o)$, where $r_o$ is the value of the range gate closest to the lidar within the fitting window and $r_i$ is the range of the $i$th range-gate within the fitting window.

4. The local calibration factor ($C_{loc}$) which normalizes the profiles calculated in the previous step with respect to the observations is calculated i.e

$$C_{loc} = \frac{\sum \left( B_{M,hv,i} + B_{R,hv,i}^{Rat} \right)}{\sum \left( B'_{M,hv,i} + B'^{Rat}_{R,hv,i} \right)} \tag{A10}$$

where the summation is carried out over the fitting window.

5. The Chi-sq difference between the local forward–modelled and the corresponding observations is calculated as well as its derivative with-respect to $S$ i.e.

$$\chi^2 = \sum \left( \frac{B_{M,hv,i} - C_{loc} B'_{M,hv,i}}{\sigma_{B_{M,hv,i}}} \right)^2 + \sum \left( \frac{B_{R,hv,i}^{Rat} - C_{loc} B'^{Rat}_{M,hv,i}}{\sigma_{B_{R,hv,i}^{Rat}}} \right)^2 \tag{A11}$$

and

$$
\begin{aligned}
\frac{d\chi^2}{dS} = \; & 2 \sum \left( \frac{C_{loc} B'_{M,hv,i} - B_{M,hv,i}}{\sigma_{B_{M,hv,i}}^2} \right) \overline{\beta_M} \left( -2 C_{loc} \exp[-2\tau_{M,i}] \frac{d\tau_{M,i}}{dS} + \exp[-2\tau_{M,i}] \frac{dC_{loc}}{dS} \right) \\
+ \; & 2 \sum \left( \frac{C_{loc} B'^{Rat}_{R,hv,i} - B_{R,hv,i}^{Rat}}{\sigma_{B_{R,hv,i}^{Rat}}^2} \right) \left( -2 C_{loc} \exp[-2\tau_{M,i}] \frac{d\tau_{M,i}}{dS} + \exp[-2\tau_{M,i}] \frac{dC_{loc}}{dS} \right)
\end{aligned}
\tag{A12}
$$

where,

$$\frac{d\tau_{M,i}}{dS} = \frac{\tau_{M,i}}{S} \tag{A13}$$

and

$$\frac{dC_{loc}}{dS} = 2 \frac{\sum \left( B_{M,hv,i} + B_{R,hv,i}^{Rat} \right)}{\left( \sum B'_{M,hv,i} + B'^{Rat}_{R,hv,i} \right)^2} \sum \left( \overline{\beta_M} + 1 \right) \exp[-2\tau_{M,i}] \frac{d\tau_{M,i}}{dS} \tag{A14}$$

Here, the error estimates (the $\sigma$ term) are calculated using standard-error propagation techniques based on the estimated random error in the observed Rayleigh and Mie attenuated backscatters.

6. The value of $S$ that minimizes $\chi^2$ is found numerically using the Secant method applied to $\frac{d\chi^2}{dS}$. For the initial iteration, the values of $S$ generated by the application of Eq. (A4) are used. The Secant iteration continues (looping back to step 5.) until a maximum number of iterating is reached (usually set to 10) or successive value of $S$ are within a set tolerance (e.g. 1%).

7. The uncertainty in the retrieved value of $S$ is estimated according using a scaled Chi-sq approach i.e.

$$\sigma_S = \sqrt{2 \left( \frac{d^2\chi^2}{dS^2} \right)^{-1}} \sqrt{\frac{\chi^2_{Min}}{i_{top} - i_{bot} - 2}} \tag{A15}$$

where $\chi^2_{Min}$ is the minimum value of the cost-function obtained in the previous step.

8. Using the backscatter profile and $S$ the extinction profile is determined along with its associated uncertainty.

The procedure described above is numerically based, but it is fast and does have an advantage compared to more conventional methods, in particular the $S$ retrievals are better behaved near the edges of scattering layers. This is illustrated in Fig.A1 where the results of an simple idealized simulation are presented. Here, a simple two–layer aerosol field was used to simulate Rayleigh and Mie attenuated backscatter profiles. Panel B shows the noiseless ATB signals both at the native ATLID resolution of 100m and at a retrieval resolution of 300 m, it is clear that the smoothing affects the Mie ATB signals more than the corresponding Rayleigh profile. This is further reflected by the large oscillation in the extinction-to-backscatter retrieved using the conventional approach. The local-forward-modelling approach though yields generally superior results near the layer boundaries.

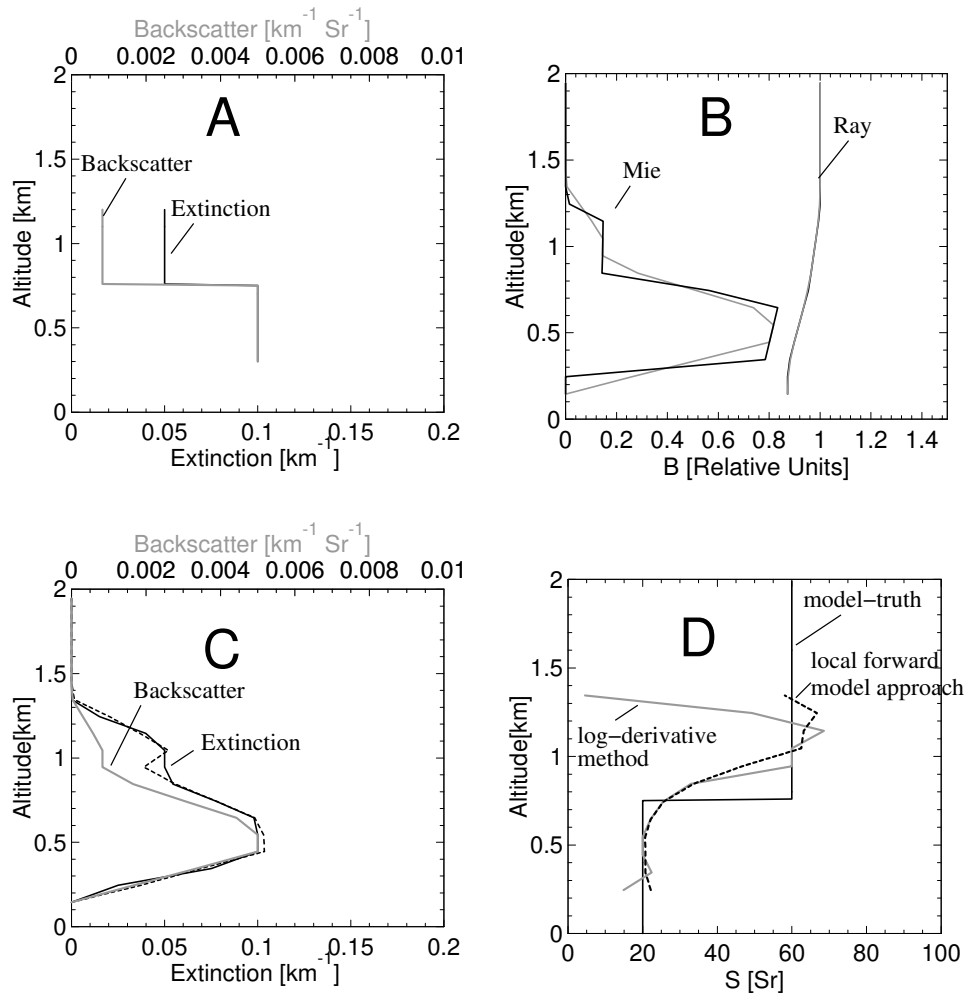

**Figure A1.** A: Profiles of model-truth extinction and backscatter. B: Attenuated Rayleigh and Mie backscatter profiles at 0.1 km resolution (black) and 0.3 km (Grey). C: Retrieved backscatter (grey line) and retrieved extinction using the conventional log-derivative approach (solid black line) and the local-forward-modelling approach (dashed black line). D: The solid black line corresponds to the model truth extinction-to-backscatter ratio while the Grey-solid-line represents the conventional log-derivative approach estimate while the dashed black line corresponds to the result from the local-forward-modelling approach.

## Appendix B: Treatment of multiple-scattering for ATLID in A-PRO

The inversion of ATLID lidar signals requires a fast, yet accurate treatment of lidar multiple scattering. Example simulated signal profiles for an idealized cirrus cloud are shown in Fig. B1. Here three different models of different complexity have been applied, namely Monte-Carlo (MC), the reduced extinction approach due to Platt (Platt, 1981), and the quasi-small-angle (QSA) Hogan (2006) and Photon Variance-Covariance Methods (PVC) (Hogan, 2008). The Monte-Carlo results were calculated using the EarthCARE Simulator (ECSIM) lidar radiative transfer model (Donovan et al., 2015) and are, here, considered the most accurate. Here it can be seen that the QSA results are close to the MC results both within and below the cloud. Platt's approach performs well within the confines of the cloud but can not follow the shape of the "tails" arising from the decay of the signal towards single-scattering levels below the cloud base.

Hogan's approach is accurate and orders of magnitude faster than MC calculations. However, it is still much slower than the corresponding single-scattering case. Platt's approach on the other hand, is as fast as calculating the single-scattering return. In this work, a novel approach is developed which, in terms of complexity, is between the approaches due to Platt and Hogan.

We first discuss Platt's effective extinction approach and discuss how this can be extended to handle the phenomenon of decaying "tails".

### B1 Platt's approach

Within Fig. B1, it can be noted that, within the cloud, that the observed signal closely resembles a less attenuated version of the single-scattering signal. This is to be expected when the particles are large compared to the wavelength of the laser light so that half the scattered energy is scattered forward in a narrow diffraction lobe and largely stays within the lidar receiver file-of-view. This result was noted by Platt (1976) and forms the basis of a simple method for accounting for Multiple-scattering effects.

If one defines

$$M_p(z) = \frac{P(z)}{P_{ss}(z)} = \exp\left[2\int_0^z (1 - \eta_p(z'))\alpha(z')dr(z')\right] \tag{B1}$$

where $M_p$ is the ratio of the total received power including all scattering orders ($P$) and the single-scatter power $P_{ss}$. $\eta_p$ is the multiple scattering effective extinction factor such that $(1 - \eta_p)$ is the fraction of scattered energy that remains within the lidar field-of-view (and thus behaves like it has not be scattered). If one then multiplies this expression by an expression for the single-scatter lidar attenuated backscatter then Platt's multiple-scattering equation is recovered i.e.

$$B(z) = \beta(z)\exp\left[-2\int_0^z \eta(z')\alpha(z')dr(z')\right] \tag{B2}$$

### B2 The origin of MS tails

As mentioned above, Platt's approach is fast but severely limited in cases where "tails" may be present. These 'tails' are not due to temporal pulse stretching, which is associated with thick clouds (see e.g. Hogan and Battaglia (2008) ). The origin of

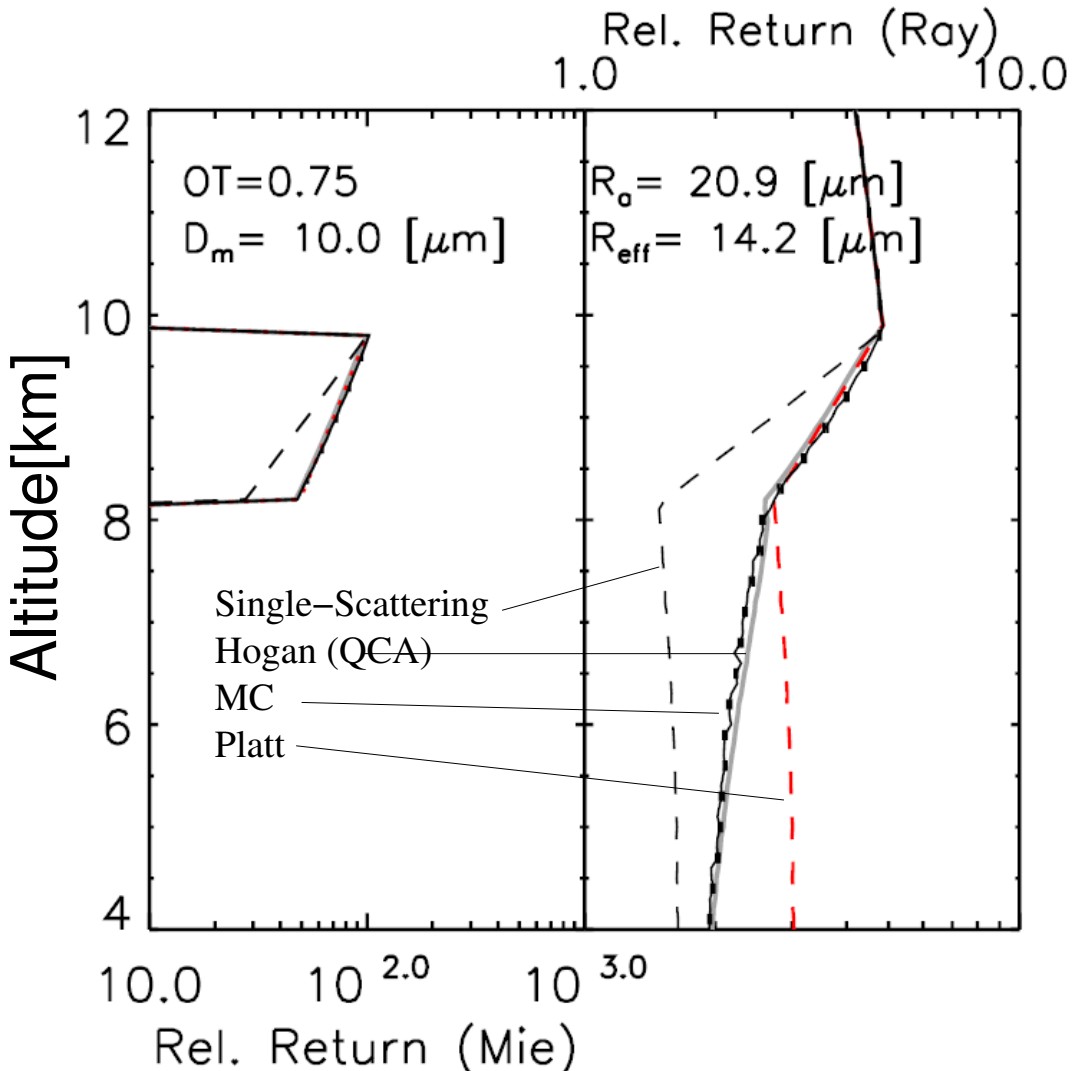

**Figure B1.** Sample comparison between the ECSIM lidar Monte-Carlo multiple scattering model (Donovan et al., 2023), the analytical model due to Hogan and Battaglia (2008), and Platt's equation B2 for an idealized single layer ice cloud of optical thickness (OT) or 0.75 and an effective radius of 30.7 microns. (Left) Mie co-polar and (Right) Rayleigh channel co-polar returns. Black-solid line (with error-bars): ECSIM results. Dashed-Black line: Single scattering results. Solid Grey line: Hogan's model results for the true value of $R_a$. Red line: Platt's equation.

the tails in question here can be simply explained. Referring to Fig. B1, within the cloud the low mean-free-path of the photons ensures that the multiple-scattered light that contributes to the detected signal tends to be confined to within the field-of-view

of the lidar, however, the angular variance of the lidar beam will increase as it propagates downwards through the cloud with more and more photons undergoing scattering events.

At cloud base, the lidar beam emerges with an effective angular divergence which increases with the optical thickness of the cloud and decreases with the size of the cloud particles. This is due to that fact that the angular-width of the cloud phase function forward lobe increases with decreasing particle size i.e.

$$\theta_{sc} = \left( \frac{\lambda}{\pi R_a} \right) \tag{B3}$$

Below cloud base the lidar beam will continue to propagate with a given divergence. However, the horizontal spread of the photons is no longer constrained by the presence of the cloud. As the beam continues to propagate downwards, depending of the lidar receiver footprint more and more of the multiple-scattered photons will travel outside of the receiver cone. Though less pronounced, mainly due to the larger fov, such tails can be found in Calipso observations.

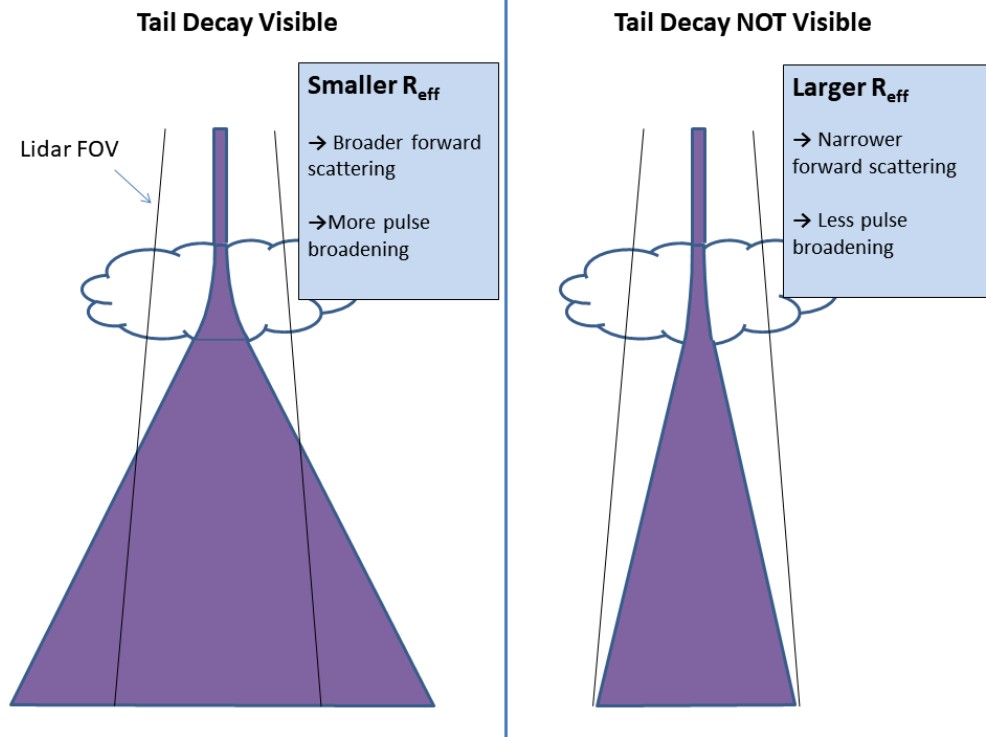

**Figure B2.** Schematic depiction of the mechanisms behind the occurrence of decaying tails below scattering layers. Here the Purple regions represent the broadened laser pulse extent.

## B3 An Extension to Platt's approach

Following the discussion in the previous section, it is apparent that the tails can be viewed as a decay of the signals towards single-scattering levels. Accordingly, we modify the form of the multiple-scattering ratio Eqn. B1 to allow for such decays to occur i.e.

$$M(z) = (1 - f(z)) + f(z) \exp\left[2\eta(z)\tau(z)\right] \tag{B4}$$

where $\eta = 1 - \eta_p$ (here assumed to be constant per atmospheric layer) and $f(z)$ is the range-dependent multiple scattering return signal fraction. Note that here we have replaced $\eta_p$ by $\eta$ for reasons of convenience, that is, $\eta$ is equal to the fraction of multiply-scattering light remaining in the field-of-view (instead of $1 - \eta_p$). Note that if $\eta$ is equal to zero then $M(r) = 1$ (regardless of the values of $f(z)$) and no multiple-scattering is detected. Also, if $f(z)$ is equal to zero $M(z) = 1$ (regardless of the values of $\eta(z)$) and if $f(z)$ is equal to 1 then Eq. B4 reduces to Eq. B1.

If we multiply Eq. B4 by the single-scatter lidar attenuated backscatter expression then

$$B(z) = \beta(z) \exp\left[-2\tau(z)\right] \left((1 - f(z)) + f(z) \exp\left[2\eta(z)\tau(z)\right]\right) \tag{B5}$$

### B3.1 Determination of $f(z)$

To be useful, a means to determine the profile of $f(z)$ must be established.

We start by considering the case of a physically thin scattering layer as schematically depicted in Fig. B3 . If we assume that the beam has a Gaussian profile, and the forward-scattering lobe of the effective layer phase function is approximated by a Gaussian (Eloranta, 2005), then the divergence of the forward scattered light will also be Gaussian with a divergence given by the convolution of the incoming beam divergence ($\theta_l$ ) with the effective scattering forward-lobe width ($\theta_{Sc}$ ) so that the effective width of the multiply scattering radiation emerging from the layer bottom is given by

$$\theta_{eff} = \left(\theta_{Sc}^2 + \theta_l^2\right)^{1/2}. \tag{B6}$$

By integrating the effective beam across the lidar Field-of-View (fov) the fraction of the light that remains within the fov is given by

$$f(z, z_l) = 1.0 - \exp\left[-\left(\frac{\rho_t{}^2(r(z) - r_{Sat})^2}{\theta_{sc}{}^2(r(z) - r(z_l))^2 + \theta_l{}^2(r(z) - r_{Sat})^2}\right)\right] \tag{B7}$$

where $\rho_t$ is the receiver telescope full-angle angular fov, $\theta_l$ is the laser full-angle divergence, $r_{sat}$ is the satellite altitude and $r(z_l)$ is the altitude of the scattering layer. This expression is only valid for a single thin scattering layer and so, by itself, is not so useful. However, we can generalize this expression to the case of a general profile in a heuristic approximate fashion. A rigorous calculation would be involved and would result in a similar formalism as the QSA model of Hogan (2008). Here we will use the information present in the signal itself to calculate the effective $f$ profile under general conditions. Since the

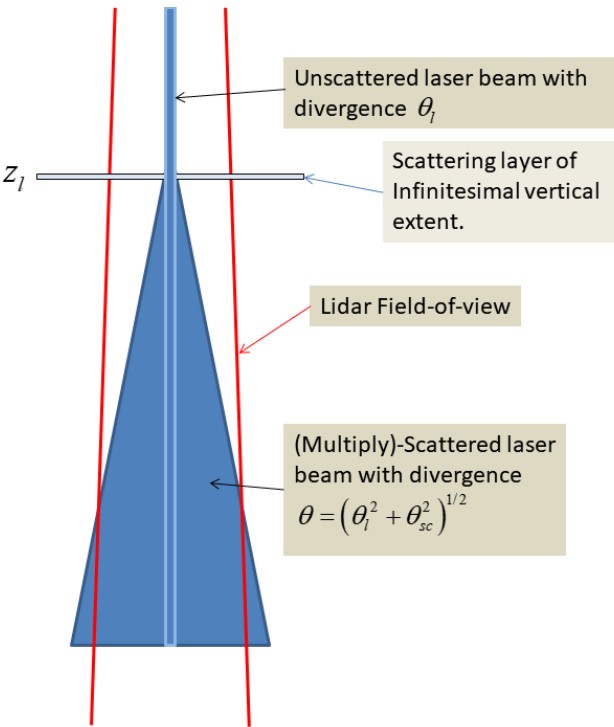

**Figure B3.** Schematic depiction of the angular broadening experienced by a lidar pulse as it interacts with a physically thin scattering layer at altitude $r(x_l)$.

observed signal itself contains information on the location and relative strength of the scattering at each level, we postulate the form

$$f_e(z) = \frac{\int_{z_{sat}}^{z} f(z, z_l) B(z) dz_l}{\int_{z_{sat}}^{z} B(z) dz_l} \tag{B8}$$

where $B$ is the total observed attenuated backscatter. That is, we use the observed backscatter profile itself as a weighting factor to determine the effective profile. In the limit of a single thin scattering layer this expression yields the correct result.

An example comparison between Monte-Carlo calculations, Platt's approach and the "Platt+Tails" approach (Eq. (B5) together with Eq. B8) is shown in Fig. B4. Here a fitting procedure was used to find the best values of $\eta$ and $\theta_{Sc}$ for each of the two layers. It can be seen that the extended Platt approach provides a very good match to the MC results for the entire profile while the normal Platt approach is deficient.

As a further refinement, in order to account for the fact that the effective backscatter coefficient for multiple-scattered light may, in general, be lower than that than associated with single-scattering (Hogan, 2008; Wandinger, 1998; Eloranta, 1998) an additional factor is added which acts to adjust Eq. B8 based on the relative amount of particulate scattering i.e.

$$f_{M,e}(z) = f_{Msp}(z) f_e(z). \tag{B9}$$

where $f_{MSp}$ is a factor which accounts for reduced backscattering of multiply scattering light due to the existence of a backscatter peak in the particle phase functions. It was previously thought that ice particles would not posses a strong backscatter peak. However, newer results indicate that that even irregular and rough crystals will possess a backscatter peak (Zhou and Yang, 2015). Molecular Rayleigh scattering possesses an effectively isotropic phasefunction in the backscatter direction, thus no adjustment is necessary for the Rayleigh scattering.

Putting all the above elements together we have, specific for calibrated crosstalk corrected attenuated backscatters

$$B_M(z) = \beta_M(z) e^{-2\tau(z)} \left[ (1 - f_e(z)) + f_{M,e}(z) e^{2\tau_\eta(z)} \right] \tag{B10}$$

$$B_R(z) = \beta_R(z) e^{-2\tau(z)} \left[ (1 - f_e(z)) + f_e(z) e^{2\tau_\eta(z)} \right] \tag{B11}$$

,

where $\tau$ and $\tau_\eta$ are given by

$$\tau(z) = \int_{z_{sat}}^{z} (\alpha_R(z') + \alpha_M(z')) dz' \tag{B12}$$

and

$$\tau_\eta(z) = \int_{z_{sat}}^{z} \eta(z') \alpha_M(z') dz'. \tag{B13}$$

A number of single-layer Monte-Carlo based simulations similar to that depicted in Fig. B4 for as range of ice cloud, water cloud and aerosol layers were conducted that indicated that specifying $\eta$ to be equal to 0.45–0.5 for both water and ice clouds, 0.375 for Dust and Sea salt and 0.1 for general aerosol types. In should be noted that, for small effective radii scatterers that $\eta$ is not very important for determining the signal-profiles as $f$ remains small.

It should be noted that in this work $\eta$ has been treated as being constant within a layer. Indeed, $\eta_p$ is often employed as being constant in practical situations e.g. (Garnier et al., 2015), however, according to Platt (1981), $\eta_p$ is treated as function of penetration depth and optical thickness. Our results indicate that by allowing for tails that $\eta$ can usefully be treated as being constant for a layer, since the use of $f_e(z)$ gives the system the necessary degree of freedom. In a heuristic sense, using a constant $\eta$ and a range dependent $f_e(z)$ is somewhat equivalent to making $\eta(z)$ range dependent.

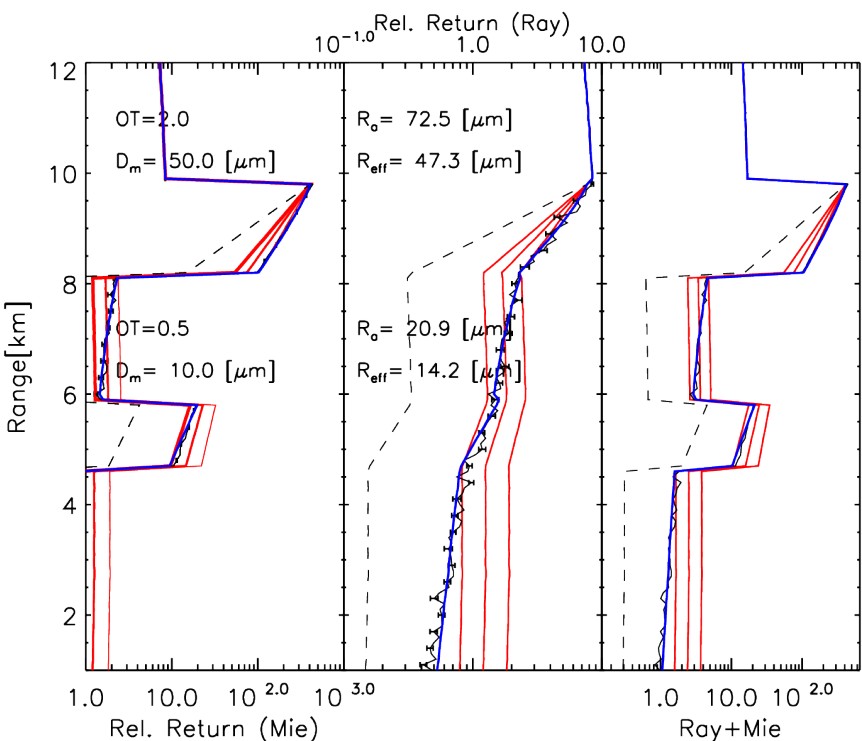

**Figure B4.** From Left to Right: Total simulated total attenuated backscatter, Rayleigh attenuated backscatter profile and the total simulated attenuated backscatter profile (difference scale than the first columns) corresponding to a two-layer ice cloud system with the given layer parameters. The black lines (with the error-bars) are the product of lidar Monte-Carlo radiative transfer calculations. The Dotted back lines are the single-scatter return values, the Red lines are the results of Platt's approach using tree different values of $\eta$ (0.4,0.5,0.6). The blue lines show the result of applying Eq. (B5) together with Eq. (B8) together with optimally chosen of values of $\eta$ and $\theta_{Sc}$ for each of the two layers.

The approach developed here, when applied to retrievals, is $\mathcal{O}(N^2)$ efficient (due to the need to evaluate Eq. B8) while the approach described in (Hogan and Battaglia, 2008) is $\mathcal{O}(N)$ efficient. The approach described here though has a 'lower-baseline' computational cost. Thus, roughly speaking, for typical case where the resolution is on the order for 100 m the two approaches have similar computational costs, however, for higher resolutions Hogans's PVC method becomes increasingly superior in this regard.

## B4 MS extinction and backscatter corrections

Using a forward-modeling approach based on Eq. B8, multiple-scattering effects, including the effects of tails can be accounted for. For *local-methods* employed to estimate the extinction, such as the approaches described in section A other correction type approaches are useful. In the following two sections the methods used to correct for multiple-scattering effects are described.

### B4.1 Extinction

First, focusing on the extinction, starting with Eq. B11 we can write

$$B_R^{Rat}(z) = \frac{B_M(z)}{\beta_R(z)} exp[2\tau_R] = e^{-2\tau(z)} \left[(1 - f_e(z)) + f_e(z) e^{2\tau_\eta(z)}\right] \tag{B14}$$

where

$$\tau_R(z) = \int_{z_{sat}}^{z} (\alpha_R(z')) dz'. \tag{B15}$$

Taking the log of Eq. B14 then yields

$$\log B_R^{Rat}(z) = -2\tau(z) + \log\left[(1 - f_e(z)) + f_e(z) e^{2\tau_\eta(z)}\right] \tag{B16}$$

then taking the derivative with range leads to

$$\frac{\log B_R^{Rat}}{dz}(z) = -2\alpha_M(z) + \frac{\left(e^{2\tau_\eta(z)} - 1\right) \frac{df_e}{dr}(z) + 2f_e(z)\eta(z)\alpha_M(z)e^{2\tau_\eta(z)}}{(1 - f_e(z)) + f_e(z) e^{2\tau_\eta(z)}} \tag{B17}$$

Now, if we define the *effective* extinction (i.e. the extinction that would be estimated assuming that no multiple scattering was occurring) as

$$\alpha_M^e(z) = -\frac{1}{2}\frac{\log B_R^{Rat}}{dr}(z) \tag{B18}$$

then

$$\alpha_M(z) = \alpha_M^e(z) + \frac{1}{2}\frac{\left(e^{2\tau_\eta(z)} - 1\right) \frac{df_e}{dr}(z) + 2f_e(z)\eta(z)\alpha_M(z)e^{2\tau_\eta(z)}}{(1 - f_e(z)) + f_e(z) e^{2\tau_\eta(z)}}. \tag{B19}$$

Eq. B19 can easily be solved iteratively as folows:

1. The $\eta$ and effective area radii $(R_a)$ profiles must first be specified.

2. The $f(z)$ profile (and it's derivative) is calculated by using Eq. B8.

3. The effective extinction is calculated using Eq. B18

4. $\tau_\eta$ is calculated using $\alpha_M^e(z)$ as a first guess for $\alpha_M$.

5. $\alpha_M$ is updated using Eq. B19.

6. $\tau_\eta$ is calculated using $\alpha_M$

7. Steps 5 and 6 are repeated until convergence (typically 2-3 times).

In order to estimate the error, for simplicity it is assumed that the uncertainty in $\alpha_M$ is dominated by the uncertainty in $\alpha_M^e$.

The application of Eq. B19 is illustrated by Figures B5 and B6. Here four cases of homogeneous layers were considered covering a range of conditions from low extinction ($\alpha_M = 0.1 \, \text{km}^{-1}$) to high extinction ($\alpha_M = 1.0 \, \text{km}^{-1}$) and small and larger particles sizes respectively. A satellite altitude of 400 km was assumed, the laser divergence and telescope FOV were set to 0.054 and 0.075 millirads (full-angle) respectivily.

Referring to Figs. B5 and B6, it can be seen that in all the cases considered, that the Platt+Tails approach can provide a good match to the the results generated by the approach of Hogan B8 if $\eta$ is set to a value of 0.425. Also it can be seen that the extinction estimates made using Eq. B19 are accurate to within under 10 % (here 3 iterations were used) while the estimates made when neglecting multiple-scattering or using Platt's approach are subject to much larger errors in general.

## B4.2   Backscatter

Now considering the backscatter. The observed(effective) backscatter can be related to the ratio of the Mie and Rayleigh attenuated backscatters i.e. using Eqns. B11 and B10,

$$\beta_M^e(z) = \beta_R(z)\left(\frac{B_M}{B_R}\right) \tag{B20}$$

$$= \beta_M \frac{\left[(1 - f_e(z)) + f_{M,e}(z)\,e^{2\tau_\eta(z)}\right]}{\left[(1 - f_e(z)) + f_e(z)\,e^{2\tau_\eta(z)}\right]}. \tag{B21}$$

When multiple scattering is not important (i.e. $f_e(z) = 0$ or $\tau_\eta(z) = 0$) then $\beta_M^e(z) = \beta_M(z)$. However, when multiple scattering is important (i.e. $f_e(z) = 1$ or $2\tau_{eta}(z) >> 1$)) then $\beta_M^e(z) = f_{M,e}(z)\beta_M(z)$. Once the extinction profile has been determined, as was described in the previous section, then the appropriate adjustment for multiple-scattering can be calculated directly using Eq. B21.

## B4.3   Sensitivity

The accuracy of the procedures described in Sections B4.1 and B4.2 is on the order of a few percent if the correct values of $R_a, \eta$ and $f_{MSp}$ are employed. However, this will not be the case in general. In order to assess the magnitude of the errors that may be expected due to the uncertainties in $Ra, \eta$ and $f_{MSp}$, the simple simulated cases shown in Fig. B5 and Fig. B6 were inverted and corrected for multiple-scattering effects using Eqns. (B19) and but using values of $Ra, \eta$ and $f_{MSp}$ different from the model-truth values.

Example of the impact of the particle size are shown in Fig. B8 and Fig. B9. Here it can be seen that halving the values of the effective area radius leads to an underestimation of the extinction by about 10-15% near the later top while doubling the effective area radius leads to an overestimation of the same rough magnitude. The corresponding errors in the lidar-ratio follow the same pattern with somewhat higher percentage values.

The relative errors resulting from errors in $Ra$ do not depend strongly on the extinction itself however, they can be strongly dependent on $R_a$. When $R_a$ is such that the associated $\theta_{sc}$ is much larger or smaller than $\rho_t$ then the impact of specifying $R_a$ is limited. This is illustrated by Fig. B9, where it can be seen that halving the value of $Ra$ used in the MS correction has a limited effect (less than 10% at the layer bottom) on the estimated extinction and lidar-ratio while doubling the value of $R_a$ used has practically no effect (since both 25 and $50\mu m$ produce a small value of $\theta_{sc}$ compared to $\rho_t$.

The values of $f_{MSp}$ used for the MS correction procedure do not impact the retrieved extinction, however, the retrieved backscatter (and hence the estimated lidar-ratio) will be impacted. In the limiting case where $f$ is close to 1, then the relative errors in the retrieved backscatter (and associated lidar-ratio) will directly correspond to the relative error in the value of $f_{MSp}$ used in the inversion.

The sensitivity of the corrections to the assumed value of $\eta$ were also investigated using trials values of 0.4 and 0.45. In all of the cases the impact was below 10% for both the extinction and lidar-ratio.

In summary, within layers, if the particle sizes care accurate within a factor of two, maximum errors in the extinction and lidar-ratio on the order of 10-15 % may be expected. The uncertainty in $f_{MSp}$ likely adds another 10-15% to the uncertainty in the lidar-ratio determination. For spherical scatterers, $f_{MSp}$ can be calculated using exact phase functions calculated using Mie theory (Hogan and Battaglia, 2008; Eloranta, 1998). For ice clouds, there is evidence for the general existence of a more pronounced backscatter-peak (which implies values of $f_{MSp}$ significantly less than 1) even for irregular crystals (Zhou and Yang, 2015).

The above conclusions are relevant for single-layer situations. In the case of e.g. semi-transparent cirrus above aerosols, the sensitivity to especially the particle size can be more significant. This aspect is discussed in Section 3 where specific cases drawn from the GEM-ECSIM test scenes are discussed.

## Appendix C:  A-EBD forward model and Jacobian

In this Section, the explicit form of the forward model used by the A-EBD optimal estimation retrieval is presented along with its Jacobian. Here $i$ is used to denote the index of the forward-model element and $n_z$ is the total number of range gates. Recall that the observation vector, and thus the forward-model, vector consists of the appended Rayleigh and Mie attenuated backscatters so that length of the forward model vector is of length $2n_z$.

For $i \leq n_z$ we have, based on a discrete form of Eq. B11,

$$F_i = B_R(z)e^{2\tau_R(z)} = B_{R,i}^c \tag{C1}$$

$$= \frac{C_{lid}}{\Delta r_i}e^{2\tau_{R,i}} \int_{r_{i,mid}-\Delta r/2}^{r_{i,mid}+\Delta r/2} \beta_R(z')e^{-2\tau(z')}\left[(1-f_e(z'))+f_e(z')e^{2\tau_\eta(z')}\right]dr(z')$$

where

$$\tau_{R,i} = \sum_{j=1}^{j=i-1} \alpha_{R,j}\Delta r_j, \tag{C2}$$

$$795 \quad \tau(z') = \sum_{j=1}^{j=i-1} (\alpha_{M,j} + \alpha_{R,j})\Delta r_j + (\alpha_{M,i} + \alpha_{R,i})(r(z) - r_{i-1}) \tag{C3}$$

$$\tau_\eta(z') = \sum_{j=1}^{j=i-1} (\eta_j \alpha_{M,j})\Delta r_j + (\eta_i \alpha_{M,i}(r(z') - r_{i-1})), \tag{C4}$$

and $f_e$ is the discrete version of Eq. B9.

For $i > n_z$ we have, based on a discrete form of Eq. B10,

$$
\begin{aligned}
F_i &= B_M(z)e^{2\tau_R(z)} = B_{M,i}^c \\
&= \frac{C_{lid}}{\Delta r_i}e^{2\tau_{R,i}} \int_{r_{i,mid}-\Delta r/2}^{r_{i,mid}+\Delta r/2} S^{-1}(z')\alpha_M(z')e^{-2\tau(z')} \left[ (1 - f_e(z')) + f_{M,e}(z')e^{2\tau_\eta(z')} \right] dr(z').
\end{aligned}
\tag{C5}
$$

Assuming that for each range-bin, that the Mie and Rayleigh extinctions, lidar-ratio, and $f$ terms can be treated as being
constant, evaluating the integral in Eq. C1 then yields, for the Rayleigh forward model,

$$B_{R,i}^c = C_{lid}\beta_{R,i} \left[ \Delta z_{c_1,i}(1 - f_{e,i}) + \Delta z_{c_2,i}f_{e,i}e^{2\tau_{\eta,i}} \right] \tag{C6}$$

and for the Mie forward model,

$$B_{M,i}^c = C_{lid}\frac{\alpha_{M,i}}{S_i} \left[ \Delta z_{c_1,i}(1 - f_{e,i}) + \Delta z_{c_2,i}f_{M,e,i}e^{2\tau_{\eta,i}} \right] \tag{C7}$$

where

$$810 \quad \Delta z_{c_1,i} = \frac{1 - \exp\left[-2(\alpha_{M,i} + \alpha_{R,i})\Delta r_i\right]}{2(\alpha_{M,i} + \alpha_{R,i})\Delta r_i} \tag{C8}$$

and

$$\Delta z_{c_2,i} = \frac{1 - \exp\left[-2(\alpha_{M,i}(1.0 - \eta_i) + \alpha_{R,i})\Delta r_i\right]}{2(\alpha_{M,i}(1.0 - \eta_i) + \alpha_{R,i})\Delta r_i} \tag{C9}$$

## C1   Gradient and Jacobian elements

In order to efficiently minimize the cost function, we must be able to compute its gradient with respect to the elements of the
state vector. The gradient of the cost-function is related to the Jacobian of the forward model as:

$$\nabla J = -2\mathbf{J_F}^T \mathbf{S_e}^{-1} (\mathbf{y} - \mathbf{F(x)}) + 2\mathbf{S_a}^{-1} (\mathbf{x_r} - \mathbf{x_a}), \tag{C10}$$

where $\mathbf{J_F}$ is the forward model Jacobian with respect to the state variables i.e.

$$
\begin{aligned}
J_{F_{i,j}} &= \frac{\partial F_i(\mathbf{x})}{\partial x_j} \\
&= \frac{\partial F_i(\mathbf{x})}{\partial \log_{10}(x_j')} = \frac{\partial F_i(\mathbf{x})}{\partial x_j'}\frac{\partial x_j'}{\partial \log_{10}(x_j')} = \frac{\partial F_i(\mathbf{x})}{\partial x_j'}\log_e(10)x_j'.
\end{aligned}
\tag{C11}
$$

where $x'_j$ refers to the linear counterpart of the element of the log state vector element $x_j$, i.e. $x_j = \log_{10}\left(x'_j\right)$.

### C1.1  Derivatives with respect to extinction.

Using the forward model the partial derivatives with respect to the particulate extinctions are:

for $j < i; i \leq n_z$

$$
\quad \frac{\partial F_i(\mathbf{x})}{\partial x'_j} = \frac{\partial B^c_{R,i}(\mathbf{x})}{\partial \alpha_j}
$$
$$
= 2C_{lid}\beta_{R,i}\left(\Delta z_{c_1,i}\exp\left[-2\tau_i\right](f_{e,i}-1)\Delta r_j + \Delta z_{c_2,i}f_{e,i}\exp\left[2(\tau_{\eta,i}-\tau_i)\right]\Delta r_j(\eta_j-1)\right). \tag{C12}
$$

For $i = j; i \leq n_z$

$$
\frac{\partial F_i(\mathbf{x})}{\partial x'_i} = \frac{\partial B^c_{R,i}(\mathbf{x})}{\partial \alpha_i}
$$
$$
= C_{lid}\beta_{R,i}\left(\frac{\partial \Delta z_{c_1,i}}{\partial \alpha_i}\exp\left[-2\tau_i\right](1-f_{e,i}) + \frac{\partial \Delta z_{c_2,i}}{\partial \alpha_i}f_{e,i}\exp\left[2\tau_{\eta,i}-2\tau_i\right]\right) \tag{C13}
$$

where

$$
\frac{\partial \Delta z_{c_1,i}}{\partial \alpha_i} = -\frac{\Delta z_{c_1,i} + \exp\left[-2(\alpha_{M,i}+\alpha_{R,i})\right]\Delta r_i}{\alpha_{M,i}+\alpha_{R,i}} \tag{C14}
$$

and

$$
\frac{\partial \Delta z_{c_2,i}}{\partial \alpha_i} = \frac{\left(2\exp\left[-2(\alpha_{M,i}+\alpha_{R,i})\right]\Delta r_i - 1\right)(1-\eta_i)\Delta r_i}{\left(2\left(\alpha_{M,i}(1-\eta_i)+\alpha_{R,i}\right)\Delta r_i\right)^2} + \frac{(1-\eta_{M,i})\exp\left[-2(\alpha_{M,i}+\alpha_{R,i})\right]\Delta r_i}{\alpha_{M,i}(1-\eta_i)+\alpha_{R,i}}. \tag{C15}
$$

For $j < i; n_z < k \leq 2n_z; k = n_z + i$

$$
\quad \frac{\partial F_k(\mathbf{x})}{\partial x'_j} = \frac{\partial B^c_{M,i}(\mathbf{x})}{\partial \alpha_j}
$$
$$
= 2C_{lid}\frac{\alpha_{M,i}}{S_i}\left(\Delta z_{c_1,i}\exp\left[-2\tau_i\right](f_{e,i}-1)\Delta r_j + \Delta z_{c_2,i}f_{M,e,i}\exp\left[2(\tau_{\eta,i}-\tau_i)\right]\Delta r_j(\eta_j-1)\right), \tag{C16}
$$

and For $j = i; n_z < k \leq 2n_z; k = n_z + i$

$$
\frac{\partial F_k(\mathbf{x})}{\partial x'_i} = \frac{\partial B^c_{M,i}(\mathbf{x})}{\partial \alpha_i}
$$
$$
= \frac{B^c_{M,i}}{\alpha_i} + C_{lid}\frac{\alpha_{M,i}}{S_i}\left(\frac{\partial \Delta z_{c_1,i}}{\partial \alpha_i}\exp\left[-2\tau_i\right](1-f_{e,i}) + f_{M,e,i}\frac{\partial \Delta z_{c_2,i}}{\partial \alpha_i}\exp\left[2(\tau_{\eta,i}-\tau_i)\right]\right). \tag{C17}
$$

**Derivatives with respect to lidar-ratio.**

For the lidar-ratio elements of the state-vector, one must take into account that the state-vector elements represent extended layers. Thus,

$$
\frac{\partial F_i(\mathbf{x})}{\partial x'_l} = \sum_{j=j_{b,l}}^{j_{t,l}} \frac{\partial B^c_{R,i}(\mathbf{x})}{\partial S_j} \tag{C18}
$$

where $n_z < l < n_z + n_l$ and $i \leq n_z$. $j_{t,j}$ is the range-index of the layer top for the layer corresponding to the $l$th element of the state-vector and $j_{b,j}$ is the range-index of the layer bottomfor the layer corresponding to the $l$th element of the state-vector.

For $n_z < i \leq 2n_z$

$$\frac{\partial F_i(\mathbf{x})}{\partial x'_l} = \sum_{j=j_{b,l}}^{j_{t,l}} \frac{\partial B^c_{M,i}(\mathbf{x})}{\partial S_j} \tag{C19}$$

For $j < i$

$$\frac{\partial B^r_{M,i}(\mathbf{x})}{\partial S_j} = 0 \tag{C20}$$

and for $i = j$

$$\frac{\partial B^r_{M,i}(\mathbf{x})}{\partial S_i} = -\frac{B^r_{M,i}}{S_i}. \tag{C21}$$

### C1.2 Derivatives with respect to particle-size.

For the $R_a$ elements of the state-vector, as is the case for thee lidar-ratio, one must take into account that the state-vector elements represent extended layers. Thus,

$$\frac{\partial F_i(\mathbf{x})}{\partial x'_l} = \sum_{j=j_{b,l}}^{j_{t,l}} \frac{\partial B^c_{R,i}(\mathbf{x})}{\partial \theta_j} \frac{\partial \theta_j}{\partial R_{a,j}} \tag{C22}$$

where $2n_z < l < n_z + 3n_l$ and $i \leq n_z$ and

$$\frac{\partial \theta_j}{\partial R_{a,j}} = -\frac{\lambda}{\pi R^2_{a,j}}. \tag{C23}$$

For $j \leq i$

$$\frac{\partial B^c_{R,i}(\mathbf{x})}{\partial \theta_j} = C_{lid}\beta_{R,i}\left(\Delta z_{c_2,i}\exp\left[2(\tau_{\eta,i} - \tau_i)\right] - \Delta z_{c_1,i}\exp\left[-2\tau_i\right]\right)\frac{\partial f_{e,i}}{\partial \theta_j}, \tag{C24}$$

where

$$\frac{\partial f_{e,i}}{\partial \theta_j} = \left[-2\left(B_{M,j} + B_{R,j}\right)\exp\left[-\frac{(r_i\rho_t)^2}{(r_i\rho_l)^2 + (\theta_j^2(r_i - r_j)^2}\right]\Delta r_j\left[\frac{(r_i\rho_t)^2}{\left[(r_i\rho_l)^2 + \theta_j^2(r_i - r_j)^2\right]^2}\right]\theta_j(r_i - r_j)^2\right]$$

$$\times\left[\sum_{k=0}^{i}\left(B_{M,k} + B_{R,k}\right)\Delta r_k\right]^{-1}. \tag{C25}$$

For $n_z < i \leq 2n_z$,

$$\frac{\partial F_i(\mathbf{x})}{\partial x'_l} = \sum_{j=j_{b,l}}^{j_{t,l}} \frac{\partial B^c_{M,i}(\mathbf{x})}{\partial \theta_j} \frac{\partial \theta_j}{\partial R_{a,j}} \tag{C26}$$

For $j \leq n_z$

$$\frac{\partial B_{M,i}^c(\mathbf{x})}{\partial \theta_j} = C_{lid}\alpha_{M,i}S_i^{-1}\left(\Delta z_{c_2,i}\exp\left[2(\tau_{\eta,i}-\tau_i)\right]-\Delta z_{c_1,i}\exp\left[-2\tau_i\right]\right)\frac{\partial f_{M,e,i}}{\partial \theta_j}. \tag{C27}$$

### C1.3   Derivatives with respect to $C_{lid}$.

For $j = n_z + 2n_l + 1$ and $i \leq n_z$

$$\frac{\partial F_i(\mathbf{x})}{\partial x_j'} = \frac{\partial B_{R,i}^c(\mathbf{x})}{\partial C_{lid}} = \frac{B_{R,i}^c}{C_{lid}} \tag{C28}$$

and for $n_z < i \leq 2n_z$

$$\frac{\partial F_i(\mathbf{x})}{\partial x_j'} = \frac{\partial B_{M,i}^c(\mathbf{x})}{\partial C_{lid}} = \frac{B_{R,i}^c}{C_{lid}} \tag{C29}$$

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

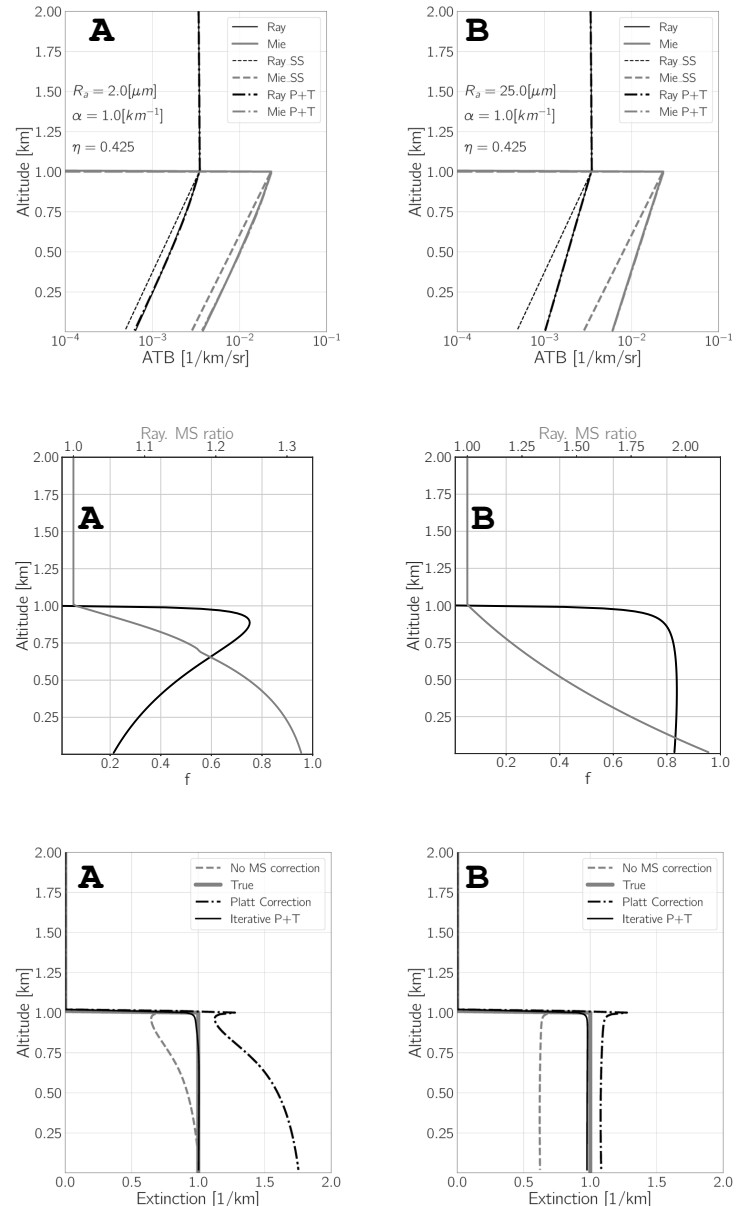

**Figure B5.** Results of idealized simulations using homogeneous layers. The top-panels show the Rayleigh and Mie attenuated backscatters assuming single-scattering ('Ray_SS' and 'Mie_SS' respectively) as well as the profiles generated using the approach due to Hogan ('Ray' and 'Mie' respectively) and the 'P+T' approach with $\eta = 0.425$. The middle panels show the $f$ profiles generated using B8 and the ratio between the Rayleigh ATBs including multiple-scattering and the single-scattering ATBs. The bottom-left panels show the model-truth extinction profile, the retrieved extinction profiles assuming no multiple-scattering effects, the extinction values that would be estimated using Platt's approach and the results of iteratively applying Eq. B19. The 'A' panels correspond to the case where $R_a = 2[\mu m]$ while the 'B' panels correspond to the case where $R_a = 25[\mu m]$

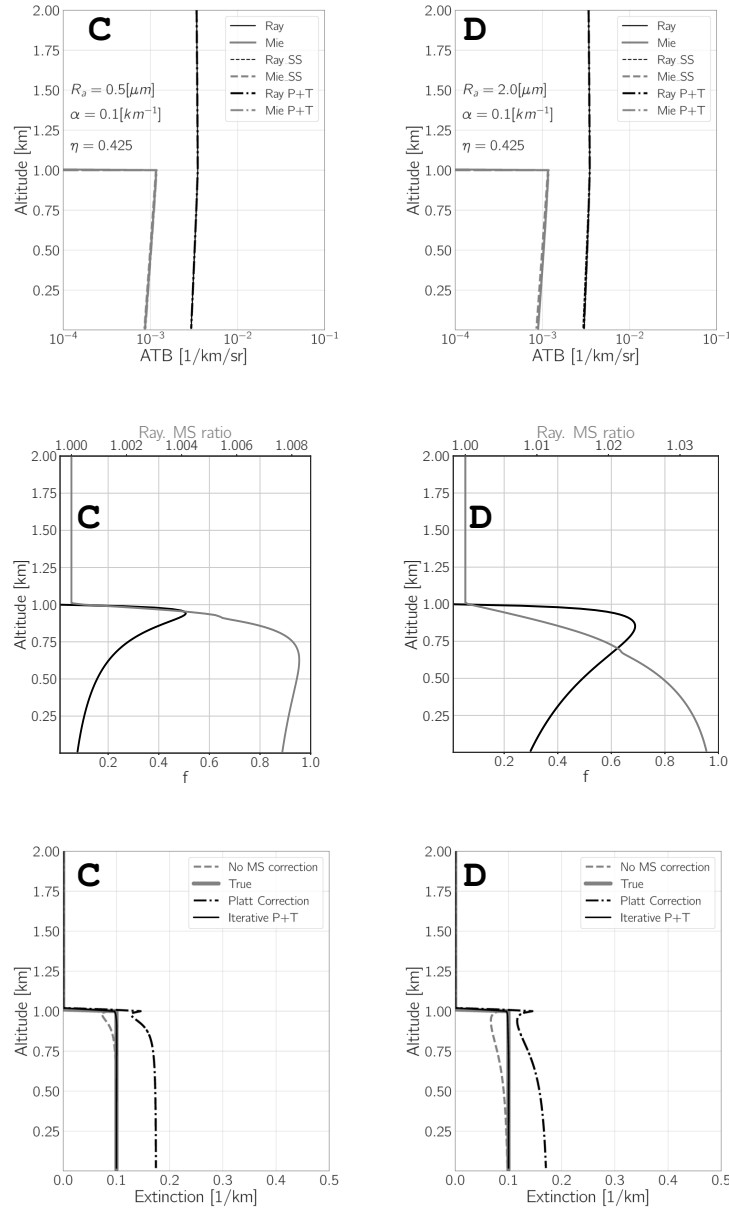

**Figure B6.** As Fig. B5 but for an extinction coefficient of 0.1 km$^{-1}$ and $Ra = 0.5$ and $2[\mu m]$ respectively.

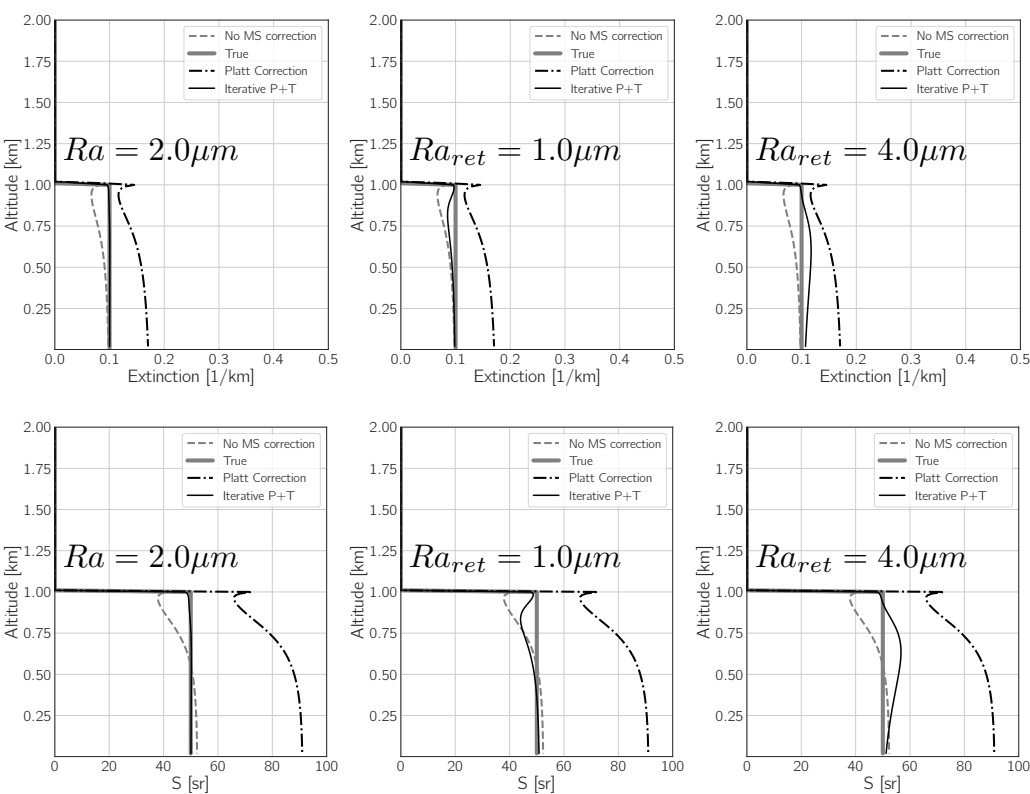

**Figure B7.** Top-panels extinction retrieved corresponding to the Case-D in Fig. B6 using the indicated values of $Ra$. The corresponding retrieved lidar-ratios are shown in the Bottom-panels.

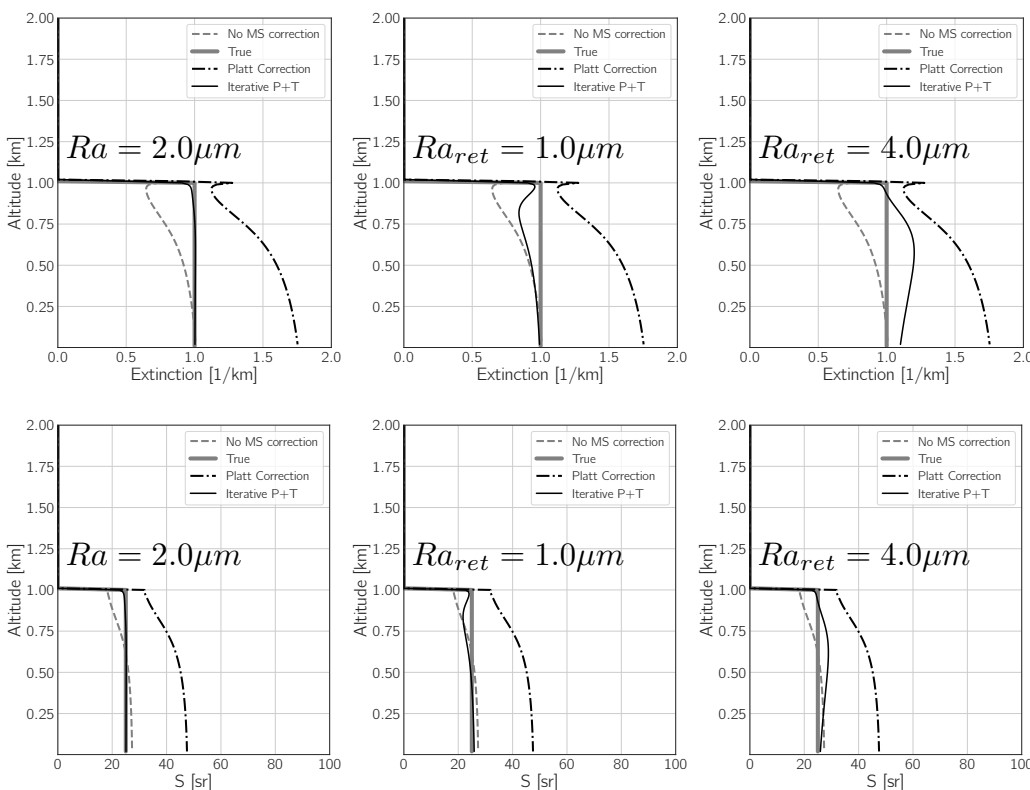

**Figure B8.** Top-panels extinction retrieved corresponding to the Case-A in Fig. B6 using the indicated values of $R_a$. The corresponding retrieved lidar-ratios are shown in the Bottom-panels.

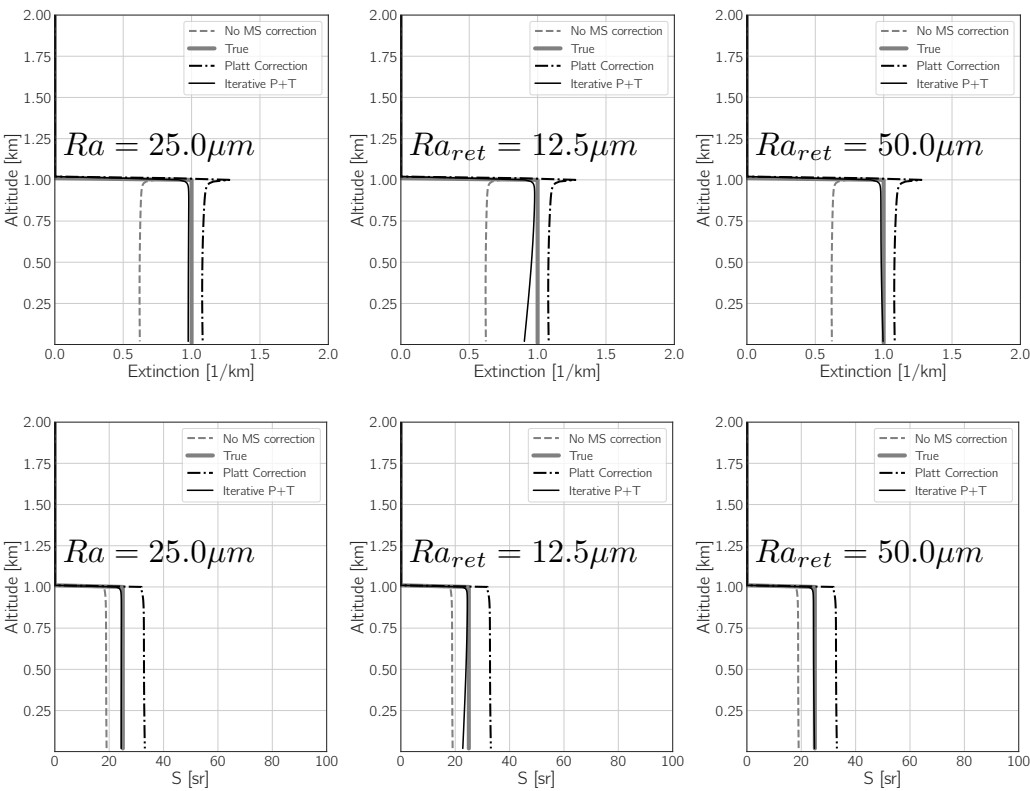

**Figure B9.** Top-panels extinction retrieved corresponding to the Case-B in Fig. B6 using the indicated values of $R_a$. The corresponding retrieved lidar-ratios are shown in the Bottom-panels.