# Peer review of "The EarthCARE lidar cloud and aerosol profile processor (A-PRO): the A-AER, A-EBD, A-TC and A-ICE products"

_EGUsphere, 2024_

## Editor Comment (EC1)

Review of "The EarthCARE lidar cloud and aerosol profile processor (A-PRO)", Donovan, van Zadelhoff and Wang

General Comments

This manuscript describes retrieval algorithms which will be used to process data from the ATLID lidar, to be launched soon on the EarthCARE satellite. Thus this is an important paper which will draw considerable interest. The manuscript gives a good summary of how the retrievals are performed but is unclear in places. There is some necessary context that is missing and some discussion of why key algorithm design choices were made would be helpful.

This paper is one of a large set of EarthCARE algorithm papers and the algorithm described here uses results from several algorithms described in other papers. The current manuscript seems to assume the reader is familiar with these other algorithms. To avoid the need to refer to other papers to understand the A-PRO processor the authors should add brief descriptions of the other algorithms and the parameters which are used as inputs to A-PRO. For example, the paper needs a paragraph or so describing the A-TC algorithm and the classification information it provides to the A-PRO processor, perhaps in the form of a table listing the target categories.

To make it easier for the reader to understand the three ATLID signals used in these retrievals, the authors should include a schematic of the optical layout of ATLID, which isn't described well by the layout in do Carmo et al. A lidar expert can derive the optical layout from Eqns 1-3, but this requires expertise and some effort. Including a layout schematic would avoid confusion and mis-interpretation.

Retrievals are performed at different horizontal resolutions. It's not clear to me how the signal averaging and retrievals are related or how results are reported in the data products. In CALIOP data processing, layers in a given scene have likely been detected at different resolutions, following a detect-remove-reaverage approach. It looks like ATLID takes the approach of averaging the entire scene at a uniform resolution, except that certain features are excluded from averaging. Some discussion of how the averaging scheme is similar or different from the one used for CALIOP would be helpful.

Retrievals are performed at 1-km and 50km scales. What was the thinking behind two, and only two, different scales vs. the CALIOP approach which uses five different averaging scales?

It is not entirely clear to me how smearing of aerosol and cirrus together is avoided in this multi-resolution scheme. Dense cloud is easily separated from aerosol (using information from A-TC?) but aerosol and cirrus can have similar scattering strengths, and sometimes even similar volume depolarization. A little description of the information coming from A-TC would be helpful – a table listing the different feature categories? Are A-TC classifications provided at the resolution of the JSG grid?

I was confused by the choice of math symbols in Sections 2.2-2.3. They are different from what I am used to and involve a variety of confusing sub- and super-scripts. Adding a table defining each symbol would be helpful.

Several times it is mentioned that an algorithm parameter is configurable or can be set to one of two or more options. Will one of these options be selected for all operational processing, or is one option or another selected by the operational processor depending on which is better for a given scene?

Specific comments

Line 43 – Please cite the papers which describe calibrations and cross-talk corrections. Maybe add a few sentences on the general approaches for calibration and cross-talk corrections?

Parts of Section 2 read like a bulleted list rather than a narrative. Some additional context is needed to understand the details provided. See next comments.

Line 83 – Describe the JSG a little bit – why is there a JSG? Explain why the L1 ATBs need to be rebinned to the JSG grid. What is the resolution of the JSG?

Lines 85-90 – It's not clear if cloud phase and aerosol type information are determined in A-FM or in A-PRO. Add a table listing the feature mask classifications provided in A-FM?

Line 125-126 – Expand a little on what the A-FM feature probability indices are. I can guess what is going on here, but the reader shouldn't be forced to go to another paper to understand this.

Line 129 – How is the "strong feature" mask created by thresholding? A little more detail please. Is a constant threshold used, is it adapted depending on signal SNR?

Line 134 – The threshold is set to "8" – 8 what? Is this a 355 nm scattering ratio threshold?

Line 141 – It is not clear what is meant by "layering structure" and how this is determined. Is this just feature detection or does it include identifying cloud and aerosol, or determining composition differences? What makes this preliminary?

Line 141-142 – Incomplete sentence (The scattering ratio calculations performed … )

Line 145 and following – It would be nice to have some detail and a figure describing Step 3

Line 151 – Does "e.g. 5" refer to a scattering ratio? If so, this should be made clear, and remind the reader that this is a scattering ratio at 355 nm as some of us think of 532 nm by default.

Lines 152-160 – Understanding what is going on here involves a lot of guesswork. A figure might help.

Line 185 – Is there a definition of fine and coarse layers? It is not clear what the difference is, or why this is done in two steps.

Line 201 – Explain what the 'classification priors' are and where they come from. I think this is the first mention.

Starting around line 243 – does 'a priori errors' refer only to random errors or both random and bias errors?

Lines 225-226 – Is there a subscript missing? Eqn 9 has $x^l_a$ and $x^l_r$. $x^l_a$ is discussed in the following text but there is no discussion of $x^l_r$, only of $x^l$.

Line 243-250 – I don't quite understand how errors are treated in the O-E retrieval. In A-EBD, errors are assumed to be uncorrelated. A constant value of lidar ratio and particle size is retrieved for each layer, but there is still an error which may be different in every range bin. Wouldn't bias errors be highly correlated within an aerosol or cloud layer? Is this just ignored? A little more discussion would be helpful.

Line 288 – The algorithm seems to estimate both IWC and effective ice particle size using one of two parameterizations from Heymsfield et al. The particle size from this approach is just a parameterized climatological average based on temperature and should not be thought of as a retrieval. This should be made clear to data users.

Line 289 – Why are there two options for estimating IWC? Are both options available to the data user or will a final selection made by the algorithm developers after launch. Please explain.

Line 353: Last paragraph of Section 3.1 – Many people are familiar with CALIOP, which uses the simpler Platt method for correction of multiple scattering effects from cirrus. This is mentioned (very briefly) in Appendix B, but it would be helpful to point out here that the small footprint of ATLID enhances the multiple scattering tail effects relative to CALIOP, which has a larger footprint. From the discussion here, it appears to be much easier to characterize errors due to incorrect MS correction for HSRL than for a backscatter lidar.

Color bars: Some of the color bars used in the figures should be improved. Copernicus journals are now making color-vision deficiency (CVD) accessibility a priority. The color bars used in Fig 7 and 8 are probably good for CVD. The rainbow color bars used in Figs 5 and 6, and others, mix red and green and may be difficult or impossible for those with non-standard color vision to interpret.

---

## Author Comment (AC4)

Reply to (late) Reviewer 3 D. Winker.

Reviewer comments are in Black while our response is in Blue.

General Comments

This manuscript describes retrieval algorithms which will be used to process data from the ATLID lidar,to be launched soon on the EarthCARE satellite. Thus this is an important paper which will draw considerable interest. The manuscript gives a good summary of how the retrievals are performed but is unclear in places. There is some necessary context that is missing and some discussion of why key algorithm design choices were made would be helpful.

Noted: We will try to add more context and the rational behind our choices.

This paper is one of a large set of EarthCARE algorithm papers and the algorithm described here uses results from several algorithms described in other papers. The current manuscript seems to assume the reader is familiar with these other algorithms. To avoid the need to refer to other papers to understand the A-PRO processor the authors should add brief descriptions of the other algorithms and the parameters which are used as inputs to A-PRO. For example, the paper needs a paragraph or so describing the A-TC algorithm and the classification information it provides to the A-PRO processor, perhaps in the form of a table listing the target categories

Perhaps the balance between not repeating what is in the other papers and making the paper more self-contained indeed needs to be adjusted. More information about A-TC will be included and how it interacts with the other A-PRO components will be added.

To make it easier for the reader to understand the three ATLID signals used in these retrievals, the authors should include a schematic of the optical layout of ATLID, which isn't described well by the layout in do Carmo et al. A lidar expert can derive the optical layout from Eqns 1-3, but this requires expertise and some effort. Including a layout schematic would avoid confusion and mis-interpretation.

Noted: A simple sketch and discussion will be added.

Retrievals are performed at different horizontal resolutions. It's not clear to me how the signal averaging and retrievals are related or how results are reported in the data products. In CALIOP data processing, layers in a given scene have likely been detected at different resolutions, following a detect- remove-reaverage approach. It looks like ATLID takes the approach of averaging the entire scene at a uniform resolution, except that certain features are excluded from averaging. Some discussion of how the averaging scheme is similar or different from the one used for CALIOP would be helpful.

The averaging strategies are described in quite some detail within the algorithm flowcharts and the associated discussion. However, we can see how the presentation is perhaps too detailed, such that the reader can miss the forest-for-the-trees (so to speak). A higher-level more accessible discussion addressing the points the reviewer raised will be included.

Retrievals are performed at 1-km and 50km scales. What was the thinking behind two, and only two,

different scales vs. the CALIOP approach which uses five different averaging scales?

A-AER (the direct HSRL based algorithm) retrievals are applied at the 1-km scale but use input signals that have been "strong-feature" screened and smoothed to between 10 and 100 km or so. A-EBD (the optimal estimation based retrieval) always uses unsmoothed 1-km resolution signals as input. High, Medium, and Low-resolution EBD output products are provided. Nominally, this corresponds to 1-km, 50 km and 100 km. They are constructed by averaging the "strong-feature" screened 1-km EBD outputs and then superimposing the "strong-feature" results. We will attempt to clarify this in the manuscript.

The feature detection algorithm A-FM, in a sense, fills the role played by the multi-scale approach used by CALIPSO. But in a "continuous" rather than "discrete" fashion. The role of A-FM and how it is used by the A-PRO processor will be discussed more in the text.

It is not entirely clear to me how smearing of aerosol and cirrus together is avoided in this multi-resolution scheme. Dense cloud is easily separated from aerosol (using information from A-TC?) but aerosol and cirrus can have similar scattering strengths, and sometimes even similar volume depolarization. A little description of the information coming from A-TC would be helpful – a table listing the different feature categories? Are A-TC classifications provided at the resolution of the JSG grid?

A-TC classifications are indeed provided at the resolution of the JSG grid. Aerosol and thin cirrus separation will be challenging and likely can not be entirely avoided. Ultimately, If the signals strengths are similar, the depol ratios similar, and the lidar-ratios are similar, then the lidar-only approach is problematic.

I was confused by the choice of math symbols in Sections 2.2-2.3. They are different from what I am used to and involve a variety of confusing sub- and super-scripts. Adding a table defining each symbol would be helpful.

Noted. We will consider adding a table.

Several times it is mentioned that an algorithm parameter is configurable or can be set to one of two or more options. Will one of these options be selected for all operational processing, or is one option or another selected by the operational processor depending on which is better for a given scene?

For operational processing the configuration options will be fixed. For off-line studies different configuration options will be investigated. Operational values will only be fixed during the course of the commissioning phase.

Specific Comments

Line 43 – Please cite the papers which describe calibrations and cross-talk corrections. Maybe add a few sentences on the general approaches for calibration and cross-talk corrections?

Noted. More information will be added.

Parts of Section 2 read like a bulleted list rather than a narrative. Some additional context is needed to understand the details provided. See next comments.

Noted.

Line 83 – Describe the JSG a little bit – why is there a JSG? Explain why the L1 ATBs need to be rebinned to the JSG grid. What is the resolution of the JSG?

The JSG exists to facilitate ``synergy'' with the other EarthCARE instruments. A common approach is used for co-lcating the lidar,radar and nadir msi pixels. This is done via the JSG. The JSG has a nominal resolution of 1-km along track.

Lines 85-90 – It's not clear if cloud phase and aerosol type information are determined in A-FM or in A-PRO. Add a table listing the feature mask classifications provided in A-FM?

A-FM only provides a target mask, not a classification ! Phase, aerosol type etc is done with A-PRO (using A-FM as one of the inputs). This will be clarified within the text.

Line 125-126 – Expand a little on what the A-FM feature probability indices are. I can guess what is going on here, but the reader shouldn't be forced to go to another paper to understand this.

Noted. More information will be provided in the manuscript.

Line 129 – How is the "strong feature" mask created by thresholding? A little more detail please. Is a constant threshold used, is it adapted depending on signal SNR?

Noted. More information will be provided in the manuscript.

Line 134 – The threshold is set to "8" – 8 what? Is this a 355 nm scattering ratio threshold?

No. It is the A-FM target probability index. More explanation will be added.

Line 141 – It is not clear what is meant by "layering structure" and how this is determined. Is this just feature detection or does it include identifying cloud and aerosol, or determining composition differences? What makes this preliminary?

This is just feature detection. More explanation will be added.

Line 141-142 – Incomplete sentence (The scattering ratio calculations performed … )

Noted

This text should read…"….is determined using the scattering ratio…"

Line 145 and following – It would be nice to have some detail and a figure describing Step 3

We will consider this.

Line 151 – Does "e.g. 5" refer to a scattering ratio? If so, this should be made clear, and remind the reader that this is a scattering ratio at 355 nm as some of us think of 532 nm by default.

Yes. This will be made clearer in the text.

Lines 152-160 – Understanding what is going on here involves a lot of guesswork. A figure might help

We will consider this.

Line 185 – Is there a definition of fine and coarse layers? It is not clear what the difference is, or why this is done in two steps.

Fine layers have a higher vertical resolution than the coarse layers and are determined using more information (which is available at this point in the algorithm flow e.g. the lidar-ratio). The fine-layer structure is important input to the optimal estimation process. More explanation will be added in the text.

Line 201 – Explain what the 'classification priors' are and where they come from. I think this is the first mention

Noted. This is explained in the A-TC paper but more info will be added here.

Starting around line 243 – does 'a priori errors' refer only to random errors or both random and bias errors?

Random and bias. This will be made clearer in the text.

Lines 225-226 – Is there a subscript missing? Eqn 9 has $x^l\_a$ and $x^l\_r$. $x^l\_a$ is discussed in the following text but there is no discussion of $x^l\_r$, only of $x^l$.

Noted: There are several typos to be fixed in this section. The use of $x^l$ will be eliminated (see our response to point 4 of Reviewer 2).

Line 243-250 – I don't quite understand how errors are treated in the O-E retrieval. In A-EBD, errors are assumed to be uncorrelated. A constant value of lidar ratio and particle size is retrieved for each layer, but there is still an error which may be different in every range bin. Wouldn't bias errors be highly correlated within an aerosol or cloud layer? Is this just ignored? A little more discussion would be helpful.

For the lidar-ratio and particle size A-EBD operates on a per (fine) layer-by-layer basis. So only the correlations between different layer averages are important. Moreover, the lidar-ratio and Reff are layer averages, no sub-fine-layer information is supplied in the output product (so it does not make sense to talk about bias errors within a fine-layer) this is one of the reasons the fine layer structure is important. This point however, should be made clearer to potential end-users of the data.

Line 288 – The algorithm seems to estimate both IWC and effective ice particle size using one of two parameterizations from Heymsfield et al. The particle size from this approach is just a parameterized climatological average based on temperature and should not be thought of as a retrieval. This should be made clear to data users.

Noted. This will be made clear.

Line 289 – Why are there two options for estimating IWC? Are both options available to the data user or will a final selection made by the algorithm developers after launch. Please explain

A final selection will be made during after launch. The data users can always make their own IWC estimates using the retrieved extinctions (after all the IWC product is only a parametrization). However, A-PRO used the Re estimates in the MS correction procedures.  So the decision will likely hand on this aspect.

Line 353: Last paragraph of Section 3.1 – Many people are familiar with CALIOP, which uses the simpler Platt method for correction of multiple scattering effects from cirrus. This is mentioned (very briefly) in Appendix B, but it would be helpful to point out here that the small footprint of ATLID enhances the multiple scattering tail effects relative to CALIOP, which has a larger footprint. From the discussion here, it appears to be much easier to characterize errors due to incorrect MS correction for HSRL than for a backscatter lidar.

Noted. The suggestion will be adopted.

Color bars: Some of the color bars used in the figures should be improved. Copernicus journals are now making color-vision deficiency (CVD) accessibility a priority. The color bars used in Fig 7 and 8 are probably good for CVD. The rainbow color bars used in Figs 5 and 6, and others, mix red and green and may be difficult or impossible for those with non-standard color vision to interpret.

Noted.

---

## Author Response (AR1)

EGUSPHERE-2024-218 | Research article

Submitted on 24 Jan 2024

**The EarthCARE lidar cloud and aerosol profile processor (A-PRO): the A-AER, A-EBD, A-TC and A-ICE products**

David Patrick Donovan, Gerd-Jan van Zadelhoff, and Ping Wang

Special issue: EarthCARE Level 2 algorithms and data products

Handling editor: Edward Nowottnick, edward.p.nowottnick@nasa.gov

In the following pages please find our response to the two reviewers. Changes were also made in response to public comments made by D. Winker but are not detailed here.

**Reply to Reviewer 1:**

Reviewer comments are in Black while our response is in Blue.

Page 2 Line 27: To call all particulate scattering Mie scattering is rather unfortune as also stated a bit further down in the text. It is recognized to reworking the paper to use particulate consistently is a lengthy and tedious task but non-the-less strongly recommended!!!

This (unfortunate) informal convention is well entrenched within the Aeolus/ATLID ESA lidar community and used within all the other relevant papers within the special issue. We have made the `disclaimer' more prominent in the text (see around line 50 in the DIFF pdf) but keep using both 'Mie' and `Particulate'.

Page 5 Line 122: ATB is not defined before in the text. One could guess that it would probably mean Attenuated Backscatter. But the definition should be made at the first occurrence of the acronym.

See line 34 in the DIFF pdf.

Page 5 Line 125: "Rather, the highest index within…". This refers to the feature mask definition, but to help the reader, the meanings of the numerical values should be given here it should be noted that the feature mask values are ordered in ascending order with potential scattering strength.

The feature-mask has now been explained in more detail (see lines 130-150 in the DIFF pdf).

Page 6 Line 134: "… are set to > 8 …" Please translate this to verbal feature type!

See line 223 in the DIFF pdf.

Page 6 Line 141: "The scattering ratio calculations…" This seems to be only a half-sentence with unclear meaning.

This text should read…"….is determined using the scattering ratio…". See text around line 234 in the DIIF pdf.

Page 6 Lines 150-151: Again, feature mask entries should be given, at least additionally, in plane words.

See revised discussion (Sections 2.2.1 and 2.2.2)

Page 7 Line 165-166: "provides also" or "enables", one should be sufficient.

This typo has been fixed (see line 258 in the DIFF pdf).

Page 7 Line185ff: The term "layer" here lets one think of straight horizontal layers, but in reality, the aerosol structure is often not stratiform but modulated for example by gravity waves with amplitudes of +-500 m or more. And even without gravity waves the layers are typically more wedge shaped, instead of being horizontal bands. Maybe the algorithm can cope with such cases, but this is not clear from the text. Please add more information on this.

The coarse and fine layering determination is done on the 1-km horizontal scale (although the signals used at this stage have been smoothed to lower horizontal resolutions (a few km)). So, in general, structures present at finer resolutions will be blurred out.

More information about how the layers are determine has been added to the manuscript (See text around lines 300-305 in the DIFF PDF and the new Fig.2)

Page 8 Line 198: Step 16 seems to be missing. Please add a note that this is on purpose – if it is…

Noted. Step 16 is indeed missing from the description and Fig.2. Note, since the initial submission changes have been made in the algorithm which will necessitate an update of this section and Fig.2 in any case (a reclassification is now performed following the MS correction procedure (step 17)). See revised Fig 4 and the associated text).

Page 9 Line 227: Using the log-form a quantity that naturally can come close to zero, like the extinction coefficient, can be problematic numerically. This introduces a pole at 0 which easily misguides the optimisation routine. Further it skews the nearly Gaussian input error statistics heavily making the minimisation of Eq. (9) not an optimal estimate any more. And the skewed error statistics leads to noise induced biases. If the whole reason for the log-form is to keep positiveness than there are methods to do constrained optimal estimates which are more robust. It is clear to the reviewer that the algorithm is, as it is at the moment and will not be changed in short term. But a few sentences more about the choice and possible alternatives should be added.

Due to some mangled editing, there are some typos in the description and equations this may have caused some confusion (see also our response 6 to Reviewer 2). Just to be clear, our state-vector is logarithm BUT the observables (y and F(x)) are linear (along with their uncertainty). Using the log form for the observables, we feel could indeed be problematic.

As for the use of log-forms in the state vector. The reviewer claims that `` it skews the nearly Gaussian input error statistics heavily making the minimisation of Eq. (9) not an optimal estimate any more''.  This is true if the a priori distribution of the state vector element is Gaussian. IF the variable is better described by a log-normal distribution, then the use of the log form (with the appropriate form of the uncertainties) is appropriate (see Q. J. R. Meteorol. Soc. 142: 274–286, January 2016 A DOI:10.1002/qj.2651).  Here, the primary use of the log form is because e.g. the lidar-ratio distribution is thought to be better described by a log-normal rather than normal form (see e.g. Fig 6 of Int. J. Environ. Res. Public Health 2016, 13, 508; doi:10.3390/ijerph13050508).

In our experience, the use of the log-form for the state-vector does not present any particular challenges for the minimization. In fact, it has been shown to have benefits in the quality of the retrievals (see e.g. Maahn, M., D. D. Turner, U. Löhnert, D. J. Posselt, K. Ebell, G. G. Mace, and J. M. Comstock, 2020: Optimal Estimation Retrievals and Their Uncertainties: What Every Atmospheric Scientist Should Know. Bull. Amer. Meteor. Soc., 101, E1512–E1523, https://doi.org/10.1175/BAMS-D-19-0027.1.)

See the lines 330-365 in the DIFF pdf.

Page 10 Line 242 and 246: "…the effective radii are specified a priory by type." And "… is the a priori (linear)uncertainty assigned…" To what value? And why this value (Ref)? Maybe a table which collects all the a priory values would be appropriate.

The priors used in this work are somewhat tuned to the simulated data sets being used for the testing and may be different from those that will be used operationally. Such a Table will be published along with the first "real" results though ! It will be pointed out, however, that for interested parties that the a priori info (along with all the algorithm settings) are accessible via the algorithm configuration files contained in the supplementary data package.

Page 10 Line 244: "It is assumed that the a prior errors are uncorrelated…" While for the observational errors it makes perfect sense to assume that different vertical bins are uncorrelated (at least for the raw signals without some averaging), this is not obvious at all for the a priory values. The atmosphere is not "white noise" and the possible deviations of a first guess from the true value no less. It is clear that it is hard to come up with sensible values just starting from scratch, and it may have serious impacts on the algorithm used for the optimisation, but this point should be discussed in more detail and a route to future improvements should be outlined.

We agree that the assumption that the priors are not correlated is likely inaccurate and merely an expedient. To improve upon this, one may envision a "boot starting" process using real observations.

See lines 375-380 in the DIFF pdf.

 Page 13 Line 345: Above or between???

This typo has been fixed

Page 20/21/31: The colour scales are rather unfortunate. It is practically not possible to distinguish vales in S between 0 and 50. The one chosen for Fig. 9 is much better. It would be good anyway to harmonize the colour scales of the different 2-d charts.

The colour scales have been adjusted.

Page 34 Line 439: use? Or find?

Will be fixed.

Page 36 Line 490: "Fig ?" Reference is missing.

Fixed.

Page 37 Figure A1 caption: "… log-derivative approach (Grey-solid-line) and …" this should be Black-solid-line, shouldn't it? It would help the reader greatly if line legends could be added to the figures, like e.g. in Fig B5-B9!

Black-solid line is correct. Line legends have been added (see page 46 in the DIFF pdf).

Page 38 Figure B1: Here also in-plot legends would be beneficial! And the plotting style very much reminds of the pen-plotters used until the early 80s of the last centuries. The authors are strongly advised to use a homogeneous and up-to-date style for their plots!

Captions have been added. The reviewer's implications that these plots look rather "old school" is taken as a compliment.

Page 40 Line 546: "… the divergence of the forward scattered light will also be Gaussian with a divergence…" This is a simplifying assumption since the foreword scattering peak is not exactly a Gaussian! Typically, this assumption is good enough, but the phrasing here suggest that this is a mathematical truth, which is not the case.

Noted:  See line 699 of the DIFF pdf.

Page 41 Eq. B8: "H(Theta_sc(z)  >  0)" It is obvious what the authors try to express here, but formally this makes no sense. H is defined for the real number, but Theta_sc(z)  >  0 is a comparison with a Boolean result. Even if one assigns 0 for false and 1 for true this does not work as for the common definition of H, H(0) = 1 and H(x) = 0 only for strictly negative x.

Noted: This section has been rewritten to remove the use of H. See text around line 715 in the DIFF pdf.

Page 43: Again, horrible figure style. See comment above.

See our earlier response (3 comments above).

Page 45 Line 628: "calculate" -> calculated

Fixed.

Page 45 Line 649: Here and from the preceding text one could get the impression that Platts eta is some sort of system constant which can be "calibrated" using some higher accuracy MS-algorithms like e.g. Monte-Carlo simulations (as done for B4). But according to Platts papers neta also depends on penetration depth and optical thickness of the cloud and the variation are much larger then +- 10%. Maybe the introduction of the additional tail-function f_e takes over some of the effects of eta and stabilises its value. This should be discussed in more detail.

This is a useful observation. As it is often employed in a practical sense (e.g. CALIPSO) Platt's eta is treated as constant, but if one consults Platt's papers then one does see he considered it as a function of e.g. OT. The work presented here indeed shows that eta can be treated as a constant since, indeed,  f_e, gives the system the necessary degrees of freedom. In a sense, eta_platt(z) is (approximately) equivalent to (eta_constant,f_e(z)). One could mathematically develop this idea. This will be considered for the revision.

See text around line 745 in the DIFF pdf.

Appendix: The reviewer had no time check that the equations for the partial derivatives of the foreword model are correct. He only hopes that they have been checked by a second person or a computer algebra program.

The derivates (as coded) were check against finite differences. A co-author has checked the derivatives as presented in the paper.

**Reply to Reviewer 2:**

Reviewer comments are in Black while our response is in Blue.

1. Introduction: please cite Wehr et al. (2023) and/or Eisinger et al. (2024) to put the products described in this paper in the full context of the mission and the other products.

Done. See text lines 13-17 in the DIFF pdf.

2. Introduction: the coverage of the relevant previous literature is inadequate. Please cite papers on the existing HSRL lidars in space (Aeolus and ACDL), e.g. Flament et al. (AMT 2021), Ehlers et al. (AMT 2022) and Liu et al. (AMT 2024), discussing briefly how this work differs from your own. Regarding the algorithm, you could also cite Mason et al. (AMT 2023 "A unified synergistic...") which builds on Delanoe & Hogan (2008) and also uses an optimal estimation approach to invert the lidar signal accounting for multiple scattering.

Done: See lines 35-45 and 330-339 of the DIFF pdf.

3. L211: my understanding is that chi-squared is not the same as a cost function, since it normally only captures the deviations of the model to the observations (i.e. only the first term in your Eq. 9). If you must use chi-squared then the chi in L211 should definitely be squared.

Noted. See the revised section 2.3

4. Eq. 9: The use of two different types of "x" (one linear and one logarithmic) in the same equation is inconsistent with previous literature. You should define your state vector (boldface "x") as whatever is returned by your minimization algorithm. In your case (and also in Delanoe & Hogan 2008) it happens to contain the logarithm of physical quantities. So there is no need for a separate x^l vector - just define x appropriately. The forward model can still be written as F(x) as it is obvious that the first step of F will be to convert the logarithmic input values back to linear. The a-priori vector (x_a) also then does not need the "l" superscript.

Noted: See the revised section 2.3

Eq. 15: this doesn't make sense to me and I can't find it in Kliewer et al. An obvious problem is that the errror variance (sigma^2) of a linear variable has the units of the square of the units of that variable. But it is added to "1", which has no units! If sigma^2_xlin is the error variance of a linear quantity xlin, then the error variance of log10(xlin) is sigma^2_xlin / (xlin^2 * ln(10)^2) by the rules of error propagation, so why not use this? Since the a-priori error used in OE algorithms is often just chosen without rigorous derivation, an alternative is to specify that the error variance of the a-priori estimates of the state variables is a constant for each type of variable, indicating a constant fractional error. For example, a "factor of 2" error in xlin implies that x=log10(xlin) has an error standard deviation of log10(2)=0.3 and hence a constant error variance of 0.3^2=0.09.

There was a typo in Eq.15. The correct equation should be

S = log10(1.0+(sigma_x/mean_x)^2)

See the revised Section 2.3

6. Eq. 16: this also seems incorrect, but for a different reason. As far as I can tell your observations and forward-modelled observations in the cost function are kept in linear space, so surely the observational error covariance matrix should be kept in linear space? It is not relevant here whether the state vector is linear or logarithmic with respect to underlying physical quantities. Also, the definition in the text immediately after Eq. 16 refers to the sigma of x, not y.

Noted: The reviewer is correct. Due to some mangled editing there are some typos in the description and equations this may have caused some confusion (see also our response to Reviewer 1). Just to be clear, our state-vector is logarithm BUT the observables (y and F(x)) are linear (along with their uncertainty). Using the log form for the observables, we feel could indeed be problematic.

See the revised section 2.3.

7. L500: it would be useful to state the difference in speed between your algorithm and the quasi-small-angle algorithm of Hogan (2008).

Noted. A small discussion will be added. In a nutshell, the PPT (Platt+Tails) approach has a lower baseline cost but is O(N2) not O(N). So, for low to medium resolutions, the PPT approach is somewhat faster, but as the resolution increases, then the QSA approach is faster.

See text around line 750 in the DIFF pdf.

8. L515: There are actually two causes of tails, but this section only describes the small-angle enhancement of the Rayleigh scattering below cloud. The other occurs only in optically thick clouds and is described in numerous places, e.g. the introduction of Hogan (2008). Since the reader might be more familiar with the other mechanism, please stress here which mechanism you are talking about.

Noted: The  see text around line 670 in the DIFF pdf.

9. Fig. B1: is this just a forward model or is a retrieval also involved? It would be better/simpler if this plot simply showed the forward model for a known profile.

There is no retrieval involved here.

10. Sections C1.x: are these really needed since you are simply taking the derivative of some expressions? Just say you implemented analytically-derived derivatives.

Since this paper is intended to be ATBD-like and serve as an algorithm baseline reference, we think it is appropriate that they are included.

Minor comments:

MINOR COMMENTS

11. L4: "will provide" -> "provides".

Fixed

12. Abstract and introduction: mention that "L2a" means "level 2a", which means retrievals based on a single instrument.

Fixed. See line 6 and 20 of the DIFF pdf.

13. L11: Surely "lidar extinction, backscatter and depolarization" only applies to the A-EBD product, not to the entire A-PRO processor? This definition appears to contradict how A-PRO is defined in the abstract.

Fixed. See line 18 of the DIFF pdf.

14. L15: second comma should be a colon to introduce the list.

Fixed.

15. L29: I think the Eloranta reference should be bracketed, e.g. "...Resolution Lidar (HSRL; Eloranta 2005)".

Fixed.

16. There are a large number of places where words are incorrectly or inconsistently capitalized.  I guess these will be picked up at the copy-editing stage but I would point out L29 (etalon), L69 (signal-to-noise), captions of Figs. 1 & 6 etc (grey, black, blue and indeed all colours), L109 (microphysical, estimation), L207 (extinction), L301 (section), L315, L322, L324, L328, L334, L411, L436, L507.

Fixed.

17. Text describing Eqs. 1-3: specify that these are backscatter or extinction *coefficients* (otherwise could be cross-sections) and that the "b" terms are *attenuated* backscatter coefficients.

Fixed. See line 75 of the DIFF pdf.

18. L61: No need to double the author names: write "...due to Platt (1981) and the approach of Hogan (2008)."

Fixed.

19. L67: This line needs to work with or without the bracket, so I assume you mean "...particularly when small (or even possibly negative) values..."

Fixed. See line 104 of the DIFF pdf.

20. L74: "tens of kilometres".

Fixed.

21. L83: "Joint Standard Grid"'

Fixed.

22. L103: as a mathematical symbol, S should be in italics. Also L183.

Fixed.

23. Section 2.2 beginning L122: please be consistent in the way you refer to the steps of the algorithm in bold, including whether or not you capitalize "step", and please replace "first step" by "step 1" and "2nd step" by "step 2". Likewise section 2.3.1.

Fixed.

24. L122: define "ATB".

Fixed. See line 34 of the DIFF pdf.

25. L123: "quadratically".

Fixed.

26. Eq. 5: the "(z)" is missing from the final denominator.

Fixed.

27. L140: the quote marks are inconsistent here. In fact there is no consistent use of single or double quotes through the paper.

Fixed.

28. L175: don't start a new paragraph until all the terms in the previous equation have been defined.

Fixed.

29. There are many places where quotation marks are used unnecessarily, e.g. L181, L301, L302, L316 (twice), L347.

Fixed.

30. L186: "insure" -> "ensure".

Fixed.

31. L187: "the each"?

Fixed.

32. L194: what is the difference between steps 14 and 15? Where is step 16? Perhaps a table or flowchart would help.

Fixed. See the revised section 2.2.2

33. L219: ratio of what to what?

Fixed.

34. Eqs. 13 and 14 (and elsewhere) I think "a" should be a subscript of "R", not be written as Ra. Best to avoid two italic letters being used as one variable as it looks more like two variables being multiplied. Another example is "nl" in L228.

Fixed.

35. L276: where is step 5?

Fixed.

36. L345: "Section B" -> "Appendix B"?

Fixed.

37. Fig. 6 and others: the red boxing and arrows is the kind of thing you'd find in a presentation - wouldn't it be better to put the boxes in Fig. 5e and here have four properly labelled panels?

We disagree (the figures may appear on different pages so we prefer to keep the heat-map plat together with the panels). No action has been taken on this point.

38. Fig. 7: I suggest you use white in the total-attenuation region to indicate no retrieval, rather than black which implies you have retrieved a lidar ratio of zero.

We disagree, in context it is clear that these regions are attenuated. No action has been taken on this point.

39. Figs. 4 & 11: it appears that you are using a coloured contour to render the feature mask, but when one class jumps abruptly to another, all the intermediate colours will be rendered between them, incorrectly.

Well spotted. The A-FM data has been replotted using an image, rather than a contour, based approach.

40. L383: "lida".

Fixed.

41. L411: section A0.1 -> section A1?

Fixed.

42. L490: missing reference to a figure - did you mean Fig. B1?

Fixed.

43. L493: define ECSIM.

Fixed.

44. L538: "irregardless" -> "regardless" or "irrespective".

Fixed.

45. L550: definite "fov" and capitalize.

Fixed.

46. L556: the effective what?

Fixed.